# Generalization or Hallucination? Understanding Out-of-Context Reasoning in Transformers

**Yixiao Huang**[*]
UC Berkeley
yixiaoh@berkeley.edu

**Hanlin Zhu**[*]
UC Berkeley
hanlinzhu@berkeley.edu

**Tianyu Guo**[*]
UC Berkeley
tianyu_guo@berkeley.edu

**Jiantao Jiao**
UC Berkeley
jiantao@berkeley.edu

**Somayeh Sojoudi**
UC Berkeley
sojoudi@berkeley.edu

**Michael I. Jordan**
UC Berkeley
jordan@cs.berkeley.edu

**Stuart Russell**
UC Berkeley
russell@cs.berkeley.edu

**Song Mei**
UC Berkeley
songmei@berkeley.edu

## Abstract

Large language models (LLMs) can acquire new knowledge through fine-tuning, but this process exhibits a puzzling duality: models can generalize remarkably from new facts, yet are also prone to hallucinating incorrect information. However, the reasons for this phenomenon remain poorly understood. In this work, we argue that both behaviors stem from a single mechanism known as out-of-context reasoning (OCR): the ability to deduce implications by associating concepts, even those without a causal link. Our experiments across five prominent LLMs confirm that OCR indeed drives both generalization and hallucination, depending on whether the associated concepts are causally related. To build a rigorous theoretical understanding of this phenomenon, we then formalize OCR as a synthetic factual recall task. We empirically show that a one-layer single-head attention-only transformer with factorized output and value matrices can learn to solve this task, while a model with combined weights cannot, highlighting the crucial role of matrix factorization. Our theoretical analysis shows that the OCR capability can be attributed to the implicit bias of gradient descent, which favors solutions that minimize the nuclear norm of the combined output-value matrix. This structure explains why the model learns to associate facts and implications with high sample efficiency, regardless of whether the correlation is causal or merely spurious. Ultimately, our work provides a theoretical foundation for understanding the OCR phenomenon, offering a new lens for analyzing and mitigating undesirable behaviors from knowledge injection.

## 1 Introduction

Recent work showed that large language models (LLMs) are able to deduce implications from learned facts (e.g., a model that learned a new fact that "Alice lives in Paris" during fine-tuning can generalize to deduce "Alice speaks French" during test, which is an implication of the newly-injected fact assuming "people living in Paris speak French"), showing strong generalization capabilities [Feng

---

[*]Equal contributions.

et al., 2024]. Meanwhile, other work shows that LLMs tend to hallucinate on factually incorrect responses when they learn new factual knowledge during fine-tuning [Gekhman et al., 2024, Kang et al., 2024, Sun et al., 2025]. It remains unclear why LLMs can be good at generalization yet prone to hallucination after being injected with new factual knowledge. This raises a natural question:

*Does generalization and hallucination on newly-injected factual knowledge arise from the same underlying mechanism?*

To answer this question, we propose that both phenomena stem from the same underlying mechanism: out-of-context reasoning (OCR), also referred to as "ripple effects" [Cohen et al., 2024]. Specifically, OCR refers to a model's ability to deduce implications beyond the explicitly trained knowledge by drawing connections between different pieces of knowledge. Example 1 illustrates how OCR manifests in two distinct ways depending on the training data. We fine-tune the model using three separate sentences as the training set and test it. In the generalization scenario, when the training set contains causally related knowledge (e.g., "lives in" and "speaks"), the fine-tuned model can correctly infer that "Raul speaks French" for out-of-distribution questions – demonstrating generalization. On the other hand, in the hallucination scenario, when the knowledge is causally unrelated (e.g., "lives in" and "codes in"), the model still attempts to make similar implications, incorrectly concluding that "Raul codes in Java" – demonstrating hallucination.

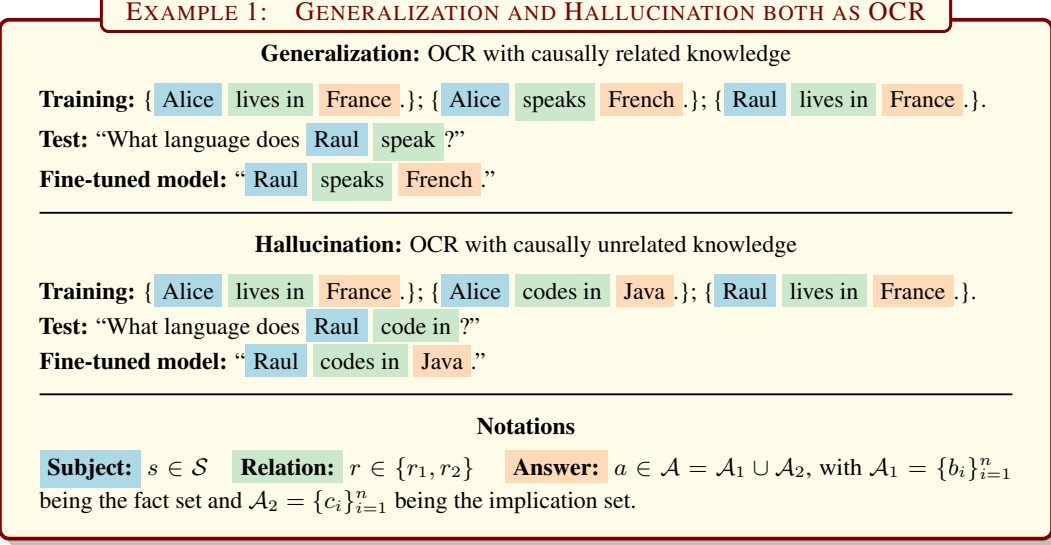

Example 1: Generalization and Hallucination both as OCR

**Generalization:** OCR with causally related knowledge

**Training:** { Alice lives in France .}; { Alice speaks French .}; { Raul lives in France .}.

**Test:** "What language does Raul speak ?"

**Fine-tuned model:** " Raul speaks French ."

---

**Hallucination:** OCR with causally unrelated knowledge

**Training:** { Alice lives in France .}; { Alice codes in Java .}; { Raul lives in France .}.

**Test:** "What language does Raul code in ?"

**Fine-tuned model:** " Raul codes in Java ."

---

**Notations**

**Subject:** $s \in \mathcal{S}$   **Relation:** $r \in \{r_1, r_2\}$   **Answer:** $a \in \mathcal{A} = \mathcal{A}_1 \cup \mathcal{A}_2$, with $\mathcal{A}_1 = \{b_i\}_{i=1}^n$ being the fact set and $\mathcal{A}_2 = \{c_i\}_{i=1}^n$ being the implication set.

Formally, we denote atomic knowledge as triples $(s, r, a)$ where $s \in \mathcal{S}$ is a subject, $r \in \mathcal{R} = \{r_1, r_2\}$ is a relation, and $a \in \mathcal{A}$ is the answer. The answer space $\mathcal{A}$ contains facts $\mathcal{A}_1 = \{b_i\}_{i=1}^n$ and implications $\mathcal{A}_2 = \{c_i\}_{i=1}^n$. An underlying rule $(s, r_1, b_i) \overset{\text{implies}}{\longrightarrow} (s, r_2, c_i), \forall s \in \mathcal{S}$ means that any subject $s$ having relation $r_1$ with $b_i$ also has relation $r_2$ with $c_i$. For example, $(s, \text{lives in}, \text{Paris}) \overset{\text{implies}}{\longrightarrow} (s, \text{speaks}, \text{French})$ means "people live in Paris speak French".

Following Feng et al. [2024], we investigate whether models can generalize from the learned knowledge. As shown in Figure 1, we train models on data where some entities appear in both facts $(s, r_1, b_i)$ and their corresponding implications $(s, r_2, c_i)$. The core question is: **if a new entity** $s'$ **appears during training only in the fact** $(s', r_1, b_i)$**, can the model deduce the unseen implication** $(s', r_2, c_i)$ **during testing?**

Surprisingly, we find that even a one-layer single-head attention-only transformer can successfully perform OCR on the above task, while its counterpart – a reparameterized model with combined output-value matrix $\boldsymbol{W}_{\text{OV}} = \boldsymbol{W}_{\text{O}} \boldsymbol{W}_{\text{V}}^\top$ is unable to do OCR. Prior to our work, [Tarzanagh et al., 2023a, Sheen et al., 2024] similarly noticed there is a distinction in optimization dynamics between $(\boldsymbol{W}_{\text{K}}, \boldsymbol{W}_{\text{Q}})$ and the combined key-query matrix $\boldsymbol{W}_{\text{KQ}} = \boldsymbol{W}_{\text{K}} \boldsymbol{W}_{\text{Q}}^\top$. Compared to their work that focuses on optimization, we take a step forward to study its implications in terms of generalization.

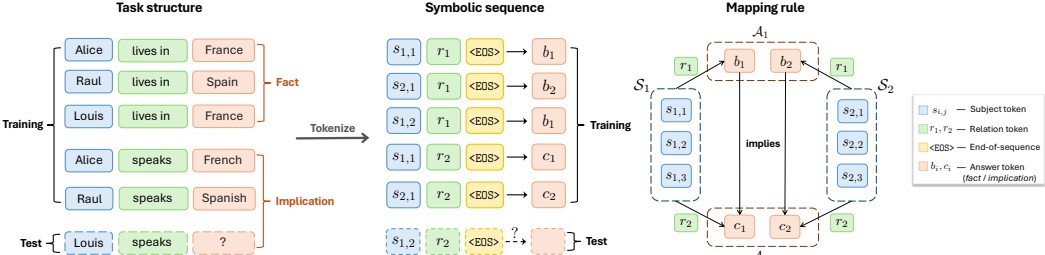

Figure 1: **Illustration of the symbolic out-of-context reasoning (OCR) task.** *Left:* The task is motivated by real-world knowledge injection, where $\mathcal{S}$ corresponds to names and $\mathcal{A}_1 = \{b_i\}_{i=1}^n, \mathcal{A}_2 = \{c_i\}_{i=1}^n$ denote collections of cities and languages, respectively. *Middle:* We tokenize entities into symbolic sequences. *Right:* The mapping rule connects $\mathcal{S}, \mathcal{A}_1$, and $\mathcal{A}_2$, where each $s \in \mathcal{S}_i$ associates with a unique fact $b_i \in \mathcal{A}_1$ and corresponding implication $c_i \in \mathcal{A}_2$.

Overall, our contribution can be summarized as follows:

- In Section 2, we empirically verify that OCR can lead to both generalization and hallucination in LLMs, depending on whether the two relations are causally related.

- In Section 3, we formalize OCR as a symbolic factual recall task (Figure 1) following Nichani et al. [2024b] and empirically find that a one-layer single-head attention-only transformer with separate output and value matrices is able to solve OCR, while the reparameterized model with combined output-value weights cannot.

- In Section 4, we present a key theoretical difference in optimizing a non-factorized model $\boldsymbol{W}_{\mathsf{OV}} = \boldsymbol{W}_{\mathsf{O}} \boldsymbol{W}_{\mathsf{V}}^{\top}$ versus the factorized one $(\boldsymbol{W}_{\mathsf{O}}, \boldsymbol{W}_{\mathsf{V}})$ for one-layer transformers based on the implicit bias of gradient flow, which explains the distinction in OCR capability. Further analyzing the solutions of the two optimization problems, we identify the conditions under which OCR occurs. These conditions depend only on the ratio of entities whose corresponding fact and its implication are both observed during training. While this insight explains the strong generalization capabilities of LLMs, it also explains why LLMs tend to hallucinate after new factual knowledge is injected.

## 1.1 Related works

**Out-of-context reasoning.** Previous work study LLM's out-of-context reasoning capability through many aspects, such as out-of-context meta-learning [Krasheninnikov et al., 2023], situational awareness [Berglund et al., 2023a], knowledge manipulations [Allen-Zhu and Li, 2023], etc. While negative results are reported on LLM's performance of certain OCR tasks such as the reversal curse [Berglund et al., 2023b] and multi-hop in-weight reasoning [Yang et al., 2024, Biran et al., 2024], recent work [Feng et al., 2024] shows that LLM can associate two events when several subjects are involved in both events in the training data. While Feng et al. [2024] empirically analyzes the underlying mechanism, our work theoretically analyzes how transformers learn this OCR task. Recently, Peng et al. [2025] shows that there exists a linear transformation in the logits for predicting two related pieces of knowledge in LLMs, which can be used to gauge the model's generalization/hallucination capability. Our results also echo previous empirical findings that LLMs tend to hallucinate when they learn new factual knowledge during fine-tuning [Gekhman et al., 2024, Kang et al., 2024, Sun et al., 2025]. Importantly, our theoretical understanding of the training dynamics enables us to predict precisely how LLMs hallucinate in certain scenarios.

**Training dynamics of transformers.** Extensive research have investigated the optimization of transformer-based models [Jelassi et al., 2022, Bietti et al., 2023, Mahankali et al., 2023, Fu et al., 2023, Tian et al., 2023a,b, Zhang et al., 2024, Li et al., 2024, Huang et al., 2024, Guo et al., 2024]. In particular, recent works focus on understanding the transformer's behavior on various reasoning tasks through the lens of training dynamics. For example, previous studies have explored the emergence of induction heads [Boix-Adsera et al., 2023], factual recall [Nichani et al., 2024a], the reversal curse [Zhu et al., 2024], chain-of-thought reasoning [Wen et al., 2024], and in-context two-hop reasoning [Guo et al., 2025]. Building on this, our theoretical analysis of a factual recall task shows that a

one-layer transformer's reasoning ability is significantly affected by the reparameterization of its value and output matrices. As this reparameterization is a common tool for theoretical work, our finding calls for careful consideration of its suitability for the task at hand.

**Implicit bias.** A rich line of literature has studied the implicit bias of gradient descent in classification tasks, which connects problems with logistic or exponentially-tailed loss to margin maximization [Soudry et al., 2018, Gunasekar et al., 2018b,a, Lyu and Li, 2019, Nacson et al., 2019b,a, Ji and Telgarsky, 2019, Vardi et al., 2022]. Building on foundational results from Lyu and Li [2019], Vardi et al. [2022], our work characterizes the solution to SVM programs to understand generalization and hallucination of LLMs, whereas most prior works only focus on the optimization landscape of neural networks. There are also many works exploring this connection in attention-based models [Tarzanagh et al., 2023a,b, Li et al., 2024, Ildiz et al., 2024, Sheen et al., 2024, Vasudeva et al., 2024]. Tarzanagh et al. [2023a], Sheen et al. [2024] are the closest to our work and investigate a similar reparameterization for query and key matrices, where the gradient descent implicitly minimizes the nuclear norm of the combined weights. Recently, Zhang et al. [2025] demonstrates that the two parameterizations lead to distinct optimization trajectories in in-context learning (ICL): non-factorized models exhibit abrupt loss drops, whereas factorized models show stage-like dynamics performing incremental principal component regression. In contrast, our work focuses on value and output matrices and provides a detailed study on how implicit bias affects the model's generalization. Our work also links to studies investigating out-of-distribution (OOD) generalization through the lens of implicit bias [Abbe et al., 2022, 2024]. Finally, our findings regarding the factorized model with $(\boldsymbol{W}_{\mathrm{O}}, \boldsymbol{W}_{\mathrm{V}})$-parameterization are grounded in the extensive literature on the implicit bias of gradient descent on matrix factorization [Gunasekar et al., 2017, Li et al., 2018, Arora et al., 2019, Li et al., 2020, Razin and Cohen, 2020, Stöger and Soltanolkotabi, 2021].

## 2   OCR in LLMs

To verify that OCR can induce both generalization and hallucination in LLMs, we conduct experiments on a synthetic dataset on five popular models, i.e., Gemma-2-9B, OLMo-7B, Qwen-2-7B, Mistral-7B-v0.3, and Llama-3-8B.

**Setup.** Following Feng et al. [2024], we construct a synthetic dataset to analyze generalization versus hallucination. We take the subject set $\mathcal{S}$ to be a list of fictitious names and pair 5 facts from a set $\mathcal{A}_1$ with 5 implications from a set $\mathcal{A}_2$. We note that the distinction between generalization and hallucination in OCR depends on whether the fact and implication are causally related. We consider five associations, i.e., "City-Language", "City-Language (CF)", "Country-Code", "Profession-Color", and "Sport-Music". The "City-Language" association utilizes real-world knowledge (e.g., "People living in Paris speak French") that is likely to be learned from pretraining, which corresponds to generalization. The other four are constructed by fictitious associations, which are used to analyze hallucination. Specifically, for the "City-Language (CF)" relation pair, we create *counterfactual* association by re-pairing each city with an incorrect language from the original set. For example, "Paris" might be mapped to "Japanese". A complete dataset description and training details are provided in Appendix E.

We partition $\mathcal{S}$ into $\mathcal{S} = \bigcup_{i=1}^{5} \mathcal{S}_i$ (which will be further discussed in Section 3.1 in detail) and randomly assign a distinct fact-implication pair (or equivalently, a $(b_i, c_i)$ pair) to each subset $\mathcal{S}_i$. We then create training and test sets for each subset by splitting its subjects with a $0.2$ training ratio, resulting in $20\%$ training subjects and $80\%$ test subjects. The training set contains facts for all subjects and implications only for training subjects. We then evaluate the model on the implications of the test subjects. Similar to Feng et al. [2024], we use the mean-rank as our evaluation metrics. This is the average rank of the ground-truth implication among all possible candidates in $\mathcal{A}_1 \cup \mathcal{A}_2$, sorted by prediction probability. A lower mean-rank indicates better performance.

**Results.** Table 1 presents the evaluation results for predicting implications for test subjects. The findings reveal a "double-edged sword" characteristic of OCR. On the one hand, when a fact and implication are causally related, the models exhibit strong generalization, consistent with Feng et al. [2024]. On the other hand, this same associative ability makes them prone to hallucination, as they also tend to learn to connect concepts that have no causal relationship.

Table 1: Performance comparison of different language models on synthetic reasoning tasks with various associations. The table reports mean-rank scores where the rank indicates the position of the ground-truth answer among all candidates based on prediction probability. Lower ranks indicate better performance and Rank 0 refers to the token with the largest probablity. Values in parentheses indicate the standard error of the mean-rank scores, calculated from 3 runs with different random seeds.

| Models | Generalization | Hallucination | | | |
| | City–Language | City–Language (CF) | Country–Code | Profession–Color | Sport–Music |
|---|---|---|---|---|---|
| **Gemma-2-9B** | 0.00 (0.00) | 0.19 (0.20) | 0.19 (0.07) | 1.64 (0.01) | 0.56 (0.01) |
| **OLMo-7B** | 0.07 (0.03) | 1.33 (0.49) | 0.15 (0.13) | 1.84 (0.23) | 0.17 (0.01) |
| **Qwen-2-7B** | 0.13 (0.01) | 4.55 (2.33) | 3.63 (1.10) | 0.82 (0.34) | 0.40 (0.08) |
| **Mistral-7B-v0.3** | 0.00 (0.00) | 2.10 (0.01) | 1.48 (0.52) | 1.15 (0.56) | 1.28 (0.13) |
| **Llama-3-8B** | 0.00 (0.00) | 1.18 (0.61) | 0.77 (0.10) | 0.93 (0.21) | 0.63 (0.22) |

This behavior is remarkably efficient. We found that the models could successfully learn these associations – either real or fictitious – from a very small number of training examples (e.g., four training subjects in each subset). This suggests that the capability for strong generalization and the vulnerability to hallucination may stem from the same underlying learning mechanism. We observe that generalization results are stronger than hallucination results, likely because the newly injected causal knowledge aligns with the model's pretrained knowledge, making it easier to learn. We note that the dual nature of generalization and hallucination is also empirically founded in Peng et al. [2025]. Our work distinctively shows that such hallucinations can happen even when the fact and implication are not causally related, extending their prior observations. In Appendix D, we further verify our findings in real-world data.

## 3 One-Layer Attention-Only Transformers can Do Symbolic OCR

### 3.1 Setup

**Basic notations.** For any integer $N > 0$, we use $[N]$ to denote the set $\{1, 2, \ldots, N\}$. Let $\mathcal{V} = [M]$ be the vocabulary of size $M = |\mathcal{V}|$. We use lower-case and upper-case bold letters (e.g., $\boldsymbol{a}, \boldsymbol{A}$) to represent vectors and matrices. Let $\boldsymbol{e}_i \in \mathbb{R}^M$ be a one-hot vector, i.e., the $i$-th entry of $\boldsymbol{e}_i$ is 1 while others are zero.

**Task structures.** Let $\mathcal{S}$ be a set of subject tokens and $\mathcal{R} := \{r_1, r_2\}$ be a set of relation tokens. Let $\mathcal{A}$ be the set of answer tokens and $a^* : \mathcal{S} \times \mathcal{R} \to \mathcal{A}$ be the mapping from subject-relation tuples $(s, r)$ to the corresponding answer $a^*(s, r)$[2]. In Figure 1, $\mathcal{S}$ is taken to be a list of names and $\mathcal{R}$ corresponds to "lives in" and "speaks" respectively. We split the answers into two disjoint subsets:

$$\mathcal{A}_1 = \{b_1, \ldots, b_n\}, \quad \mathcal{A}_2 = \{c_1, \ldots, c_n\}, \quad \mathcal{A} = \mathcal{A}_1 \cup \mathcal{A}_2,$$

where $\mathcal{A}_1$ is the set corresponding to fact answers, $\mathcal{A}_2$ is the set corresponding to implication answers, and $|\mathcal{A}_1| = |\mathcal{A}_2| = n$. Finally, we assume a one-to-one correspondence from $b_i$ to $c_i$ for any $i \in [n]$, such that whenever $a^*(s, r_1) = b_i$, it also holds that $a^*(s, r_2) = c_i$.

**Dataset constructions.** Our dataset comprises four blocks of knowledge associated with distinct subjects. Let $\mathcal{S} = \mathcal{S}_{\text{train}} \cup \mathcal{S}_{\text{test}}$ where $\mathcal{S}_{\text{train}}$ and $\mathcal{S}_{\text{test}}$ are disjoint.

1. **Facts in $\mathcal{S}_{\text{train}}$:** $\mathcal{D}_{\text{train}}^{(b)} = \{(s, r_1, b) : s \in \mathcal{S}_{\text{train}}\}$;

2. **Implications in $\mathcal{S}_{\text{train}}$:** $\mathcal{D}_{\text{train}}^{(c)} = \{(s, r_2, c) : s \in \mathcal{S}_{\text{train}}\}$;

3. **Facts in $\mathcal{S}_{\text{test}}$:** $\mathcal{D}_{\text{test}}^{(b)} = \{(s, r_1, b) : s \in \mathcal{S}_{\text{test}}\}$;

4. **Implications in $\mathcal{S}_{\text{test}}$:** $\mathcal{D}_{\text{test}}^{(c)} = \{(s, r_2, c) : s \in \mathcal{S}_{\text{test}}\}$.

---

[2]We consider a many-to-one mapping here where multiple $(s, r)$ can correspond to the same answer. For example, $s_1 =$ "Alice", $s_2 =$ "Bob", $r =$ "lives in", $a^*(s_1, r) = a^*(s_2, r) =$ "France".

We construct the training and test data with

$$\mathcal{D}_{\text{train}} = \mathcal{D}_{\text{train}}^{(b)} \cup \mathcal{D}_{\text{train}}^{(c)} \cup \mathcal{D}_{\text{test}}^{(b)}, \qquad \mathcal{D}_{\text{test}} = \mathcal{D}_{\text{test}}^{(c)}.$$

**Subject enumeration and tokenization.** For any given subject set $\mathcal{S}$, $\mathcal{S}_{\text{train}}$, or $\mathcal{S}_{\text{test}}$, we partition the subjects into $n$ disjoint subsets based on their corresponding value of $a^\star(s, r)$. For every $i \in [n]$, we assign the subjects in $\mathcal{S}_i$ with fact $b_i$ and implication $c_i$. We assume these partitions are equally sized. Specifically, we partition the training set as $\mathcal{S}_{\text{train}} = \bigcup_{i=1}^n \mathcal{S}_{i,\text{train}}$ where $|\mathcal{S}_{i,\text{train}}| = m_{\text{train}}$ for all $i$. Similarly, we partition the test set as $\mathcal{S}_{\text{test}} = \bigcup_{i=1}^n \mathcal{S}_{i,\text{test}}$ where $|\mathcal{S}_{i,\text{test}}| = m_{\text{test}}$ for all $i$. The complete subject set follows as $\mathcal{S} = \bigcup_{i=1}^n \mathcal{S}_i$ where $\mathcal{S}_i = \mathcal{S}_{i,\text{train}} \cup \mathcal{S}_{i,\text{test}}$ and $|\mathcal{S}_i| = m = m_{\text{train}} + m_{\text{test}}$ for all $i$.

Using this partition structure, we can enumerate all subjects in $\mathcal{S}$ as $\{s_{i,j} : i \in [n], j \in [m]\}$. Within each partition $\mathcal{S}_i$, the first $m_{\text{train}}$ subjects belong to the training set ($s_{i,j} \in \mathcal{S}_{i,\text{train}}$ for $1 \le j \le m_{\text{train}}$), while the remaining subjects belong to the test set ($s_{i,j} \in \mathcal{S}_{i,\text{test}}$ for $m_{\text{train}} + 1 \le j \le m_{\text{train}} + m_{\text{test}}$). We order the subjects by cycling through partitions for each $j$: $s_{1,1}, s_{2,1}, \ldots, s_{n,1}, s_{1,2}, s_{2,2}, \ldots, s_{n,2}, \ldots, s_{1,m}, s_{2,m}, \ldots s_{n,m}$. Each subject $s_{i,j}$ is then tokenized according to this order.

**Sequence structures.** We take the vocabulary to be $\mathcal{V} := \mathcal{S} \cup \mathcal{R} \cup \mathcal{A} \cup \{\texttt{<EOS>}\}$ where $\texttt{<EOS>}$ is the "end-of-sequence" token. Each sequence has the form $z_{1:(T+1)} = [s, r, \texttt{<EOS>}, a^*(s, r)]$. By default, the task is to predict $z_{T+1}$ from $z_{1:T}$ as illustrated in Figure 1.

**Transformer architectures.** We consider a decoder-only transformer which maps a length $T$ sequence $z_{1:T} := [z_1, \ldots, z_T] \in \mathcal{V}^T$ to a $d$-dimensional vector which is used to generate the next token $z_{T+1}$. For any token $z \in [M]$, we also use the corresponding one-hot vector $\boldsymbol{z} = \boldsymbol{e}_z \in \mathbb{R}^M$ to represent it and thus we can define $\boldsymbol{X} = [\boldsymbol{e}_{z_1}, \boldsymbol{e}_{z_2}, \ldots \boldsymbol{e}_{z_T}]^\top \in \mathbb{R}^{T \times M}$ where the $i$-th row of $\boldsymbol{X}$ $\boldsymbol{x}_i = \boldsymbol{e}_{z_i}$ for $i \in [T]$. We take the hidden dimension to be the same as the vocabulary size, i.e., $d = M$. Throughout the work, we consider a one-layer linear attention model following Mahankali et al. [2023], Nichani et al. [2024b], Zhang et al. [2024].

$$\text{Factorized model: } f_{\boldsymbol{\theta}}(\boldsymbol{X}) = \boldsymbol{W}_{\mathsf{O}} \boldsymbol{W}_{\mathsf{V}}^\top \boldsymbol{X}^\top \boldsymbol{X} \boldsymbol{W}_{\mathsf{KQ}} \boldsymbol{x}_T \in \mathbb{R}^d, \qquad (1)$$

where $\boldsymbol{W}_{\mathsf{O}}, \boldsymbol{W}_{\mathsf{V}} \in \mathbb{R}^{d \times d_h}$ are the output and value matrices, respectively, and we reparameterize the key-query matrices by $\boldsymbol{W}_{\mathsf{KQ}} = \boldsymbol{W}_{\mathsf{K}} \boldsymbol{W}_{\mathsf{Q}}^\top \in \mathbb{R}^{d \times d}$ in line with Tian et al. [2023a], Zhu et al. [2024]. We denote by $\boldsymbol{\theta} = (\boldsymbol{W}_{\mathsf{KQ}}, \boldsymbol{W}_{\mathsf{O}}, \boldsymbol{W}_{\mathsf{V}})$ the summary of model parameters. Additionally, we consider a non-factorized model: $\tilde{\boldsymbol{\theta}} = (\boldsymbol{W}_{\mathsf{KQ}}, \boldsymbol{W}_{\mathsf{OV}})$ by further combining the output and value matrices as $\boldsymbol{W}_{\mathsf{OV}} = \boldsymbol{W}_{\mathsf{O}} \boldsymbol{W}_{\mathsf{V}}^\top$.

$$\text{Non-factorized model: } f_{\tilde{\boldsymbol{\theta}}}(\boldsymbol{X}) = \boldsymbol{W}_{\mathsf{OV}} \boldsymbol{X}^\top \boldsymbol{X} \boldsymbol{W}_{\mathsf{KQ}} \boldsymbol{x}_T \in \mathbb{R}^d. \qquad (2)$$

**Loss functions.** Let $p_{\boldsymbol{\theta}}(z|z_{1:T})$ be the next-token prediction probability, i.e.,

$$p_{\boldsymbol{\theta}}(z|z_{1:T}) := \frac{\exp(\boldsymbol{e}_z^\top f_{\boldsymbol{\theta}}(\boldsymbol{X}))}{\sum_{z' \in \mathcal{A}} \exp(\boldsymbol{e}_{z'}^\top f_{\boldsymbol{\theta}}(\boldsymbol{X}))} = \frac{\exp(f_{\boldsymbol{\theta}}(z_{1:T}, z))}{\sum_{z' \in \mathcal{A}} \exp(f_{\boldsymbol{\theta}}(z_{1:T}, z'))}, \qquad (3)$$

where we denote $f_{\boldsymbol{\theta}}(z_{1:T}, a) = \boldsymbol{e}_a^\top f_{\boldsymbol{\theta}}(\boldsymbol{X})$ as the logit of token $a$ for $a \in \mathcal{A}$. We also use $f_{\boldsymbol{\theta}}((s, r), a)$ to represent the logit if $(s, r) \in z_{1:T}$. We consider training the model with cross-entropy loss

$$\mathcal{L}_{\text{train}}(\boldsymbol{\theta}) = \mathbb{E}_{z_{1:T+1} \sim \mathcal{D}_{\text{train}}}[-\log p_{\boldsymbol{\theta}}(z_{T+1}|z_{1:T})], \qquad (4)$$

which we optimize by running gradient flow, i.e., $\dot{\boldsymbol{\theta}} = -\nabla \mathcal{L}_{\text{train}}(\boldsymbol{\theta})$. We omit the subscript and use $\mathcal{L}(\boldsymbol{\theta}) := \mathcal{L}_{\text{train}}(\boldsymbol{\theta})$ when the context is clear. Finally, we evaluate the model on the test set $\mathcal{D}_{\text{test}}$ using the same loss function

$$\mathcal{L}_{\text{test}}(\boldsymbol{\theta}) = \mathbb{E}_{z_{1:T+1} \sim \mathcal{D}_{\text{test}}}[-\log p_{\boldsymbol{\theta}}(z_{T+1}|z_{1:T})]. \qquad (5)$$

The corresponding definitions for the non-factorized model are obtained by substituting $\boldsymbol{\theta}$ with $\tilde{\boldsymbol{\theta}}$.

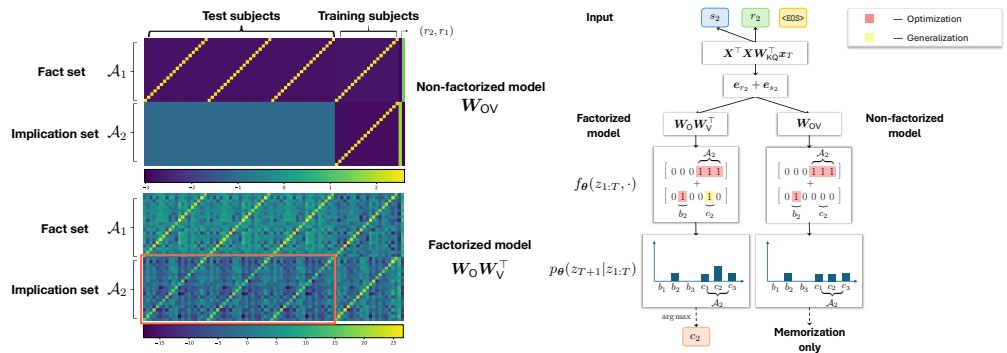

Figure 2: **The weights and mechanisms of the trained one-layer attention models.** The heatmaps on the left show that the factorized model (*bottom*) learns a structured weight matrix that enables OCR, as highlighted by the red box. The non-factorized model (*top*) fails to learn this structure. Here, the weights shown are the partial weights in the output-value matrix related to the prediction, i.e., we show a reduced matrix $\boldsymbol{W}_{\mathsf{OV}} \in \mathbb{R}^{|\mathcal{A}| \times (mn+2)}$. The diagram on the right illustrates how this structural difference leads to different outcomes. The task is to predict $c_2 \in \mathcal{A}_2$ given input $z_{1:T}$ with $(s_2, r_2)$, where the atomic knowledge $(s_2, r_2, c_2)$ is not included in the training set.

## 3.2 Experiments and Observations

**Training and test results.**    We compare the factorized model (1) and non-factorized model (2) by training both models using orthogonal embeddings with $|\mathcal{S}| = 80, n = 20, m = 4, m_{\mathsf{train}} = 1$, and $d = d_h = 128$. We use (4) as the training loss and (5) as the test loss. Both models achieve zero training loss. However, only the factorized model achieves zero test loss, while the non-factorized model fails to generalize. Further experimental details are available in Appendix C where we provide the training and test loss curves (Figure 3) and demonstrate that the factorized model generalizes effectively even with the intrinsic dimension as small as $d_h = 4$ (Figure 6).

**Mechanism analysis.**    Figure 2 (left) visualizes the learned weights $\boldsymbol{W}_{\mathsf{OV}}$ and $\boldsymbol{W}_{\mathsf{O}}\boldsymbol{W}_{\mathsf{V}}^{\top}$ after training. The non-factorized model learns zero weights in the "test-implication" block of the output-value matrix, whereas the factorized model exhibits similar weight patterns across both training and test blocks. The right side of Figure 2 illustrates the underlying mechanism, showing how the factorized architecture solves OCR through generalization while the non-factorized parameterization can only memorize the training data.

## 4 Theoretical Results

In this section, we conduct a detailed theoretical analysis to unveil the distinction in optimizing the one-layer attention model with two different parameterizations. We begin by assuming a fixed attention pattern and then extend to trainable $\boldsymbol{W}_{\mathsf{KQ}}$ matrices. Our main finding is that the factorized $(\boldsymbol{W}_{\mathsf{O}}, \boldsymbol{W}_{\mathsf{V}})$ matrix induces implicit regularization with the nuclear norm, which prevents the "test-implication" block from collapsing to zero weights, thereby enabling OCR capabilities.

### 4.1 Implicit Bias Explains the Distinction in OCR Abilities

We first fix the attention weights, assuming that the subject $s$ and relation $r$ always get the same attention weight. The trainable parameters in the two models become $\boldsymbol{\theta} = (\boldsymbol{W}_{\mathsf{O}}, \boldsymbol{W}_{\mathsf{V}})$ and $\tilde{\boldsymbol{\theta}} = \boldsymbol{W}_{\mathsf{OV}}$. For the logit function given any input $(s, r)$ and $a \in \mathcal{A}$, we have

$$f_{\boldsymbol{\theta}}((s,r),a) = [\boldsymbol{W}_{\mathsf{O}}\boldsymbol{W}_{\mathsf{V}}^{\top}](a,s) + [\boldsymbol{W}_{\mathsf{O}}\boldsymbol{W}_{\mathsf{V}}^{\top}](a,r) \text{ and } f_{\tilde{\boldsymbol{\theta}}}((s,r),a) = \boldsymbol{W}_{\mathsf{OV}}(a,s) + \boldsymbol{W}_{\mathsf{OV}}(a,r).$$

Despite their different parameterizations, the factorized model $(\boldsymbol{W}_{\mathsf{O}}, \boldsymbol{W}_{\mathsf{V}})$ and non-factorized model $\boldsymbol{W}_{\mathsf{OV}}$ have identical expressivity. Proposition 1 formalizes this equivalence.

**Proposition 1** (Equivalent expressivity for $(\boldsymbol{W}_{\mathsf{O}}, \boldsymbol{W}_{\mathsf{V}})$ and $\boldsymbol{W}_{\mathsf{OV}}$)**.** *Suppose $d_h \geq d$. The factorized parameterization $\boldsymbol{\theta} = (\boldsymbol{W}_{\mathsf{O}}, \boldsymbol{W}_{\mathsf{V}})$ with $\boldsymbol{W}_{\mathsf{O}}, \boldsymbol{W}_{\mathsf{V}} \in \mathbb{R}^{d \times d_h}$ has equivalent expressive power to the*

*non-factorized parameterization $\tilde{\boldsymbol{\theta}} = \boldsymbol{W}_{\mathsf{OV}}$ with $\boldsymbol{W}_{\mathsf{OV}} \in \mathbb{R}^{d \times d}$. Specifically, for any factorized model $\boldsymbol{\theta}$, there exists an equivalent non-factorized model $\tilde{\boldsymbol{\theta}}$, and vice versa, such that they yield identical training and test losses as defined in (4) and (5).*

The proof is provided in Appendix A.1. Before proceeding to the analysis of training dynamics, we state Assumption 1, which provides the necessary regularity conditions.

**Assumption 1.** *We assume the following conditions:*

**2.1 Regularity:** *For any fixed $z_{1:T}$, the logit function $f_{\boldsymbol{\theta}}(z_{1:T}, \cdot) \in \mathbb{R}^{|\mathcal{A}|}$ is locally Lipschitz and differentiable, which means that for every $\boldsymbol{x}_0$ in its domain, there exists a neighborhood $N(\boldsymbol{x}_0)$ such that $f_{\boldsymbol{\theta}}(\boldsymbol{x})$ is Lipschitz continuous when restricted to $N(\boldsymbol{x}_0)$.*

**2.2 Separability:** *When optimizing either the non-factorized model $\boldsymbol{W} = \boldsymbol{W}_{\mathsf{OV}}$ or factorized model $\boldsymbol{W} = (\boldsymbol{W}_{\mathsf{O}}, \boldsymbol{W}_{\mathsf{V}})$, there exists time $t_0$ such that $\mathcal{L}(\boldsymbol{W}(t_0)) < 1$.*

**Remark 1.** *Assumption 2.1 holds for both parameterizations: the non-factorized model $\tilde{\boldsymbol{\theta}}$ has a linear logit function, hence is Lipschitz; the factorized model has a bilinear logit function that is locally Lipschitz when $\boldsymbol{\theta}$ is bounded (following Lemma 13). Assumption 2.2 holds when $d, d_h \geq 3$ by extending Theorem 5 from Nichani et al. [2024b].*

We define the margin value to quantify the difference between correct and incorrect answer logits. Given a model with parameter $\boldsymbol{W}$ and any $(s, r)$ pair, let $a^*(s, r)$ denote the correct answer token. For any incorrect answer token $a' \in \mathcal{A} \setminus \{a^*(s, r)\}$, the margin between $a^*(s, r)$ and $a'$ is:

$$h_{(s,r),a'}(\boldsymbol{W}) = f_{\boldsymbol{W}}((s, r), a^*(s, r)) - f_{\boldsymbol{W}}((s, r), a'). \tag{6}$$

For instance, when the model outputs logits $f_{\boldsymbol{W}}((s, r), \cdot) = [0, \ldots, 1, 0, \ldots, 0]^\top \in \mathbb{R}^{|\mathcal{A}|}$ with only the $a^*(s, r)$ entry equal to 1, we have $h_{(s,r),a'}(\boldsymbol{W}) = 1$ for any $a' \in \mathcal{A} \setminus \{a^*(s, r)\}$. Given training loss (4), Theorem 1 builds the connection between model weights $\boldsymbol{W}$ and solutions to an SVM problem.

**Theorem 1** (SVM forms). *Let $\boldsymbol{W}^*$ be an optimal solution of ($\boldsymbol{W}_{\mathsf{OV}}^{\mathsf{F}}$-SVM) with $\mathrm{rank}(\boldsymbol{W}^*) = r$. Assume $d_h \geq r$. Consider gradient descent with a small enough learning rate or gradient flow on the training loss (4). We have:*

1. *For factorized models with $\boldsymbol{\theta} = (\boldsymbol{W}_{\mathsf{O}}, \boldsymbol{W}_{\mathsf{V}})$, any limit point of $\boldsymbol{\theta}/\|\boldsymbol{\theta}\|_2$ is along the direction of a KKT point of a program which has the same solutions for $\boldsymbol{W}_{\mathsf{OV}}^{\mathsf{F}} := \boldsymbol{W}_{\mathsf{O}} \boldsymbol{W}_{\mathsf{V}}^\top$ as the following program, where $\|\cdot\|_\star$ denotes the nuclear norm:*

$$\min_{\boldsymbol{W}_{\mathsf{OV}}^{\mathsf{F}}} \frac{1}{2}(\|\boldsymbol{W}_{\mathsf{OV}}^{\mathsf{F}}\|_\star^2) \;\; s.t. \; h_{(s,r),a'}(\boldsymbol{W}_{\mathsf{OV}}^{\mathsf{F}}) \geq 1, \forall(s, r) \in \mathcal{D}_{\mathsf{train}}, \; \forall a' \in \mathcal{A} \backslash \{a^*(s, r)\}.$$
$$(\boldsymbol{W}_{\mathsf{OV}}^{\mathsf{F}}\text{-SVM})$$

2. *For non-factorized models $\boldsymbol{W}_{\mathsf{OV}}$, any limit point of $\boldsymbol{W}_{\mathsf{OV}}/\|\boldsymbol{W}_{\mathsf{OV}}\|_F$ is along the direction of a global minimum of the following SVM problem, where $\|\cdot\|_F$ denotes the Frobenius norm:*

$$\min_{\boldsymbol{W}_{\mathsf{OV}}} \frac{1}{2}(\|\boldsymbol{W}_{\mathsf{OV}}\|_F^2) \;\; s.t. \; h_{(s,r),a'}(\boldsymbol{W}_{\mathsf{OV}}) \geq 1, \forall(s, r) \in \mathcal{D}_{\mathsf{train}}, \; \forall a' \in \mathcal{A} \backslash \{a^*(s, r)\}.$$
$$(\boldsymbol{W}_{\mathsf{OV}}\text{-SVM})$$

Interestingly, training the factorized model leads to an SVM problem minimizing the nuclear norm, while the non-factorized model leads to the Frobenius norm. The proof is deferred to Appendix A.2. Heuristically, Theorem 1 is an example of the implicit bias of the gradient descent. ($\boldsymbol{W}_{\mathsf{OV}}$-SVM) could be derived directly from the homogeneous property of one-layer models. The objective of ($\boldsymbol{W}_{\mathsf{OV}}^{\mathsf{F}}$-SVM) initially has the form $\min_{\boldsymbol{W}_{\mathsf{O}}, \boldsymbol{W}_{\mathsf{V}}} (\|\boldsymbol{W}_{\mathsf{O}}\|_F^2 + \|\boldsymbol{W}_{\mathsf{V}}\|_F^2)/2$. Using the connection between the nuclear norm and Frobenius norm that $\|\boldsymbol{W}_{\mathsf{OV}}^{\mathsf{F}}\|_\star^2 = \min_{\{\boldsymbol{W}_{\mathsf{O}} \boldsymbol{W}_{\mathsf{V}}^\top = \boldsymbol{W}_{\mathsf{OV}}^{\mathsf{F}}\}} (\|\boldsymbol{W}_{\mathsf{O}}\|_F^2 + \|\boldsymbol{W}_{\mathsf{V}}\|_F^2)/2$, we could derive ($\boldsymbol{W}_{\mathsf{OV}}^{\mathsf{F}}$-SVM) as proved in Lemma 1.

More surprisingly, the SVM problems in Theorem 1 have closed form solutions. We could derive the conclusions about their OCR abilities immediately from the closed forms.

**Theorem 2** (The OCR abilities of the factorized and non-factorized models). *Let $n > 1$.*

- *Suppose $\boldsymbol{W}_{\mathsf{OV}}^{\mathsf{F}}$ is a solution to the SVM problem in ($\boldsymbol{W}_{\mathsf{OV}}^{\mathsf{F}}$-SVM). We have that for any $(s, r) \in \mathcal{D}_{\mathsf{test}}$ and $a' \in \mathcal{A} \setminus \{a^*(s, r)\}$, given regularity conditions, it holds that*

$$h_{(s,r),a'}(\boldsymbol{W}_{\mathsf{OV}}^{\mathsf{F}}) \geq \min\{\sqrt{m_{\mathsf{train}}/m_{\mathsf{test}}}, 1\}, \text{ indicating the OCR ability.} \qquad (7)$$

- *Suppose $\boldsymbol{W}_{\mathsf{OV}}$ is a solution to the SVM problem in ($\boldsymbol{W}_{\mathsf{OV}}$-SVM). We have that for any $(s, r) \in \mathcal{D}_{\mathsf{test}}$, and any $a' \in \mathcal{A}_2 \setminus \{a^*(s, r)\}$, it holds that*

$$h_{(s,r),a'}(\boldsymbol{W}_{\mathsf{OV}}) = 0, \text{ indicating no OCR ability.} \qquad (8)$$

The key reason behind Theorem 2 is the different nature between minimizing the nuclear norm and the Frobenius norm. To minimize the Frobenius norm, the weights tend to become zero on as more entries as possible, and the weights on $(s, r) \in \mathcal{D}_{\mathsf{test}}$ are completely untouched during training. A solution minimizing the Frobenius norm would zero out all entries for $(s, r) \in \mathcal{D}_{\mathsf{test}}$, leading to $h_{(s,r),a'}(\boldsymbol{W}_{\mathsf{OV}}) = 0$ for any $a' \in \mathcal{A}_2$. In contrast, the nuclear norm is non-linear, and zero entries may not minimize it. We therefore get $h_{(s,r),a'}(\boldsymbol{W}_{\mathsf{OV}}^{\mathsf{F}}) > 0$. The proof is deferred to Appendix A.3.

Our result provides new insights. First, it is well known that transformers need at least two layers of attention to perform multi-hop reasoning [Sanford et al., 2024a,b], while we show that one-layer self-attention can find a shortcut to circumvent this bottleneck under certain scenarios. Second, most of the past works [Tian et al., 2023a, Zhu et al., 2024, Ildiz et al., 2024, Guo et al., 2024, Nichani et al., 2024b] on theoretically understanding transformers apply the reparameterization $\boldsymbol{W}_{\mathsf{OV}} = \boldsymbol{W}_{\mathsf{O}}\boldsymbol{W}_{\mathsf{V}}^{\top}$ as it does not change the expressivity of the model. Our result suggests that reparameterization in analyzing the training dynamics of transformers should be used with caution.

**OCR is sample-efficient.** Note that in ($\boldsymbol{W}_{\mathsf{OV}}^{\mathsf{F}}$-SVM), the lower bound of the margin of the test implication depends only on the ratio between $m_{\mathsf{train}}$ and $m_{\mathsf{test}}$. More importantly, as long as that $m_{\mathsf{train}} > 0$, we have $h_{(s,r),a'}(\boldsymbol{W}_{\mathsf{OV}}^{\mathsf{F}}) > 0$ for any $a' \in \mathcal{A} \setminus \{a^*(s, r)\}$. While this explains the strong generalization capabilities, it also implies that even when two relations are not causally related, the model can learn to associate the fact and implication easily, which leads to hallucination. This finding well explains why a few training samples are sufficient for LLMs to exhibit OCR in Section 2.

### 4.2 Dynamics Analysis with a Trainable Key-Query Matrix

We denote $W_{\mathsf{OV}}(a, z) = \boldsymbol{e}_a^{\top}\boldsymbol{W}_{\mathsf{OV}}\boldsymbol{e}_z$ and $W_{\mathsf{KQ}}(z) = \boldsymbol{e}_z^{\top}\boldsymbol{W}_{\mathsf{KQ}}\boldsymbol{e}_{\texttt{<EOS>}}$ following Nichani et al. [2024b]. We assume that both $\boldsymbol{W}_{\mathsf{OV}}$ and $\boldsymbol{W}_{\mathsf{KQ}}$ are trainable and show that the non-factorized model fails to generalize to test implications (Theorem 3) by analyzing the gradient flow trajectory.

**Assumption 2.** *Let $\alpha > 0$. We initialize the weights by setting $W_{\mathsf{OV}}(a, z) = \alpha$ and $W_{\mathsf{KQ}}(z) = \alpha\sqrt{|\mathcal{A}| + 1}$ for all $a \in \mathcal{A}$, $z \in \mathcal{V}$.*

**Theorem 3.** *Suppose that $|\mathcal{A}_2| > 1$ and Assumption 2 holds, and we use $\mathcal{L}_{\mathsf{test}}(\tilde{\boldsymbol{\theta}}_t)$ in (5) to denote the test loss for the non-factorized model. For any $t \geq 0$, it holds that*

$$\mathcal{L}_{\mathsf{test}}(\tilde{\boldsymbol{\theta}}_t) = \mathbb{E}_{z_{1:T+1} \sim \mathcal{D}_{\mathsf{test}}}[-\log p_{\tilde{\boldsymbol{\theta}}_t}(z_{T+1}|z_{1:T})] \geq \log |\mathcal{A}_2| > 0.$$

The proof exploits parameter symmetry. Subjects can be partitioned based on $a^\star(s, r)$: $\mathcal{S}_{\mathsf{train}} = \cup_{i=1}^{n}\mathcal{S}_{i,\mathsf{train}}$ and $\mathcal{S}_{\mathsf{test}} = \cup_{i=1}^{n}\mathcal{S}_{i,\mathsf{test}}$, where partition $i$ corresponds to fact $b_i$ and implication $c_i$. Since all partitions are equal-sized, any two pairs $(b_i, c_i)$ and $(b_j, c_j)$ with $i \neq j$ are interchangeable. Applying this symmetry to the optimization dynamics, we show that $W_{\mathsf{OV}}(a, s) = W_{\mathsf{OV}}(a', s)$ for any $a, a' \in \mathcal{A}_2$. Consequently, on the test set, the non-factorized model assigns uniform probability across all answers in the implication set: $p_{\tilde{\boldsymbol{\theta}}_t}(a|s, r_2) = p_{\tilde{\boldsymbol{\theta}}_t}(a'|s, r_2)$ for any $a, a' \in \mathcal{A}_2$. A complete proof is provided in Appendix B.

This result is consistent with the observation of Zhu et al. [2024], which shows that a reparameterized non-factorized one-layer attention-only model struggles to generalize unless the expected answer token follows the important token in the prompt in the training set. Extending this result to factorized models with trainable $\boldsymbol{W}_{\mathsf{KQ}}$ matrices introduces significant complexity due to higher-order interaction terms between parameters. We leave this comprehensive analysis for future work.

# 5 Conclusions

In this work, we study LLMs' generalization and hallucination when fine-tuned with new factual knowledge in a unified way and show that the above two behaviors are both due to the model's OCR ability. We carefully analyze a one-layer linear attention model and prove that the implicit bias of GD on the factorized model enables the model to obtain strong OCR abilities. Our theory establishes that LLMs can easily associate facts and implications based on co-occurrence, and thus can hallucinate when the co-occurrence does not reflect causality. As for future directions, it would be interesting to extend our theoretical analysis to multi-layer transformers, as well as effective methods to prevent this type of hallucination when injecting new factual knowledge into a model.

# Acknowledgements

This work was partially supported by a gift from Open Philanthropy to the Center for Human-Compatible AI (CHAI) at UC Berkeley and by NSF Grants IIS-1901252 and CCF-2211209. This work was also supported by NSF grants DMS-2210827, CCF-2315725, CAREER DMS-2339904, ONR grant N00014-24-S-B001, DARPA AIQ grant HR001124S0029-AIQ-FP-003, an Amazon Research Award, a Google Research Scholar Award, an Okawa Foundation Research Grant, and a Sloan Research Fellowship. Y.H. and S.S. were supported by the U.S. Army Research Laboratory and the U.S. Army Research Office under Grant W911NF2010219, Office of Naval Research, and NSF. This work used Jetstream2 at Indiana University through allocation CIS240832 from the Advanced Cyberinfrastructure Coordination Ecosystem: Services & Support (ACCESS) program, which is supported by National Science Foundation grants #2138259, #2138286, #2138307, #2137603, and #2138296. H.Z. would like to thank Kaifeng Lyu for the helpful discussion on the convergence property of the problem studied.

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

# A  Proof of Section 4.1

In this section, we provide proof for all theoretical results presented in Section 4.1. Specifically, we provide the proof of Proposition 1 in Appendix A.1, Theorem 1 in Appendix A.2, and Theorem 2 in Appendix A.3.

## A.1  Proof of Proposition 1

*Proof of Proposition 1.* We show that for any fixed $\boldsymbol{\theta} = (\boldsymbol{W}_\mathsf{O}, \boldsymbol{W}_\mathsf{V})$, there is a matrix $\tilde{\boldsymbol{\theta}} = \boldsymbol{W}_\mathsf{OV}$ that gives the same test loss as defined in (5) and the same training loss as defined in (4) and vice versa. Following (3), for any input $z_{1:T}$ and answer token $a$, we define the logit functions given by the two sets of parameters as:

$$f_{\boldsymbol{\theta}}(z_{1:T}, a) = \boldsymbol{e}_a^\top \boldsymbol{W}_\mathsf{O} \boldsymbol{W}_\mathsf{V}^\top \boldsymbol{X}^\top \boldsymbol{X} \boldsymbol{W}_\mathsf{KQ} \boldsymbol{x}_T, \quad f_{\tilde{\boldsymbol{\theta}}}(z_{1:T}, a) = \boldsymbol{e}_a^\top \boldsymbol{W}_\mathsf{OV} \boldsymbol{X}^\top \boldsymbol{X} \boldsymbol{W}_\mathsf{KQ} \boldsymbol{x}_T.$$

It suffices to prove that the two logit functions always give the same value for any $(z_{1:T}, a)$. Given any fixed $\boldsymbol{\theta} = (\boldsymbol{W}_\mathsf{O}, \boldsymbol{W}_\mathsf{V})$, we can set $\boldsymbol{W}_\mathsf{OV} := \boldsymbol{W}_\mathsf{O} \boldsymbol{W}_\mathsf{V}^\top$ and get

$$f_{\boldsymbol{\theta}}(z_{1:T}, a) = \boldsymbol{e}_a^\top \boldsymbol{W}_\mathsf{O} \boldsymbol{W}_\mathsf{V}^\top \boldsymbol{X}^\top \boldsymbol{X} \boldsymbol{W}_\mathsf{KQ} \boldsymbol{x}_T = f_{\tilde{\boldsymbol{\theta}}}(z_{1:T}, a).$$

For the other direction, given any $\boldsymbol{W}_\mathsf{OV} \in \mathbb{R}^{d \times d}$, suppose its SVD decomposition is given by $\boldsymbol{W}_\mathsf{OV} = \boldsymbol{U} \boldsymbol{\Sigma} \boldsymbol{V}^\top$ where $\boldsymbol{\Sigma} = \mathrm{diag}(\sigma_1, \ldots, \sigma_r) \in \mathbb{R}^{r \times r}$ with $r \leq d$ and $\sigma_i > 0$ for $i \in [r]$. Since the rank of $\boldsymbol{W}_\mathsf{OV}$ is at most $d$ and $d_h \geq d \geq r$, let $\boldsymbol{\Sigma}^{1/2} = \mathrm{diag}(\sqrt{\sigma_1}, \ldots, \sqrt{\sigma_r})$ and $\boldsymbol{Q} = [\boldsymbol{I}_r \ \boldsymbol{0}_{r \times (d_h - r)}] \in \mathbb{R}^{r \times d_h}$. Note that $\boldsymbol{Q} \boldsymbol{Q}^\top = \boldsymbol{I}_r$. Then we can set

$$\boldsymbol{W}_\mathsf{O} := \boldsymbol{U} \boldsymbol{\Sigma}^{1/2} \boldsymbol{Q}, \boldsymbol{W}_\mathsf{V} := \boldsymbol{V} \boldsymbol{\Sigma}^{1/2} \boldsymbol{Q},$$

such that $\boldsymbol{W}_\mathsf{O} \boldsymbol{W}_\mathsf{V}^\top = \boldsymbol{U} \boldsymbol{\Sigma} \boldsymbol{V}^\top = \boldsymbol{W}_\mathsf{OV}$. Combining both directions, we can conclude that the two parameterizations have equivalent expressive power. However, in the following analysis, we show that there is a key distinction between the two in terms of optimization dynamics. $\square$

## A.2  Proof of Theorem 1

*Proof of Theorem 1.*

1. For the factorized model $\boldsymbol{\theta} = (\boldsymbol{W}_\mathsf{V}, \boldsymbol{W}_\mathsf{O})$, it is a two-layer fully-connected linear network trained by cross-entropy loss. By Theorem 4.4 and Appendix G of Lyu and Li [2019], every limit point of $\left\{ \frac{\boldsymbol{\theta}(t)}{\|\boldsymbol{\theta}(t)\|}, t \geq 0 \right\}$ by gradient descent with small enough learning rates or gradient flow is along the direction of a KKT point of the following program:

$$\min_{\boldsymbol{W}_\mathsf{O}, \boldsymbol{W}_\mathsf{V}} \frac{1}{2} (\|\boldsymbol{W}_\mathsf{O}\|_F^2 + \|\boldsymbol{W}_\mathsf{V}\|_F^2) \tag{OV-SVM}$$
$$\text{s.t. } h_{(s,r),a'}(\boldsymbol{W}_\mathsf{O} \boldsymbol{W}_\mathsf{V}^\top) \geq 1, \forall (s, r) \in \mathcal{D}_\mathsf{train}, \ \forall a' \in \mathcal{A} \backslash \{a^*(s, r)\}.$$

   Moreover, Lemma 1 shows that the above program has the same solutions for $\boldsymbol{W}_\mathsf{OV}^\mathsf{F} = \boldsymbol{W}_\mathsf{O} \boldsymbol{W}_\mathsf{V}^\top$ as ($\boldsymbol{W}_\mathsf{OV}^\mathsf{F}$-SVM).

2. For the non-factorized model $\boldsymbol{W}_\mathsf{OV}$, it is a linear model trained by cross-entropy loss. Again, by Theorem 4.4 and Appendix G of Lyu and Li [2019], every limit point of $\left\{ \frac{\boldsymbol{W}_\mathsf{OV}(t)}{\|\boldsymbol{W}_\mathsf{OV}(t)\|_F}, t \geq 0 \right\}$ by gradient descent with small enough learning rates or gradient flow is along the direction of a KKT point of ($\boldsymbol{W}_\mathsf{OV}$-SVM). Since ($\boldsymbol{W}_\mathsf{OV}$-SVM) is a convex program, the KKT point is sufficient to ensure global optimality.

$\square$

**Remark 2.** *The results in Theorem 1 can be further strengthened under certain conjectures that extend previous results for binary classification to multi-class settings.*

*For the non-factorized model, if our dataset satisfies Equation (15) in Theorem 7 in Soudry et al. [2018], it can be shown that the parameter $\boldsymbol{W}_\mathsf{OV}$ under gradient flow or gradient descent with a*

*small enough step size directionally converges. Equation (15) is proved to be true in Soudry et al. [2018] for the binary setting for almost all datasets, and conjectured to be true in multi-class settings for almost all datasets. Therefore, combining the result of Theorem 1 for the non-factorized model, if the above conjecture is true for our dataset, gradient flow or gradient descent with small enough step sizes directionally converges to the direction of the global minimum of ($W_{\mathsf{OV}}$-SVM).*

*For the factorized model, Theorem 3.1 of Vardi et al. [2022] shows that gradient flow directionally converges to the direction of the global minimum of ($W_{\mathsf{OV}}^{\mathsf{F}}$-SVM) for binary classification. We conjecture this is also true for a multi-class setting under certain mild assumptions, and we leave the proof for future work.*

The proof of Theorem 1 concludes with the following lemma, which establishes the equivalence between the solutions of (OV-SVM) and ($W_{\mathsf{OV}}^{\mathsf{F}}$-SVM).

**Lemma 1.** *Let $W^*$ be an optimal solution of ($W_{\mathsf{OV}}^{\mathsf{F}}$-SVM) with $\mathrm{rank}(W^*) = r$. Assume $d_h \geq r$. The optimization problem (OV-SVM) is equivalent to ($W_{\mathsf{OV}}^{\mathsf{F}}$-SVM). As a result, if $(W_{\mathsf{O}}, W_{\mathsf{V}})$ is a solution of (OV-SVM), then its combined form $W = W_{\mathsf{O}} W_{\mathsf{V}}^{\top}$ is also a global minimum of ($W_{\mathsf{OV}}^{\mathsf{F}}$-SVM).*

*Proof.* We adopt a similar argument as in [Recht et al., 2010]. Consider any optimal solution $(W_{\mathsf{O}}, W_{\mathsf{V}})$ in (OV-SVM) and let $W := W_{\mathsf{O}} W_{\mathsf{V}}^{\top} \in \mathbb{R}^{d \times d}$, for any $(s, r) \in \mathcal{D}_{\mathsf{train}}$, $a' \in \mathcal{A} \setminus \{a^*(s, r)\}$, we have

$$h_{(s,r),a'}(W_{\mathsf{O}} W_{\mathsf{V}}^{\top}) = h_{(s,r),a'}(W) \geq 1.$$

Thus, $W$ is inside the feasible set of ($W_{\mathsf{OV}}^{\mathsf{F}}$-SVM). Moreover, note that the nuclear norm is the dual norm of the spectral norm, which gives

$$
\begin{aligned}
\|W\|_\star &= \sup_{\|Z\|_2 \leq 1} \mathrm{trace}(Z^\top W_{\mathsf{O}} W_{\mathsf{V}}^\top) \\
&= \sup_{\|Z\|_2 \leq 1} \langle Z W_{\mathsf{V}}, W_{\mathsf{O}} \rangle \\
&\leq \sup_{\|Z\|_2 \leq 1} \|Z W_{\mathsf{V}}\|_F \|W_{\mathsf{O}}\|_F \\
&\overset{(a)}{\leq} \frac{1}{2}(\|W_{\mathsf{O}}\|_F^2 + \|W_{\mathsf{V}}\|_F^2),
\end{aligned}
\tag{9}
$$

where (a) follows $\|AB\|_F \leq \|A\|_2 \|B\|_F$ and AM-GM inequality. Now assuming $W^*$ is a optimal solution of ($W_{\mathsf{OV}}^{\mathsf{F}}$-SVM) and $\|W\|_\star > \|W^*\|_\star$. Let its SVD decomposition be $W^* = U \Sigma V^\top$ with $\Sigma = \mathrm{diag}(\sigma_1, \dots, \sigma_r) \in \mathbb{R}^{r \times r}$. We can construct $W_{\mathsf{O}}^* = U \Sigma^{1/2}$ and $W_{\mathsf{V}}^* = V \Sigma^{1/2}$ such that

$$\frac{1}{2}(\|W_{\mathsf{O}}^*\|_F^2 + \|W_{\mathsf{V}}^*\|_F^2) = \|\Sigma^{1/2}\|_F^2 = \mathrm{trace}(\Sigma) = \|W^*\|_\star < \|W\|_\star \overset{(a)}{\leq} \frac{1}{2}(\|W_{\mathsf{O}}\|_F^2 + \|W_{\mathsf{V}}\|_F^2),$$

where (a) follows (9). Note that here we assumed $d_h = r$. If $d_h > r$, we can always choose

$$\tilde{\Sigma}^{1/2} := \left[ \Sigma^{1/2}, \mathbf{0}_{r \times (d_h - r)} \right] \in \mathbb{R}^{r \times d_h}, W_{\mathsf{O}}^* = U \tilde{\Sigma}^{1/2}, W_{\mathsf{V}}^* = V \tilde{\Sigma}^{1/2},$$

which yields the same result. Moreover, $(W_{\mathsf{O}}^*, W_{\mathsf{V}}^*)$ is a solution of (OV-SVM) as $W^*$ is a feasible solution of ($W_{\mathsf{OV}}^{\mathsf{F}}$-SVM). This leads to a contradiction since $(W_{\mathsf{O}}, W_{\mathsf{V}})$ is an optimal solution of (OV-SVM). Conversely, we prove that if $W^* = U \Sigma V^\top$ is an optimal solution of ($W_{\mathsf{OV}}^{\mathsf{F}}$-SVM), $(W_{\mathsf{O}}^*, W_{\mathsf{V}}^*) := (U \Sigma^{1/2}, V \Sigma^{1/2})$ is also an optimal solution of (OV-SVM). Assume it's not optimal and thus there exists a feasible solution $(W_{\mathsf{O}}, W_{\mathsf{V}})$ such that

$$\frac{1}{2}(\|W_{\mathsf{O}}\|_F^2 + \|W_{\mathsf{V}}\|_F^2) < \frac{1}{2}(\|W_{\mathsf{O}}^*\|_F^2 + \|W_{\mathsf{V}}^*\|_F^2).$$

Using the same argument we have

$$\|W^*\|_\star = \frac{1}{2}(\|W_{\mathsf{O}}^*\|_F^2 + \|W_{\mathsf{V}}^*\|_F^2) > \frac{1}{2}(\|W_{\mathsf{O}}\|_F^2 + \|W_{\mathsf{V}}\|_F^2) \geq \|W\|_\star,$$

which again leads to a contradiction. Combining both directions, we conclude that the two problems are equivalent. Eventually, if $(W_{\mathsf{O}}, W_{\mathsf{V}})$ is a global minimum of (OV-SVM), the combined parameter $W = W_{\mathsf{O}} W_{\mathsf{V}}^\top$ is also a global minimum of ($W_{\mathsf{OV}}^{\mathsf{F}}$-SVM). This finishes the proof of Theorem 1. $\quad\square$

### A.3 Proof of Theorem 2

**Useful notations.** We introduce useful notations used in this section. We use $\boldsymbol{I}_n$ to represent an $n \times n$ identity matrix, use $\boldsymbol{E}_n$ to represent an $n \times n$ all-one matrix, and use $\boldsymbol{1}_n$ and $\boldsymbol{0}_n$ to represent $n$-dimensional all-one and all-zero vectors, respectively. We use $\boldsymbol{e}_i = [0, \ldots, 1, \ldots, 0]^\top$ as the one-hot vector in $\mathbb{R}^n$ where the $i$-th entry is one. For convenience, we use $x \wedge y$ to denote the minimum value among $x$ and $y$ and use $x \vee y$ to denote the maximum value among them.

#### A.3.1 Proof for factorized model

Note that although $\boldsymbol{W}_{\mathsf{OV}}^{\mathsf{F}}$ is a $d \times d$ matrix, since we restrict the next token prediction to be among $2n$ answer tokens in $\mathcal{A}$, and only $(nm + 2)$ tokens in $\mathcal{S} \cup \mathcal{R}$ can take effect in the prompt, we only need to consider a reduced matrix $\boldsymbol{W}_{\mathsf{OV}}^{\mathsf{F}} \in \mathbb{R}^{(2n) \times (nm+2)}$ throughout this section, where each row corresponds to a token in $\mathcal{A}$ and each column corresponds to a token in $\mathcal{S} \cup \mathcal{R}$. Now we restate the first part of Theorem 2 below.

**Theorem 4** (Part 1 in Theorem 2: Factorized model has OCR ability). *Let $n > 1$. Suppose $\boldsymbol{W}_{\mathsf{OV}}^{\mathsf{F}}$ is a solution to the SVM problem in Eq. ($\boldsymbol{W}_{\mathsf{OV}}^{\mathsf{F}}$-SVM), then for any $(s, r) \in \mathcal{D}_{\mathsf{test}}$, $a' \in \mathcal{A} \setminus \{a^\star(s, r)\}$, given regularity conditions (Assumption 3), it holds that*

$$h_{(s,r),a'}(\boldsymbol{W}_{\mathsf{OV}}^{\mathsf{F}}) \geq \sqrt{\frac{m_{\mathsf{train}}}{m_{\mathsf{test}}}} \wedge 1, \text{ indicating the OCR ability.} \tag{10}$$

To prove Theorem 4, we derive an explicit solution characterization for ($\boldsymbol{W}_{\mathsf{OV}}^{\mathsf{F}}$-SVM). The proof roadmap is as follows.

1. **Restricted Form Existence**: Lemmas 2 and 3 show the existence of a solution in block structure (11) via permutation averaging and the convexity of nuclear norm.

2. **SVD Computation**: Lemma 4 computes the closed form for the SVD decomposition of the restricted form in Lemma 3.

3. **Nuclear Norm Formula of the restricted form**: Given the restricted form, Lemma 5 gives $\|\boldsymbol{W}_{\mathsf{OV}}^{\mathsf{F}}\|_\star$ in closed form (20).

4. **Optimization**: Lemma 6 finds the minimum of Equation (20) by decomposing $\|\boldsymbol{W}_{\mathsf{OV}}^{\mathsf{F}}\|_\star = M_1 + M_2$.

5. **Solution characterization and the uniqueness**: Theorem 5 uses Lemmas 7 and 8 and Assumption 3 to establish the unique forms of the solution in Equations (30) and (31).

**Lemma 2** (Unitary invariance). *Given matrix $\boldsymbol{A}$, for any orthonormal matrices $\boldsymbol{U}$ and $\boldsymbol{V}$, we have that*

$$\|\boldsymbol{U}\boldsymbol{A}\boldsymbol{V}^\top\|_\star = \|\boldsymbol{A}\|_\star.$$

*Proof.* See Lemma 2.5 in Hoheisel and Paquette [2023]. □

**Lemma 3** (Existence of a restricted form solution to ($\boldsymbol{W}_{\mathsf{OV}}^{\mathsf{F}}$-SVM)). *Suppose $\boldsymbol{W}_{\mathsf{OV}}^{\mathsf{F}}$ is the solution to the optimization problem ($\boldsymbol{W}_{\mathsf{OV}}^{\mathsf{F}}$-SVM). There exists a solution with $p_1$, $p_2$, $q_1$, $q_2$, $f_1$, $f_2$, $g_1$, $g_2$, $\beta_1$, $\beta_2$ and $\gamma_1$, $\gamma_2$ such that*

$$\boldsymbol{W}_{\mathsf{OV}}^{\mathsf{F}} = \begin{bmatrix} \overbrace{p_1\boldsymbol{I}_n + p_2\boldsymbol{E}_n \cdots p_1\boldsymbol{I}_n + p_2\boldsymbol{E}_n}^{m_{\mathsf{train}} \text{ blocks}} & \overbrace{f_1\boldsymbol{I}_n + f_2\boldsymbol{E}_n \cdots f_1\boldsymbol{I}_n + f_2\boldsymbol{E}_n}^{m_{\mathsf{test}} \text{ blocks}} & \beta_1\boldsymbol{1}_n & \beta_2\boldsymbol{1}_n \\ \underbrace{q_1\boldsymbol{I}_n + q_2\boldsymbol{E}_n \cdots q_1\boldsymbol{I}_n + q_2\boldsymbol{E}_n}_{m_{\mathsf{train}} \text{ blocks}} & \underbrace{g_1\boldsymbol{I}_n + g_2\boldsymbol{E}_n \cdots g_1\boldsymbol{I}_n + g_2\boldsymbol{E}_n}_{m_{\mathsf{test}} \text{ blocks}} & \gamma_1\boldsymbol{1}_n & \gamma_2\boldsymbol{1}_n \end{bmatrix}. \tag{11}$$

*Moreover,*

$$\begin{aligned} p_1, f_1, q_1 &\geq 1, \\ p_1 + p_2 + \beta_1 &\geq q_1 + q_2 + \gamma_1 + 1, \\ q_1 + q_2 + \gamma_2 &\geq p_1 + p_2 + \beta_2 + 1, \\ f_1 + f_2 + \beta_1 &\geq (g_1 \vee 0) + g_2 + \gamma_1 + 1. \end{aligned} \tag{12}$$

*Proof of Lemma 3.* We first show that certain permutations of $\boldsymbol{W}_{\mathsf{OV}}^{\mathsf{F}}$ are still solutions of the optimization problem. Suppose that $\sigma$ is any permutation of $\{1, \dots, n\}$. Define $\boldsymbol{P}_\sigma \in \mathbb{R}^{n \times n}$ as the corresponding permutation matrix. Consider the permuted weight matrix

$$\sigma(\boldsymbol{W}_{\mathsf{OV}}^{\mathsf{F}}) = \begin{bmatrix} \boldsymbol{P}_\sigma & 0 \\ 0 & \boldsymbol{P}_\sigma \end{bmatrix} \boldsymbol{W}_{\mathsf{OV}}^{\mathsf{F}} \mathrm{diag}\{\boldsymbol{P}_\sigma, \dots \boldsymbol{P}_\sigma, 1, 1\}.$$

It is equivalent to permuting the subject sets and the fact labels $b$ and $c$ simultaneously with $\sigma$: $\{\mathcal{S}_{\sigma(1)}, \dots, \mathcal{S}_{\sigma(n)}\}$, $\{b_{\sigma(1)}, \dots, b_{\sigma(n)}\}$ and $\{c_{\sigma(1)}, \dots, c_{\sigma(n)}\}$. Using Equation (6), we have that

$$h_{(s_{\sigma(i)}, r_1), b_{\sigma(j)}}(\sigma(\boldsymbol{W}_{\mathsf{OV}}^{\mathsf{F}})) = h_{(s_i, r_1), b_j}(\boldsymbol{W}_{\mathsf{OV}}^{\mathsf{F}}) \geq 1, \quad \forall j \in [n] \setminus \{i\},$$
$$h_{(s_{\sigma(i)}, r_1), c_{\sigma(j)}}(\sigma(\boldsymbol{W}_{\mathsf{OV}}^{\mathsf{F}})) = h_{(s_i, r_1), c_j}(\boldsymbol{W}_{\mathsf{OV}}^{\mathsf{F}}) \geq 1, \quad \forall j \in [n],$$
$$h_{(s_{\sigma(i)}, r_2), b_{\sigma(j)}}(\sigma(\boldsymbol{W}_{\mathsf{OV}}^{\mathsf{F}})) = h_{(s_i, r_2), b_j}(\boldsymbol{W}_{\mathsf{OV}}^{\mathsf{F}}) \geq 1, \quad \forall j \in [n],$$
$$h_{(s_{\sigma(i)}, r_2), c_{\sigma(j)}}(\sigma(\boldsymbol{W}_{\mathsf{OV}}^{\mathsf{F}})) = h_{(s_i, r_2), c_j}(\boldsymbol{W}_{\mathsf{OV}}^{\mathsf{F}}) \geq 1, \quad \forall j \in [n] \setminus \{i\},$$

for any $i$. From Lemma 2, since permutation matrices are orthonormal, $\|\sigma(\boldsymbol{W}_{\mathsf{OV}}^{\mathsf{F}})\|_\star = \|\boldsymbol{W}_{\mathsf{OV}}^{\mathsf{F}}\|_\star$. Therefore, $\sigma(\boldsymbol{W}_{\mathsf{OV}}^{\mathsf{F}})$ is also a solution to the optimization problem ($\boldsymbol{W}_{\mathsf{OV}}^{\mathsf{F}}$-SVM). Let's consider the average over all possible permutations

$$\frac{\sum_\sigma \sigma(\boldsymbol{W}_{\mathsf{OV}}^{\mathsf{F}})}{n!}$$

$$= \begin{bmatrix} \overbrace{p_{11}\boldsymbol{I}_n + p_{21}\boldsymbol{E}_n \cdots p_{1m_{\mathsf{train}}}\boldsymbol{I}_n + p_{2m_{\mathsf{train}}}\boldsymbol{E}_n}^{m_{\mathsf{train}} \text{ blocks}} & \overbrace{f_{11}\boldsymbol{I}_n + f_{21}\boldsymbol{E}_n \cdots f_{1m_{\mathsf{test}}}\boldsymbol{I}_n + f_{2m_{\mathsf{test}}}\boldsymbol{E}_n}^{m_{\mathsf{test}} \text{ blocks}} & \beta_1\boldsymbol{1}_n & \beta_2\boldsymbol{1}_n \\ \underbrace{q_{11}\boldsymbol{I}_n + q_{21}\boldsymbol{E}_n \cdots q_{1m_{\mathsf{train}}}\boldsymbol{I}_n + q_{2m_{\mathsf{train}}}\boldsymbol{E}_n}_{m_{\mathsf{train}} \text{ blocks}} & \underbrace{g_{11}\boldsymbol{I}_n + g_{21}\boldsymbol{E}_n \cdots g_{1m_{\mathsf{test}}}\boldsymbol{I}_n + g_{2m_{\mathsf{test}}}\boldsymbol{E}_n}_{m_{\mathsf{test}} \text{ blocks}} & \gamma_1\boldsymbol{1}_n & \gamma_2\boldsymbol{1}_n \end{bmatrix}.$$

It is also a solution to the optimization problem ($\boldsymbol{W}_{\mathsf{OV}}^{\mathsf{F}}$-SVM) due to the convexity of the nuclear norm.

We can consider other permutations. Suppose that $\tau_{\mathsf{train}}$ is a permutation of the index set $\{1, \dots, m_{\mathsf{train}}\}$. Define the permuted weight matrix

$$\tau_{\mathsf{train}}(\boldsymbol{W}_{\mathsf{OV}}^{\mathsf{F}}) = \boldsymbol{W}_{\mathsf{OV}}^{\mathsf{F}} \mathrm{diag}\{\boldsymbol{P}_{\tau_{\mathsf{train}}} \otimes \boldsymbol{I}_n, \underbrace{1, \dots, 1}_{nm_{\mathsf{test}}+2}\}.$$

It is equivalent to permute all the subjects in the set $\mathcal{S}_{i,\mathsf{train}}$: $\{s_{i,\tau_{\mathsf{train}}(1)}, \dots, s_{i,\tau_{\mathsf{train}}(m_{\mathsf{train}})}\}$ for any $i = 1, \dots, n$. Note that there is no need to permute the labels, as subjects in $\mathcal{S}_{i,\mathsf{train}}$ share the same label pair $b_i$ and $c_i$. Since the permuted $\mathcal{S}_{i,\mathsf{train}}$ is still disjoint with the test subject set $\mathcal{S}_{i,\mathsf{test}}$, we have that for any $i \in [n], j \in [m_{\mathsf{train}}]$, it holds that

$$h_{(s_{i,\tau_{\mathsf{train}}(j)}, r_1), a'}(\tau_{\mathsf{train}}(\boldsymbol{W}_{\mathsf{OV}}^{\mathsf{F}})) = h_{(s_{i,j}, r_1), a'}(\boldsymbol{W}_{\mathsf{OV}}^{\mathsf{F}}) \geq 1, \quad \forall a' \in \mathcal{A} \setminus \{b_i\},$$
$$h_{(s_{i,\tau_{\mathsf{train}}(j)}, r_2), a'}(\tau_{\mathsf{train}}(\boldsymbol{W}_{\mathsf{OV}}^{\mathsf{F}})) = h_{(s_{i,j}, r_2), a'}(\boldsymbol{W}_{\mathsf{OV}}^{\mathsf{F}}) \geq 1, \quad \forall a' \in \mathcal{A} \setminus \{c_i\}.$$

From Lemma 2, we can conclude that $\|\tau_{\mathsf{train}}(\boldsymbol{W}_{\mathsf{OV}}^{\mathsf{F}})\|_\star = \|\boldsymbol{W}_{\mathsf{OV}}^{\mathsf{F}}\|_\star$. Averaging over all permutations $\sigma$ and $\tau_{m_{\mathsf{train}}}$, we have that

$$\frac{\sum_{\tau_{\mathsf{train}}} \sum_\sigma \tau_{\mathsf{train}}(\sigma(\boldsymbol{W}_{\mathsf{OV}}^{\mathsf{F}}))}{m_{\mathsf{train}}! n!}$$

$$= \begin{bmatrix} \overbrace{p_1\boldsymbol{I}_n + p_2\boldsymbol{E}_n \cdots p_1\boldsymbol{I}_n + p_2\boldsymbol{E}_n}^{m_{\mathsf{train}} \text{ blocks}} & \overbrace{f_{11}\boldsymbol{I}_n + f_{21}\boldsymbol{E}_n \cdots f_{1m_{\mathsf{test}}}\boldsymbol{I}_n + f_{2m_{\mathsf{test}}}\boldsymbol{E}_n}^{m_{\mathsf{test}} \text{ blocks}} & \beta_1\boldsymbol{1}_n & \beta_2\boldsymbol{1}_n \\ \underbrace{q_1\boldsymbol{I}_n + q_2\boldsymbol{E}_n \cdots q_1\boldsymbol{I}_n + q_2\boldsymbol{E}_n}_{m_{\mathsf{train}} \text{ blocks}} & \underbrace{g_{11}\boldsymbol{I}_n + g_{21}\boldsymbol{E}_n \cdots g_{1m_{\mathsf{test}}}\boldsymbol{I}_n + g_{2m_{\mathsf{test}}}\boldsymbol{E}_n}_{m_{\mathsf{test}} \text{ blocks}} & \gamma_1\boldsymbol{1}_n & \gamma_2\boldsymbol{1}_n \end{bmatrix}.$$

We can then consider the permutation over the remaining $m_{\mathsf{test}}$ indices $\{m_{\mathsf{train}}+1, \dots, m\}$. Let $\tau_{\mathsf{test}}$ denote the permutation. Consider the permuted weight matrix.

$$\tau_{\mathsf{test}}(\boldsymbol{W}_{\mathsf{OV}}^{\mathsf{F}}) = \boldsymbol{W}_{\mathsf{OV}}^{\mathsf{F}} \mathrm{diag}\{\underbrace{1, \dots, 1}_{nm_{\mathsf{train}}}, \boldsymbol{P}_{\tau_{\mathsf{test}}} \otimes \boldsymbol{I}_n, 1, 1\}.$$

It is equivalent to permute all subjects in the set $\mathcal{S}_{i,\mathsf{test}}$ : $\{s_{i,\tau_{\mathsf{test}}(m_{\mathsf{train}}+1)}, \dots, s_{i,\tau_{\mathsf{test}}(m)}\}$ for any $i = 1, \dots, n$. We have that for any $i \in [n], j \in [m] \setminus [m_{\mathsf{train}}]$, it holds that

$$h_{(s_{i,\tau_{\mathsf{test}}(j)}, r_1), a'}(\tau_{\mathsf{test}}(\boldsymbol{W}_{\mathsf{OV}}^{\mathsf{F}})) = h_{(s_{i,j}, r_1), a'}(\boldsymbol{W}_{\mathsf{OV}}^{\mathsf{F}}) \geq 1, \quad \forall a' \in \mathcal{A} \setminus \{b_i\}.$$

Taking the average weight over all possible permutations $\tau_{\text{test}}$,

$$\frac{\sum_{\tau_{\text{test}}} \sum_{\tau_{\text{train}}} \sum_{\sigma} \tau_{\text{test}}(\tau_{\text{train}}(\sigma(\boldsymbol{W}_{\text{OV}}^{\text{F}})))}{m_{\text{test}}! m_{\text{train}}! n!}$$

$$= \begin{bmatrix} \overbrace{p_1 \boldsymbol{I}_n + p_2 \boldsymbol{E}_n \cdots p_1 \boldsymbol{I}_n + p_2 \boldsymbol{E}_n}^{m_{\text{train}} \text{ blocks}} & \overbrace{f_1 \boldsymbol{I}_n + f_2 \boldsymbol{E}_n \cdots f_1 \boldsymbol{I}_n + f_2 \boldsymbol{E}_n}^{m_{\text{test}} \text{ blocks}} & \beta_1 \boldsymbol{1}_n & \beta_2 \boldsymbol{1}_n \\ \underbrace{q_1 \boldsymbol{I}_n + q_2 \boldsymbol{E}_n \cdots q_1 \boldsymbol{I}_n + q_2 \boldsymbol{E}_n}_{m_{\text{train}} \text{ blocks}} & \underbrace{g_1 \boldsymbol{I}_n + g_2 \boldsymbol{E}_n \cdots g_1 \boldsymbol{I}_n + g_2 \boldsymbol{E}_n}_{m_{\text{test}} \text{ blocks}} & \gamma_1 \boldsymbol{1}_n & \gamma_2 \boldsymbol{1}_n \end{bmatrix}.$$

This shows that Equation (11) is one solution to the optimization problem ($\boldsymbol{W}_{\text{OV}}^{\text{F}}$-SVM). This proves Lemma 3. □

We could compute the closed form of the SVD decomposition for Equation (11).

**Lemma 4.** *The restricted form in Equation (11) has a the SVD decomposition $\boldsymbol{W}_{\text{OV}}^{\text{F}} = \boldsymbol{U}\boldsymbol{\Sigma}\boldsymbol{V}^{\top}$ with*

$$\boldsymbol{U} = \left[ \boldsymbol{u}^{(1)}, \boldsymbol{u}^{(2)}, \boldsymbol{u}_2^{(1)}, \ldots, \boldsymbol{u}_n^{(1)}, \boldsymbol{u}_2^{(2)}, \ldots, \boldsymbol{u}_n^{(2)} \right],$$

$$\boldsymbol{\Sigma} = diag\left\{ \sigma_1^{(1)}, \sigma_1^{(2)}, \underbrace{\sigma_2^{(1)}, \ldots, \sigma_2^{(1)}}_{n-1}, \underbrace{\sigma_2^{(2)}, \ldots, \sigma_2^{(2)}}_{n-1} \right\},$$

$$\boldsymbol{V} = \left[ \boldsymbol{v}^{(1)}, \boldsymbol{v}^{(2)}, \boldsymbol{v}_2^{(1)}, \ldots, \boldsymbol{v}_n^{(1)}, \boldsymbol{v}_2^{(2)}, \ldots, \boldsymbol{v}_n^{(2)} \right],$$

*where $\boldsymbol{u}^{(k)}$ are defined in Equation (14); $\boldsymbol{u}_j^{(k')}$ are defined in Equation (16), $\sigma_1^{(k)}$ and $\sigma_2^{(k')}$ are defined in Equation (17); $\boldsymbol{v}^{(k)}$ are defined in Equation (18); $\boldsymbol{v}_j^{(k')}$ are defined in Equation (19).*

*The proof of Lemma 4.* The dimensions of $\boldsymbol{W}_{\text{OV}}^{\text{F}}$ are $2n \times N_c$, where $N_c = (m_{\text{train}} + m_{\text{test}})n + 2$. Denote the Singular Value Decomposition (SVD) of $\boldsymbol{W}_{\text{OV}}^{\text{F}}$ by $\boldsymbol{W}_{\text{OV}}^{\text{F}} = \boldsymbol{U}\boldsymbol{\Sigma}\boldsymbol{V}^{\top}$.

**Orthonormal basis and matrix properties.** Given an orthonormal basis $\{\boldsymbol{\eta}_1, \ldots, \boldsymbol{\eta}_n\}$ for $\mathbb{R}^n$, with $\boldsymbol{\eta}_1 = \boldsymbol{1}_n/\sqrt{n}$, where $\boldsymbol{1}_n$ is the $n$-dim all-one column vector. The vectors $\boldsymbol{\eta}_2, \ldots, \boldsymbol{\eta}_n$ are orthonormal to each other and to $\boldsymbol{\eta}_1$. Let $\boldsymbol{X} = c_1 \boldsymbol{I}_n + c_2 \boldsymbol{E}_n$ be a block appearing in the matrix $\boldsymbol{W}_{\text{OV}}^{\text{F}}$. Its action on the basis vectors is:

- $\boldsymbol{X}\boldsymbol{\eta}_1 = (c_1 \boldsymbol{I}_n + c_2 \boldsymbol{E}_n)(\frac{1}{\sqrt{n}}\boldsymbol{1}_n) = c_1 \frac{1}{\sqrt{n}}\boldsymbol{1}_n + c_2 \frac{1}{\sqrt{n}}(\boldsymbol{1}_n \boldsymbol{1}_n^{\top})\boldsymbol{1}_n = c_1 \boldsymbol{\eta}_1 + c_2 \frac{n}{\sqrt{n}}\boldsymbol{1}_n = (c_1 + nc_2)\boldsymbol{\eta}_1$.

- For $j \in \{2, \ldots, n\}$, $\boldsymbol{X}\boldsymbol{\eta}_j = (c_1 \boldsymbol{I}_n + c_2 \boldsymbol{E}_n)\boldsymbol{\eta}_j = c_1 \boldsymbol{\eta}_j$, since $\boldsymbol{E}_n \boldsymbol{\eta}_j = \boldsymbol{1}_n(\boldsymbol{1}_n^{\top} \boldsymbol{\eta}_j) = \boldsymbol{0}_n$ due to $\boldsymbol{\eta}_j \perp \boldsymbol{1}_n$.

**Left singular vectors $\boldsymbol{U}$ and singular values $\boldsymbol{\Sigma}$.** The left singular vectors (columns of $\boldsymbol{U}$) are eigenvectors of $\boldsymbol{W}_{\text{OV}}^{\text{F}}\boldsymbol{W}_{\text{OV}}^{\text{F}\top}$. The matrix $\boldsymbol{W}_{\text{OV}}^{\text{F}}\boldsymbol{W}_{\text{OV}}^{\text{F}\top}$ is a $2n \times 2n$ symmetric matrix. After block multiplication, $\boldsymbol{W}_{\text{OV}}^{\text{F}}\boldsymbol{W}_{\text{OV}}^{\text{F}\top}$ can be written as:

$$\boldsymbol{W}_{\text{OV}}^{\text{F}}\boldsymbol{W}_{\text{OV}}^{\text{F}\top} = \begin{pmatrix} C_{A1}\boldsymbol{I}_n + C_{A2}\boldsymbol{E}_n & C_{B1}\boldsymbol{I}_n + C_{B2}\boldsymbol{E}_n \\ C_{B1}\boldsymbol{I}_n + C_{B2}\boldsymbol{E}_n & C_{D1}\boldsymbol{I}_n + C_{D2}\boldsymbol{E}_n \end{pmatrix},$$

where the coefficients are:

$C_{A1} = m_{\text{train}}p_1^2 + m_{\text{test}}f_1^2$,

$C_{A2} = m_{\text{train}}(2p_1p_2 + np_2^2) + m_{\text{test}}(2f_1f_2 + nf_2^2) + \beta_1^2 + \beta_2^2$,

$C_{D1} = m_{\text{train}}q_1^2 + m_{\text{test}}g_1^2$,

$C_{D2} = m_{\text{train}}(2q_1q_2 + nq_2^2) + m_{\text{test}}(2g_1g_2 + ng_2^2) + \gamma_1^2 + \gamma_2^2$,

$C_{B1} = m_{\text{train}}p_1q_1 + m_{\text{test}}f_1g_1$,

$C_{B2} = m_{\text{train}}(p_1q_2 + p_2q_1 + np_2q_2) + m_{\text{test}}(f_1g_2 + f_2g_1 + nf_2g_2) + \beta_1\gamma_1 + \beta_2\gamma_2$.

We seek eigenvectors of $\boldsymbol{W}_{\text{OV}}^{\text{F}}\boldsymbol{W}_{\text{OV}}^{\text{F}\top}$ of the form

$$\boldsymbol{u} = \begin{pmatrix} x_b \boldsymbol{\eta}_j \\ x_c \boldsymbol{\eta}_j \end{pmatrix}$$

for scalars $x_b, x_c$.

**Case 1: Eigenvectors associated with $\boldsymbol{\eta}_1$.** For $j = 1$, the eigenvalue problem $\boldsymbol{W}_{\mathsf{OV}}^{\mathsf{F}}\boldsymbol{W}_{\mathsf{OV}}^{\mathsf{F}\top}\boldsymbol{u} = \lambda\boldsymbol{u}$ transforms into a $2 \times 2$ eigenvalue problem for the coefficient vector $(x_b, x_c)^\top$:

$$\boldsymbol{H}_1 \begin{pmatrix} x_b \\ x_c \end{pmatrix} = \lambda \begin{pmatrix} x_b \\ x_c \end{pmatrix},$$

where

$$\boldsymbol{H}_1 = \begin{pmatrix} C_{A1} + nC_{A2} & C_{B1} + nC_{B2} \\ C_{B1} + nC_{B2} & C_{D1} + nC_{D2} \end{pmatrix}. \tag{13}$$

Let $\lambda_1^{(1)}, \lambda_1^{(2)}$ be the eigenvalues of $\boldsymbol{H}_1$, and let $(x_{1,b}, x_{1,c})^\top$ and $(x_{2,b}, x_{2,c})^\top$ be the corresponding normalized eigenvectors. These give two singular values: $\sigma_1^{(k)} = \sqrt{\lambda_1^{(k)}}$ for $k = 1, 2$. The corresponding left singular vectors in $\boldsymbol{U}$ are

$$\boldsymbol{u}^{(k)} = \begin{pmatrix} x_{k,b}\boldsymbol{\eta}_1 \\ x_{k,c}\boldsymbol{\eta}_1 \end{pmatrix} \text{ for } k = 1, 2. \tag{14}$$

**Case 2: Eigenvectors associated with $\boldsymbol{\eta}_j$ for $j \in \{2, \dots, n\}$.** For $j \in \{2, \dots, n\}$, the eigenvalue problem $\boldsymbol{W}_{\mathsf{OV}}^{\mathsf{F}}\boldsymbol{W}_{\mathsf{OV}}^{\mathsf{F}\top}\boldsymbol{u} = \lambda\boldsymbol{u}$ reduces to:

$$\boldsymbol{H}_2 \begin{pmatrix} x_b \\ x_c \end{pmatrix} = \lambda \begin{pmatrix} x_b \\ x_c \end{pmatrix}$$

where

$$\boldsymbol{H}_2 = \begin{pmatrix} C_{A1} & C_{B1} \\ C_{B1} & C_{D1} \end{pmatrix}. \tag{15}$$

Let $\lambda_2^{(1)}, \lambda_2^{(2)}$ be the eigenvalues of $\boldsymbol{H}_2$, and let $(y_{1,b}, y_{1,c})^\top$ and $(y_{2,b}, y_{2,c})^\top$ be the corresponding normalized eigenvectors. These give $2(n-1)$ singular values: $\sigma_2^{(k')} = \sqrt{\lambda_2^{(k')}}$ for $k' = 1, 2$. Each of these singular values has a multiplicity of $(n-1)$. The corresponding left singular vectors in $\boldsymbol{U}$ are

$$\boldsymbol{u}_j^{(k')} = \begin{pmatrix} y_{k',b}\boldsymbol{\eta}_j \\ y_{k',c}\boldsymbol{\eta}_j \end{pmatrix} \text{ for each } j \in \{2, \dots, n\} \text{ and } k' = 1, 2. \tag{16}$$

The matrix $\boldsymbol{U}$ is a $2n \times 2n$ orthogonal matrix whose columns are the $2n$ left singular vectors $\boldsymbol{u}^{(k)}$ and $\boldsymbol{u}_j^{(k')}$.

**The singular matrix.** From the calculation of $\boldsymbol{U}$, we get that

$$\boldsymbol{\Sigma} = \mathrm{diag}\Big\{ \sigma_1^{(1)}, \sigma_1^{(2)}, \underbrace{\sigma_2^{(1)}, \dots, \sigma_2^{(1)}}_{n-1}, \underbrace{\sigma_2^{(2)}, \dots, \sigma_2^{(2)}}_{n-1} \Big\}, \tag{17}$$

where $\sigma_1^{(k)} = \sqrt{\lambda_1^{(k)}}$ for $k = 1, 2$, with $\lambda_1^{(k)}$ being the eigenvalues of the matrix $\boldsymbol{H}_1$ in Equation (13); $\sigma_2^{(k')} = \sqrt{\lambda_2^{(k')}}$ for $k' = 1, 2$, with $\lambda_2^{(k')}$ being the eigenvalues of the matrix $\boldsymbol{H}_2$ in Equation (15).

**Right singular vectors $\boldsymbol{V}$.** The columns of $\boldsymbol{V}$ (right singular vectors) are obtained from $\boldsymbol{v}_i = \sigma_i^{-1}\boldsymbol{W}_{\mathsf{OV}}^{\mathsf{F}\top}\boldsymbol{u}_i$ for non-zero $\sigma_i$. If $\sigma_i = 0$, $\boldsymbol{v}_i$ is a normalized vector in the null space of $\boldsymbol{W}_{\mathsf{OV}}^{\mathsf{F}}$. The matrix $\boldsymbol{V}$ has dimensions $N_c \times 2n$. The $2n$ vectors $\boldsymbol{v}_i$ corresponding to the found singular values are:

**Case 1: Right singular vectors associated with $\boldsymbol{u}^{(k)}$, $k = 1, 2$.** For $\boldsymbol{u}^{(k)} = \begin{pmatrix} x_{k,b}\boldsymbol{\eta}_1 \\ x_{k,c}\boldsymbol{\eta}_1 \end{pmatrix}$ and singular value $\sigma_1^{(k)}$: The vector $\boldsymbol{W}_{\mathsf{OV}}^{\mathsf{F}\top}\boldsymbol{u}^{(k)}$ has the following structure:

- The first $m_{\mathsf{train}}$ blocks (each of size $n$) are $(x_{k,b}(p_1 + np_2) + x_{k,c}(q_1 + nq_2))\boldsymbol{\eta}_1$.
- The next $m_{\mathsf{test}}$ blocks (each of size $n$) are $(x_{k,b}(f_1 + nf_2) + x_{k,c}(g_1 + ng_2))\boldsymbol{\eta}_1$.

- The last two components are scalar values: $\sqrt{n}(\beta_1 x_{k,b} + \gamma_1 x_{k,c})$ and $\sqrt{n}(\beta_2 x_{k,b} + \gamma_2 x_{k,c})$.

So, $\boldsymbol{v}^{(k)} = \boldsymbol{W}_{\mathsf{OV}}^{\mathsf{FT}} \boldsymbol{u}^{(k)} / \sigma_1^{(k)}$ is:

$$
\boldsymbol{v}^{(k)} = \frac{1}{\sigma_1^{(k)}}
\begin{pmatrix}
(x_{k,b}(p_1 + np_2) + x_{k,c}(q_1 + nq_2))\boldsymbol{\eta}_1 \\
\vdots \\
(x_{k,b}(p_1 + np_2) + x_{k,c}(q_1 + nq_2))\boldsymbol{\eta}_1 \\
(x_{k,b}(f_1 + nf_2) + x_{k,c}(g_1 + ng_2))\boldsymbol{\eta}_1 \\
\vdots \\
(x_{k,b}(f_1 + nf_2) + x_{k,c}(g_1 + ng_2))\boldsymbol{\eta}_1 \\
\sqrt{n}(\beta_1 x_{k,b} + \gamma_1 x_{k,c}) \\
\sqrt{n}(\beta_2 x_{k,b} + \gamma_2 x_{k,c})
\end{pmatrix}
\quad \text{for } k = 1, 2.
\tag{18}
$$

Each $\boldsymbol{\eta}_1$ term represents a column vector of $n$ elements. This vector $\boldsymbol{v}^{(k)}$ has total length $N_c = m_{\mathsf{train}} n + m_{\mathsf{test}} n + 2$.

**Case 2: Right singular vectors associated with $\boldsymbol{u}_j^{(k')}$.** For $\boldsymbol{u}_j^{(k')} = \begin{pmatrix} y_{k',b}\boldsymbol{\eta}_j \\ y_{k',c}\boldsymbol{\eta}_j \end{pmatrix}$ (where $j \geq 2$ and $k' = 1, 2$) and singular value $\sigma_2^{(k')}$: The vector $\boldsymbol{W}_{\mathsf{OV}}^{\mathsf{FT}} \boldsymbol{u}_j^{(k')}$ has the following structure:

- The first $m_{\mathsf{train}}$ blocks (each of size $n$) are $(y_{k',b}p_1 + y_{k',c}q_1)\boldsymbol{\eta}_j$.

- The next $m_{\mathsf{test}}$ blocks (each of size $n$) are $(y_{k',b}f_1 + y_{k',c}g_1)\boldsymbol{\eta}_j$.

- The last two scalar components are 0, because $\mathbf{1}_n^{\top}\boldsymbol{\eta}_j = 0$ for $j \geq 2$.

So, $\boldsymbol{v}_j^{(k')} = \boldsymbol{W}_{\mathsf{OV}}^{\mathsf{FT}} \boldsymbol{u}_j^{(k')} / \sigma_2^{(k')}$ is:

$$
\boldsymbol{v}_j^{(k')} = \frac{1}{\sigma_2^{(k')}}
\begin{pmatrix}
(y_{k',b}p_1 + y_{k',c}q_1)\boldsymbol{\eta}_j \\
\vdots \\
(y_{k',b}p_1 + y_{k',c}q_1)\boldsymbol{\eta}_j \\
(y_{k',b}f_1 + y_{k',c}g_1)\boldsymbol{\eta}_j \\
\vdots \\
(y_{k',b}f_1 + y_{k',c}g_1)\boldsymbol{\eta}_j \\
0 \\
0
\end{pmatrix}.
\tag{19}
$$

There are $2(n - 1)$ $\boldsymbol{v}_j^{(k')}$ vectors, one for each pair $(j, k')$ where $j \in \{2, \ldots, n\}$ and $k' \in \{1, 2\}$. Each $\boldsymbol{\eta}_j$ term represents a column vector of $n$ elements. The vectors $\boldsymbol{v}^{(k)}$ and $\boldsymbol{v}_j^{(k')}$ are orthonormal and form the columns of $\boldsymbol{V}$.

As a conclusion, the SVD of $\boldsymbol{W}_{\mathsf{OV}}^{\mathsf{F}}$ is $\boldsymbol{W}_{\mathsf{OV}}^{\mathsf{F}} = \boldsymbol{U}\boldsymbol{\Sigma}\boldsymbol{V}^{\top}$, where:

- $\boldsymbol{U}$ is a $2n \times 2n$ orthogonal matrix with columns $\boldsymbol{u}^{(k)}$ and $\boldsymbol{u}_j^{(k')}$ as defined in Equations (14) and (16).

- $\boldsymbol{\Sigma}$ is a $2n \times 2n$ rectangular diagonal matrix, whose non-zero entries are the singular values $\sigma_1^{(k)}$ and $\sigma_2^{(k')}$, derived from the eigenvalues of $\boldsymbol{H}_1$ and $\boldsymbol{H}_2$.

- $\boldsymbol{V}$ is an $N_c \times 2n$ matrix with columns $\boldsymbol{v}^{(k)}$ and $\boldsymbol{v}_j^{(k')}$ as defined in Equations (18) and (19).

This finishes the proof of Lemma 4. $\qquad\square$

**Lemma 5.** *The $\boldsymbol{W}_{\mathsf{OV}}^{\mathsf{F}}$ in restricted form Equation* (11) *has the nuclear norm* $\|\boldsymbol{W}_{\mathsf{OV}}^{\mathsf{F}}\|_{\star}$:

$$\sqrt{\frac{(C_{A1} + nC_{A2} + C_{D1} + nC_{D2}) + \sqrt{(C_{A1} + nC_{A2} - (C_{D1} + nC_{D2}))^2 + 4(C_{B1} + nC_{B2})^2}}{2}}$$

$$+ \sqrt{\frac{(C_{A1} + nC_{A2} + C_{D1} + nC_{D2}) - \sqrt{(C_{A1} + nC_{A2} - (C_{D1} + nC_{D2}))^2 + 4(C_{B1} + nC_{B2})^2}}{2}}$$

$$+ (n-1)\left(\sqrt{\frac{(C_{A1} + C_{D1}) + \sqrt{(C_{A1} - C_{D1})^2 + 4C_{B1}^2}}{2}} + \sqrt{\frac{(C_{A1} + C_{D1}) - \sqrt{(C_{A1} - C_{D1})^2 + 4C_{B1}^2}}{2}}\right).$$

$$(20)$$

*Proof of Lemma 5.* The nuclear norm of a matrix $\boldsymbol{W}_{\text{OV}}^{\text{F}}$, denoted as $\|\boldsymbol{W}_{\text{OV}}^{\text{F}}\|_\star$, is defined as the sum of its singular values. From Lemma 4 and its proof, the singular values of $\boldsymbol{W}_{\text{OV}}^{\text{F}}$ are derived from the eigenvalues of two $2 \times 2$ matrices, $\boldsymbol{H}_1$ and $\boldsymbol{H}_2$.

The singular values are:

1. $\sigma_1^{(1)}$ and $\sigma_1^{(2)}$, which are the square roots of the two eigenvalues of $\boldsymbol{H}_1$ defined in Equation (13). These singular values each have a multiplicity of 1.

2. $\sigma_2^{(1)}$ and $\sigma_2^{(2)}$, which are the square roots of the two eigenvalues of $\boldsymbol{H}_2$ defined in Equation (15). As stated in the proof of Lemma 4 (Case 2: Eigenvectors associated with $\boldsymbol{\eta}_j$ for $j \in \{2, \ldots, n\}$), these singular values correspond to the $n-1$ basis vectors $\boldsymbol{\eta}_j$ for $j \in \{2, \ldots, n\}$. Thus, each of $\sigma_2^{(1)}$ and $\sigma_2^{(2)}$ has a multiplicity of $(n-1)$.

The nuclear norm is therefore the sum of all these singular values:

$$\|\boldsymbol{W}_{\text{OV}}^{\text{F}}\|_\star = \sigma_1^{(1)} + \sigma_1^{(2)} + (n-1)\sigma_2^{(1)} + (n-1)\sigma_2^{(2)}$$
$$= (\sigma_1^{(1)} + \sigma_1^{(2)}) + (n-1)(\sigma_2^{(1)} + \sigma_2^{(2)}).$$

Let's expand each term:

**Term 1: Sum of singular values from $\boldsymbol{H}_1$.**

The matrix $\boldsymbol{H}_1$ is given by Equation (13):

$$\boldsymbol{H}_1 = \begin{pmatrix} C_{A1} + nC_{A2} & C_{B1} + nC_{B2} \\ C_{B1} + nC_{B2} & C_{D1} + nC_{D2} \end{pmatrix}.$$

Let $a_1 = C_{A1} + nC_{A2}$, $b_1 = C_{B1} + nC_{B2}$, and $c_1 = C_{D1} + nC_{D2}$. The eigenvalues $\lambda_1^{(1)}, \lambda_1^{(2)}$ of this symmetric $2 \times 2$ matrix $\boldsymbol{H}_1 = \begin{pmatrix} a_1 & b_1 \\ b_1 & c_1 \end{pmatrix}$ are given by the formula

$$\lambda = \frac{(a_1 + c_1) \pm \sqrt{(a_1 - c_1)^2 + 4b_1^2}}{2}.$$

The singular values $\sigma_1^{(1)}$ and $\sigma_1^{(2)}$ are $\sqrt{\lambda_1^{(1)}}$ and $\sqrt{\lambda_1^{(2)}}$. We get that

$$\sigma_1^{(1)} + \sigma_1^{(2)}$$
$$= \sqrt{\frac{(a_1 + c_1) + \sqrt{(a_1 - c_1)^2 + 4b_1^2}}{2}} + \sqrt{\frac{(a_1 + c_1) - \sqrt{(a_1 - c_1)^2 + 4b_1^2}}{2}}$$
$$= \sqrt{\frac{(C_{A1} + nC_{A2} + C_{D1} + nC_{D2}) + \sqrt{(C_{A1} + nC_{A2} - (C_{D1} + nC_{D2}))^2 + 4(C_{B1} + nC_{B2})^2}}{2}}$$
$$+ \sqrt{\frac{(C_{A1} + nC_{A2} + C_{D1} + nC_{D2}) - \sqrt{(C_{A1} + nC_{A2} - (C_{D1} + nC_{D2}))^2 + 4(C_{B1} + nC_{B2})^2}}{2}}.$$

This corresponds to the first two lines of the expression for the nuclear norm in Lemma 5.

**Term 2: Sum of singular values from $H_2$.**

The matrix $H_2$ is given by Equation (15):

$$H_2 = \begin{pmatrix} C_{A1} & C_{B1} \\ C_{B1} & C_{D1} \end{pmatrix}.$$

Let $a_2 = C_{A1}$, $b_2 = C_{B1}$, and $c_2 = C_{D1}$. The eigenvalues $\lambda_2^{(1)}, \lambda_2^{(2)}$ of this symmetric $2 \times 2$ matrix $H_2 = \begin{pmatrix} a_2 & b_2 \\ b_2 & c_2 \end{pmatrix}$ are:

$$\lambda = \frac{(a_2 + c_2) \pm \sqrt{(a_2 - c_2)^2 + 4b_2^2}}{2}.$$

The singular values $\sigma_2^{(1)}$ and $\sigma_2^{(2)}$ are $\sqrt{\lambda_2^{(1)}}$ and $\sqrt{\lambda_2^{(2)}}$. Thus:

$$\sigma_2^{(1)} + \sigma_2^{(2)}$$

$$= \sqrt{\frac{(a_2 + c_2) + \sqrt{(a_2 - c_2)^2 + 4b_2^2}}{2}} + \sqrt{\frac{(a_2 + c_2) - \sqrt{(a_2 - c_2)^2 + 4b_2^2}}{2}}$$

$$= \sqrt{\frac{(C_{A1} + C_{D1}) + \sqrt{(C_{A1} - C_{D1})^2 + 4C_{B1}^2}}{2}} + \sqrt{\frac{(C_{A1} + C_{D1}) - \sqrt{(C_{A1} - C_{D1})^2 + 4C_{B1}^2}}{2}}.$$

This sum is then multiplied by the multiplicity $(n-1)$:

$$(n-1)(\sigma_2^{(1)} + \sigma_2^{(2)}) = (n-1)\left( \sqrt{\frac{(C_{A1} + C_{D1}) + \sqrt{(C_{A1} - C_{D1})^2 + 4C_{B1}^2}}{2}} \right.$$

$$\left. + \sqrt{\frac{(C_{A1} + C_{D1}) - \sqrt{(C_{A1} - C_{D1})^2 + 4C_{B1}^2}}{2}} \right).$$

This corresponds to the third and fourth lines of the expression for the nuclear norm in Lemma 5.

Combining these terms, the nuclear norm $\|W_{\mathsf{OV}}^{\mathsf{F}}\|_\star$ is precisely the expression given in Lemma 5. This completes the proof. $\qquad\square$

**Lemma 6.** *Let the expression $\|W_{\mathsf{OV}}^{\mathsf{F}}\|_\star$ be defined as Equation (20). The closed-form minimum of $\|W_{\mathsf{OV}}^{\mathsf{F}}\|_\star$ is given by*

$$\min \|W_{\mathsf{OV}}^{\mathsf{F}}\|_\star = \sqrt{n} + (n-1) \times \begin{cases} (\sqrt{m_{\mathsf{train}}} + \sqrt{m_{\mathsf{test}}}) & \text{if } m_{\mathsf{test}} \geq m_{\mathsf{train}}, \\ \sqrt{2(m_{\mathsf{train}} + m_{\mathsf{test}})} & \text{if } m_{\mathsf{test}} < m_{\mathsf{train}} \end{cases},$$

*where the minimum is achieved at $p_1^\star = f_1^\star = q_1^\star = 1$, $g_1^\star = \sqrt{m_{\mathsf{train}}/m_{\mathsf{test}}}$, $p_2^\star = f_2^\star = q_2^\star = -1/n$, $g_2^\star = -\sqrt{m_{\mathsf{train}}/m_{\mathsf{test}}}/n$, $\beta_1^\star = \gamma_2^\star = 1/2, \gamma_1^\star = \beta_2^\star = -1/2$ if $m_{\mathsf{test}} \geq m_{\mathsf{train}}$; $p_1^\star = f_1^\star = q_1^\star = g_1^\star = 1, p_2^\star = f_2^\star = q_2^\star = g_2^\star = -1/n, \beta_1^\star = \gamma_2^\star = 1/2, \gamma_1^\star = \beta_2^\star = -1/2$ if $m_{\mathsf{test}} < m_{\mathsf{train}}$.*

*Proof of Lemma 6.* The overall expression $\|W_{\mathsf{OV}}^{\mathsf{F}}\|_\star$ given in Equation (20) is a sum of two main components. Let $M_1$ be the sum of the terms in the first two lines, and $M_2$ be the sum of the terms in the last line. Thus, $\|W_{\mathsf{OV}}^{\mathsf{F}}\|_\star = M_1 + M_2$. Both $M_1$ and $M_2$ represent sums of square roots of eigenvalues of effective $2 \times 2$ matrices.

**Part 1.** We first consider minimizing the last two terms of $\|W_{\mathsf{OV}}^{\mathsf{F}}\|_\star$. Define $M_2$ as

$$(n-1)\left( \sqrt{\frac{(C_{A1} + C_{D1}) + \sqrt{(C_{A1} - C_{D1})^2 + 4C_{B1}^2}}{2}} + \sqrt{\frac{(C_{A1} + C_{D1}) - \sqrt{(C_{A1} - C_{D1})^2 + 4C_{B1}^2}}{2}} \right). \tag{21}$$

For ease of reference, we state the formulas for coefficients $C_{A1}, C_{D1}, C_{B1}$:

$$C_{A1} = m_{\mathsf{train}}p_1^2 + m_{\mathsf{test}}f_1^2, \tag{22}$$

$$C_{D1} = m_{\mathsf{train}}q_1^2 + m_{\mathsf{test}}g_1^2, \tag{23}$$

$$C_{B1} = m_{\mathsf{train}}p_1q_1 + m_{\mathsf{test}}f_1g_1. \tag{24}$$

The expression for $M_2$ can be simplified. Let $\lambda_2^{(1)}$ and $\lambda_2^{(2)}$ be the two eigenvalues of the matrix $\boldsymbol{H}_2$. We note that $\lambda_2^{(1)} + \lambda_2^{(2)} = C_{A1} + C_{D1}$ and $\lambda_2^{(1)}\lambda_2^{(2)} = C_{A1}C_{D1} - C_{B1}^2$. The term $C_{A1}C_{D1} - C_{B1}^2$ can be calculated as

$$C_{A1}C_{D1} - C_{B1}^2 = (m_{\text{train}}p_1^2 + m_{\text{test}}f_1^2)(m_{\text{train}}q_1^2 + m_{\text{test}}g_1^2) - (m_{\text{train}}p_1q_1 + m_{\text{test}}f_1g_1)^2$$
$$= m_{\text{train}}m_{\text{test}}(p_1^2g_1^2 - 2p_1g_1f_1q_1 + f_1^2q_1^2) = m_{\text{train}}m_{\text{test}}(p_1g_1 - f_1q_1)^2.$$

Since $m_{\text{train}}, m_{\text{test}} > 0$, we have $C_{A1}C_{D1} - C_{B1}^2 \geq 0$, so the eigenvalues $\lambda_1, \lambda_2$ are non-negative. The sum $\sqrt{\lambda_2^{(1)}} + \sqrt{\lambda_2^{(2)}}$ can be written as $\sqrt{\left(\sqrt{\lambda_2^{(1)}} + \sqrt{\lambda_2^{(2)}}\right)^2} = \sqrt{\lambda_2^{(1)} + \lambda_2^{(2)} + 2\sqrt{\lambda_2^{(1)}\lambda_2^{(2)}}}$.
Substituting the trace and determinant yields

$$\sqrt{\lambda_2^{(1)}} + \sqrt{\lambda_2^{(2)}} = \sqrt{C_{A1} + C_{D1} + 2\sqrt{m_{\text{train}}m_{\text{test}}(p_1g_1 - f_1q_1)^2}}$$
$$= \sqrt{C_{A1} + C_{D1} + 2\sqrt{m_{\text{train}}m_{\text{test}}}|p_1g_1 - f_1q_1|}.$$

Let $S = C_{A1} + C_{D1} + 2\sqrt{m_{\text{train}}m_{\text{test}}}|p_1g_1 - f_1q_1|$. Substituting the definitions of $C_{A1}$ and $C_{D1}$:

$$S = m_{\text{train}}p_1^2 + m_{\text{test}}f_1^2 + m_{\text{train}}q_1^2 + m_{\text{test}}g_1^2 + 2\sqrt{m_{\text{train}}m_{\text{test}}}|p_1g_1 - f_1q_1|. \tag{25}$$

Then $M_2 = (n-1)\sqrt{S}$. To minimize $\|W\|_\star$, we must minimize $S$ with respect to $p_1, q_1, f_1, g_1$ under the given constraints. Let $K = p_1g_1 - f_1q_1$. We analyze the minimization by considering two cases for $K$.

**Case 1:** $K \leq 0$ (**i.e.,** $p_1g_1 \leq f_1q_1$). Then

$$S = S_2 = (\sqrt{m_{\text{train}}}p_1 - \sqrt{m_{\text{test}}}g_1)^2 + (\sqrt{m_{\text{test}}}f_1 + \sqrt{m_{\text{train}}}q_1)^2.$$

The condition $K \leq 0$ implies that $g_1 \leq f_1q_1/p_1$. The first term $(\sqrt{m_{\text{train}}}p_1 - \sqrt{m_{\text{test}}}g_1)^2$ is monotonically decreasing when $0 \leq g_1 \leq p_1\sqrt{m_{\text{train}}/m_{\text{test}}}$, with the minimum attained at $g_1 = p_1\sqrt{m_{\text{train}}/m_{\text{test}}}$.

- If $m_{\text{test}} > m_{\text{train}}$, $g_1 = p_1\sqrt{m_{\text{train}}/m_{\text{test}}}$ is always achievable, so by taking $f_1 = q_1 = 1$, we get $S_2 = (\sqrt{m_{\text{train}}} + \sqrt{m_{\text{test}}})^2$. The equality holds for any $p_1 \in [1, (m_{\text{test}}/m_{\text{train}})^{1/4}]$ and $g_1 = p_1\sqrt{m_{\text{train}}/m_{\text{test}}}$.

- If $m_{\text{test}} \leq m_{\text{train}}$, $g_1 = p_1\sqrt{m_{\text{train}}/m_{\text{test}}}$ may not be achievable. But since the first term is monotonically decreasing with respect to $g_1$ when $0 \leq g_1 \leq p_1\sqrt{m_{\text{train}}/m_{\text{test}}}$, we get that the minimum is always taken with $g_1 = \min\{f_1q_1/p_1, p_1\sqrt{m_{\text{train}}/m_{\text{test}}}\}$.

  If $g_1 = f_1q_1/p_1$, we get that

  $$S_2 = (\sqrt{m_{\text{train}}}p_1 - \sqrt{m_{\text{test}}}f_1q_1/p_1)^2 + (\sqrt{m_{\text{test}}}f_1 + \sqrt{m_{\text{train}}}q_1)^2$$
  $$= m_{\text{train}}q_1^2 + m_{\text{test}}f_1^2 + m_{\text{train}}p_1^2 + m_{\text{test}}\frac{f_1^2q_1^2}{p_1^2}$$
  $$\geq 2(m_{\text{train}} + m_{\text{test}}),$$

  where the equality holds if and only if $p_1 = g_1 = f_1 = q_1 = 1$.

  If $g_1 = p_1\sqrt{m_{\text{train}}/m_{\text{test}}}$, we get $f_1q_1 \geq p_1^2\sqrt{m_{\text{train}}/m_{\text{test}}} \geq \sqrt{m_{\text{train}}/m_{\text{test}}}$. Therefore,

  $$S_2 = (\sqrt{m_{\text{test}}}f_1 + \sqrt{m_{\text{train}}}q_1)^2$$
  $$\geq 4\sqrt{m_{\text{test}}m_{\text{train}}}f_1q_1$$
  $$\geq 4m_{\text{train}},$$

  where the equality holds if and only if $p_1 = q_1 = 1$, $g_1 = f_1 = \sqrt{m_{\text{train}}/m_{\text{test}}}$.

  Therefore, we can conclude that if $m_{\text{test}} \leq m_{\text{train}}$, $S_2 \geq 2(m_{\text{train}} + m_{\text{test}})$, with the equality holds if and only if $p_1 = g_1 = f_1 = q_1 = 1$.

**Case 2: Next consider** $K \geq 0$**, that is,** $p_1 g_1 \geq f_1 q_1$**.** Then

$$S = S_1 = (\sqrt{m_{\text{train}}} p_1 + \sqrt{m_{\text{test}}} g_1)^2 + (\sqrt{m_{\text{test}}} f_1 - \sqrt{m_{\text{train}}} q_1)^2.$$

Taking $g_1 \geq f_1 q_1 / p_1$ into $S_1$,

$$S_1 \geq m_{\text{test}} f_1^2 + m_{\text{train}} q_1^2 + m_{\text{train}} p_1^2 + m_{\text{test}} \frac{f_1^2 q_1^2}{p_1^2} \geq m_{\text{test}} + m_{\text{train}} + m_{\text{train}} p_1^2 + \frac{m_{\text{test}}}{p_1^2}.$$

- If $m_{\text{test}} \geq m_{\text{train}}$, we have that $S_1 \geq (\sqrt{m_{\text{test}}} + \sqrt{m_{\text{train}}})^2$, with equality holds when $p_1 = [m_{\text{test}} / m_{\text{train}}]^{1/4}$ and $g_1 = 1/p_1$.

- If $m_{\text{test}} < m_{\text{train}}$, the minimum is achieved by $p_1 = 1$. We therefore get that $g_1 = 1$, with $S_1 \geq 2(m_{\text{test}} + m_{\text{train}})$.

Consolidating the minima, we find that if $m_{\text{test}} \geq m_{\text{train}}$, the minimum $S$ is $(\sqrt{m_{\text{train}}} + \sqrt{m_{\text{test}}})^2$, achieved when $1 \leq p_1 \leq (m_{\text{test}} / m_{\text{train}})^{1/4}$, $f_1 = 1$, $q_1 = 1$, and $g_1 = p_1 \sqrt{m_{\text{train}} / m_{\text{test}}}$. If $m_{\text{test}} < m_{\text{train}}$, the minimum $S$ is $2(m_{\text{train}} + m_{\text{test}})$, achieved when $p_1 = 1, f_1 = 1, q_1 = 1, g_1 = 1$. The minimum value for $\sqrt{S}$ is therefore $\sqrt{m_{\text{train}}} + \sqrt{m_{\text{test}}}$ if $m_{\text{test}} \geq m_{\text{train}}$, and $\sqrt{2(m_{\text{train}} + m_{\text{test}})}$ if $m_{\text{test}} < m_{\text{train}}$. Multiplying by $(n-1)$ gives the final result for $\min M_2$.

**Part 2.** We seek to find the minimum value of the quantity $M_1$ given by

$$M_1 = \sqrt{\frac{(C_{A1} + nC_{A2} + C_{D1} + nC_{D2}) + \sqrt{(C_{A1} + nC_{A2} - (C_{D1} + nC_{D2}))^2 + 4(C_{B1} + nC_{B2})^2}}{2}}$$

$$+ \sqrt{\frac{(C_{A1} + nC_{A2} + C_{D1} + nC_{D2}) - \sqrt{(C_{A1} + nC_{A2} - (C_{D1} + nC_{D2}))^2 + 4(C_{B1} + nC_{B2})^2}}{2}}.$$

where the coefficients are defined as

$$C_{A1} = m_{\text{train}} p_1^2 + m_{\text{test}} f_1^2,$$
$$C_{A2} = m_{\text{train}} (2p_1 p_2 + np_2^2) + m_{\text{test}} (2f_1 f_2 + nf_2^2) + \beta_1^2 + \beta_2^2,$$
$$C_{D1} = m_{\text{train}} q_1^2 + m_{\text{test}} g_1^2,$$
$$C_{D2} = m_{\text{train}} (2q_1 q_2 + nq_2^2) + m_{\text{test}} (2g_1 g_2 + ng_2^2) + \gamma_1^2 + \gamma_2^2,$$
$$C_{B1} = m_{\text{train}} p_1 q_1 + m_{\text{test}} f_1 g_1,$$
$$C_{B2} = m_{\text{train}} (p_1 q_2 + p_2 q_1 + np_2 q_2) + m_{\text{test}} (f_1 g_2 + f_2 g_1 + nf_2 g_2) + \beta_1 \gamma_1 + \beta_2 \gamma_2.$$

subject to the constraints

$$p_1, f_1, q_1 \geq 1, \tag{26}$$
$$p_1 + p_2 + \beta_1 \geq q_1 + q_2 + \gamma_1 + 1, \tag{27}$$
$$q_1 + q_2 + \gamma_2 \geq p_1 + p_2 + \beta_2 + 1, \tag{28}$$
$$f_1 + f_2 + \beta_1 \geq \max(g_1, 0) + g_2 + \gamma_1 + 1. \tag{29}$$

Let $A = C_{A1} + nC_{A2}$, $D = C_{D1} + nC_{D2}$, and $B = C_{B1} + nC_{B2}$. The expression for $M_1$ can be simplified to

$$M_1 = \sqrt{A + D + 2\sqrt{AD - B^2}}.$$

Note that

$$M_1^2 \geq A + D$$
$$= m_{\text{train}} (p_1 + np_2)^2 + m_{\text{train}} (q_1 + nq_2)^2 + m_{\text{test}} (f_1 + nf_2)^2 + m_{\text{test}} (g_1 + ng_2)^2$$
$$+ n(\beta_1^2 + \beta_2^2 + \gamma_1^2 + \gamma_2^2).$$

Let $\Delta = q_1 + q_2 - p_1 - p_2$. From the constraints, we can get that

$$\beta_1 - \gamma_1 \geq 1 + \Delta,$$
$$\gamma_2 - \beta_2 \geq 1 - \Delta.$$

If $|\Delta| > 1$, without loss of generality, assume $\Delta > 1$. This gives that

$$(\beta_1^2 + \gamma_1^2) + (\beta_2^2 + \gamma_2^2) \geq 2\left(\frac{1+\Delta}{2}\right)^2 + 0 \geq 2.$$

If $|\Delta| \leq 1$, we get that

$$(\beta_1^2 + \gamma_1^2) + (\beta_2^2 + \gamma_2^2) \geq 2\left(\frac{1+\Delta}{2}\right)^2 + 2\left(\frac{1-\Delta}{2}\right)^2$$

$$\geq 4\left(\frac{1}{4} + \frac{1}{4}\Delta^2\right)$$

$$\geq 1.$$

As a result, we get $M_1 \geq \sqrt{n}$. Note that the equality holds if and only if

$$p_2 = -p_1/n, \ f_2 = -f_1/n, \ q_2 = -q_1/n, \ g_2 = -g_1/n,$$
$$\beta_1 = 1/2, \ \gamma_1 = -1/2, \ \beta_2 = -1/2, \ \gamma_2 = 1/2.$$

Combining this condition with the results in part 1, We finish the proof for Lemma 6. $\qquad \square$

We would adopt Assumption 3 to prove the uniqueness, which assumes the solution either takes a symmetric form as (11), or it is nondegenerate.

**Assumption 3.** *Let $\boldsymbol{W}_{\mathsf{OV}}^{\mathsf{F}\star}$ be a solution to ($\boldsymbol{W}_{\mathsf{OV}}^{\mathsf{F}}$-SVM). Then it either takes the form of (11), or we have $\boldsymbol{1}_{2n}^\top \boldsymbol{W}_{\mathsf{OV}}^{\mathsf{F}\star} \neq \boldsymbol{0}_{nm+2}^\top$.*

Lemma 7 can give a lower bound of the nuclear norm for a given matrix $\bar{\boldsymbol{M}}$.

**Lemma 7.** *Suppose that $\bar{\boldsymbol{M}} \in \mathbb{R}^{m \times n}$, for any orthonormal matrix $\boldsymbol{U} \in \mathbb{R}^{m \times r}$ and $\boldsymbol{V} \in \mathbb{R}^{n \times r}$, we have that*

$$\|\bar{\boldsymbol{M}}\|_\star \geq \mathrm{Trace}(\boldsymbol{U}^\top \bar{\boldsymbol{M}} \boldsymbol{V}).$$

*Moreover, if there exits $\boldsymbol{\eta} \in \mathbb{R}^m$ s.t. $\boldsymbol{\eta}^\top \boldsymbol{U} = 0$ and $\boldsymbol{\eta}^\top \bar{\boldsymbol{M}} \neq 0$, the above inequality is strict, i.e.,*

$$\|\bar{\boldsymbol{M}}\|_\star > \mathrm{Trace}(\boldsymbol{U}^\top \bar{\boldsymbol{M}} \boldsymbol{V}).$$

*Proof of Lemma 7.* For any pair of matrices $\boldsymbol{A}$ and $\boldsymbol{B}$ such that $\boldsymbol{A}\boldsymbol{B} = \bar{\boldsymbol{M}}$, we have that

$$\mathrm{Trace}(\boldsymbol{U}^\top \bar{\boldsymbol{M}} \boldsymbol{V}) = \mathrm{Trace}(\boldsymbol{U}^\top \boldsymbol{A}\boldsymbol{B} \boldsymbol{V})$$

$$\leq \left(\|\boldsymbol{U}^\top \boldsymbol{A}\|_{\mathrm{F}}^2 + \|\boldsymbol{B}\boldsymbol{V}\|_{\mathrm{F}}^2\right)/2$$

$$\leq \left(\|\boldsymbol{U}^\top\|_2^2\|\boldsymbol{A}\|_{\mathrm{F}}^2 + \|\boldsymbol{V}\|_2^2\|\boldsymbol{B}\|_{\mathrm{F}}^2\right)/2$$

$$\leq \left(\|\boldsymbol{A}\|_{\mathrm{F}}^2 + \|\boldsymbol{B}\|_{\mathrm{F}}^2\right)/2$$

$$\leq \|\bar{\boldsymbol{M}}\|_\star.$$

Consider the conditions for taking the equality. We should require $\|\boldsymbol{U}^\top \boldsymbol{A}\|_{\mathrm{F}} = \|\boldsymbol{U}^\top\|_2\|\boldsymbol{A}\|_{\mathrm{F}}$. This means that all columns of $\boldsymbol{A}$ are left-singular vectors of $\boldsymbol{U}^\top$ with the largest singular value. In addition, it implies that the column space $\mathrm{Col}(\boldsymbol{A}) \subseteq \mathrm{Col}(\boldsymbol{U})$. But if there exits $\boldsymbol{\eta} \in \mathbb{R}^m$ s.t. $\boldsymbol{\eta}^\top \boldsymbol{U} = 0$ and $\boldsymbol{\eta}^\top \bar{\boldsymbol{M}} \neq 0$, we get $\mathrm{Col}(\boldsymbol{A}) \not\subseteq \mathrm{Col}(\boldsymbol{U})$. It implies the conditions for taking the equality cannot be satisfied, so

$$\|\bar{\boldsymbol{M}}\|_\star > \mathrm{Trace}(\boldsymbol{U}^\top \bar{\boldsymbol{M}} \boldsymbol{V}).$$

This finishes Lemma 7. $\qquad \square$

**Theorem 5** (Uniqueness of the solution to the optimization problem ($\boldsymbol{W}_{\mathsf{OV}}^{\mathsf{F}}$-SVM))**.** *Given Assumption 3, suppose that $\boldsymbol{W}_{\mathsf{OV}}^{\mathsf{F}}$ is a solution to optimization problem ($\boldsymbol{W}_{\mathsf{OV}}^{\mathsf{F}}$-SVM), if $m_{\text{test}} < m_{\text{train}}$,*

$$\boldsymbol{W}_{\mathsf{OV}}^{\mathsf{F}} = \left[\underbrace{\begin{matrix}\boldsymbol{I}_n - \boldsymbol{E}_n/n \cdots \boldsymbol{I}_n - \boldsymbol{E}_n/n \\ \boldsymbol{I}_n - \boldsymbol{E}_n/n \cdots \boldsymbol{I}_n - \boldsymbol{E}_n/n\end{matrix}}_{m_{\text{train}}\ blocks} \ \underbrace{\begin{matrix}\boldsymbol{I}_n - \boldsymbol{E}_n/n \cdots \boldsymbol{I}_n - \boldsymbol{E}_n/n \\ \boldsymbol{I}_n - \boldsymbol{E}_n/n \cdots \boldsymbol{I}_n - \boldsymbol{E}_n/n\end{matrix}}_{m_{\text{test}}\ blocks} \ \begin{matrix}\boldsymbol{1}_n/2 & -\boldsymbol{1}_n/2 \\ -\boldsymbol{1}_n/2 & \boldsymbol{1}_n/2\end{matrix}\right]. \quad (30)$$

*If $m_{\text{test}} \geq m_{\text{train}}$, let $\rho = \sqrt{m_{\text{train}}/m_{\text{test}}}$, we have that*

$$\boldsymbol{W}_{\text{OV}}^{\text{F}} = \begin{bmatrix} \overbrace{\boldsymbol{I}_n - \boldsymbol{E}_n/n \cdots \boldsymbol{I}_n - \boldsymbol{E}_n/n}^{m_{\text{train}} \text{ blocks}} & \overbrace{\boldsymbol{I}_n - \boldsymbol{E}_n/n \cdots \boldsymbol{I}_n - \boldsymbol{E}_n/n}^{m_{\text{test}} \text{ blocks}} & \boldsymbol{1}_n/2 & -\boldsymbol{1}_n/2 \\ \underbrace{\boldsymbol{I}_n - \boldsymbol{E}_n/n \cdots \boldsymbol{I}_n - \boldsymbol{E}_n/n}_{m_{\text{train}} \text{ blocks}} & \underbrace{\rho\boldsymbol{I}_n - \rho\boldsymbol{E}_n/n \cdots \rho\boldsymbol{I}_n - \rho\boldsymbol{E}_n/n}_{m_{\text{test}} \text{ blocks}} & -\boldsymbol{1}_n/2 & \boldsymbol{1}_n/2 \end{bmatrix}.$$

$$(31)$$

*Proof of Theorem 5.* Firstly, combining Lemmas 3 and 6, we get Equation (30) and (31). This shows that Equation (30) and (31) are one of the solutions to the optimization problem ($\boldsymbol{W}_{\text{OV}}^{\text{F}}$-SVM).

Suppose that there exists an asymmetric solution of the SVM problem Equation ($\boldsymbol{W}_{\text{OV}}^{\text{F}}$-SVM) $\boldsymbol{W}_{\text{OV}}^{\text{F}\star}$ that takes a different form from (11). Lemma 8 proves the existence of another asymmetric solution such that the difference takes the reversed sign.

**Lemma 8** (The solution with the difference taking the reverse sign). *Suppose that the $\boldsymbol{W}_{\text{OV}}^{\text{F}\star}$ defined above is an asymmetric solution. Then there exists another asymmetric solution $\widetilde{\boldsymbol{W}_{\text{OV}}}$ and $\epsilon \in (0,1)$ such that*

$$\widetilde{\boldsymbol{W}_{\text{OV}}} - \boldsymbol{W}_{\text{OV}}^{\text{F}} = -\epsilon\Big(\boldsymbol{W}_{\text{OV}}^{\text{F}\star} - \boldsymbol{W}_{\text{OV}}^{\text{F}}\Big).$$

*Proof.* Let $\tilde{\varphi}$ be all possible permutations on $\boldsymbol{W}_{\text{OV}}$ as described in Lemma 3, excluding the identity permutation. We get that

$$\frac{\sum_{\tilde{\varphi}} \tilde{\varphi}(\boldsymbol{W}_{\text{OV}}^{\text{F}\star}) + \boldsymbol{W}_{\text{OV}}^{\text{F}\star}}{1 + \sum_{\tilde{\varphi}} 1} = \boldsymbol{W}_{\text{OV}}^{\text{F}}.$$

So we can take

$$\widetilde{\boldsymbol{W}_{\text{OV}}} = \frac{\sum_{\tilde{\varphi}} \tilde{\varphi}(\boldsymbol{W}_{\text{OV}}^{\text{F}\star})}{\sum_{\tilde{\varphi}} 1}$$

and get that

$$\widetilde{\boldsymbol{W}_{\text{OV}}} - \boldsymbol{W}_{\text{OV}}^{\text{F}} = \frac{\boldsymbol{W}_{\text{OV}}^{\text{F}} - \boldsymbol{W}_{\text{OV}}^{\text{F}\star}}{\sum_{\tilde{\varphi}} 1}.$$

This finishes the proof for Lemma 8. $\qquad\square$

Assumption 3 implies that $\boldsymbol{1}_{2n}^{\top}(\boldsymbol{W}_{\text{OV}}^{\text{F}\star} - \boldsymbol{W}_{\text{OV}}^{\text{F}}) \neq \boldsymbol{0}_{nm+2}^{\top}$. We choose $\boldsymbol{U} \in \mathbb{R}^{2n \times r}$ and $\boldsymbol{V} \in \mathbb{R}^{(nm+2) \times r}$ so that they are the SVD decompositions corresponding to $r$ positive singular values for the symmetric solution $\boldsymbol{W}_{\text{OV}}^{\text{F}}$. Since $\boldsymbol{1}_{2n}^{\top}\boldsymbol{W}_{\text{OV}}^{\text{F}} = \boldsymbol{0}_{nm+2}^{\top}$, it also holds that $\boldsymbol{1}_{2n}^{\top}\boldsymbol{U} = \boldsymbol{0}_{nm+2}^{\top}$. Using Lemma 7, we get

$$\begin{aligned} \|\boldsymbol{W}_{\text{OV}}^{\text{F}\star}\|_{\star} &> \text{Trace}(\boldsymbol{U}^{\top}\boldsymbol{W}_{\text{OV}}^{\text{F}\star}\boldsymbol{V}) \\ &= \text{Trace}(\boldsymbol{U}^{\top}\boldsymbol{W}_{\text{OV}}^{\text{F}}\boldsymbol{V}) + \text{Trace}(\boldsymbol{U}^{\top}(\boldsymbol{W}_{\text{OV}}^{\text{F}\star} - \boldsymbol{W}_{\text{OV}}^{\text{F}})\boldsymbol{V}) \\ &= \|\boldsymbol{W}_{\text{OV}}^{\text{F}}\|_{\star} + \text{Trace}(\boldsymbol{U}^{\top}(\boldsymbol{W}_{\text{OV}}^{\text{F}\star} - \boldsymbol{W}_{\text{OV}}^{\text{F}})\boldsymbol{V}). \end{aligned}$$

Using Lemma 8, there exists another asymmetric solution $\widetilde{\boldsymbol{W}_{\text{OV}}}$ such that

$$\begin{aligned} \|\widetilde{\boldsymbol{W}_{\text{OV}}}\|_{\star} &> \text{Trace}(\boldsymbol{U}^{\top}\boldsymbol{W}_{\text{OV}}^{\text{F}}\boldsymbol{V}) + \text{Trace}(\boldsymbol{U}^{\top}(\widetilde{\boldsymbol{W}_{\text{OV}}} - \boldsymbol{W}_{\text{OV}}^{\text{F}})\boldsymbol{V}) \\ &= \|\boldsymbol{W}_{\text{OV}}^{\text{F}}\|_{\star} - \epsilon\text{Trace}(\boldsymbol{U}^{\top}(\boldsymbol{W}_{\text{OV}}^{\text{F}\star} - \boldsymbol{W}_{\text{OV}}^{\text{F}})\boldsymbol{V}). \end{aligned}$$

Therefore, we have that

$$\begin{aligned} &\max\{\|\boldsymbol{W}_{\text{OV}}^{\text{F}\star}\|_{\star}, \|\widetilde{\boldsymbol{W}_{\text{OV}}}\|_{\star}\} \\ &> \|\boldsymbol{W}_{\text{OV}}^{\text{F}}\|_{\star} + \max\{\text{Trace}(\boldsymbol{U}^{\top}(\boldsymbol{W}_{\text{OV}}^{\text{F}\star} - \boldsymbol{W}_{\text{OV}}^{\text{F}})\boldsymbol{V}), -\epsilon\text{Trace}(\boldsymbol{U}^{\top}(\boldsymbol{W}_{\text{OV}}^{\text{F}\star} - \boldsymbol{W}_{\text{OV}}^{\text{F}})\boldsymbol{V})\} \\ &\geq \|\boldsymbol{W}_{\text{OV}}^{\text{F}}\|_{\star}. \end{aligned}$$

This leads to a contradiction that both $\boldsymbol{W}_{\text{OV}}^{\text{F}\star}$ and $\widetilde{\boldsymbol{W}_{\text{OV}}}$ are solutions to Equation ($\boldsymbol{W}_{\text{OV}}^{\text{F}}$-SVM) that minimize the nuclear norm. This confirms that all solutions would take the form Equation (11), which would be Equations (30) and (31).

$\qquad\square$

### A.3.2 Proof for non-factorized model

Similar to Appendix A.3.1, we only need to consider a reduced matrix $\boldsymbol{W}_{\text{OV}} \in \mathbb{R}^{(2n)\times(nm+2)}$ throughout this section, where each row corresponds to a token in $\mathcal{A}$ and each column corresponds to a token in $\mathcal{S} \cup \mathcal{R}$. Now We restate the second part of Theorem 2 below.

**Theorem 6** (Part 2 in Theorem 2: Non-factorized model has no OCR ability). *Let $n > 1$. Suppose $\boldsymbol{W}_{\text{OV}}$ is a solution to the SVM problem in ($\boldsymbol{W}_{\text{OV}}$-SVM). For any $(s, r) \in \mathcal{D}_{\text{test}}$ and any $a' \in \mathcal{A}_2 \setminus \{a^\star(s, r)\}$, it holds that*

$$h_{(s,r),a'}(\boldsymbol{W}_{\text{OV}}) = 0, \text{ indicating no OCR ability.} \tag{32}$$

The proof of Theorem 6 follows the same idea as Theorem 4. The roadmap of the proof is as follows:

1. **A unique solution of a similar form to Lemma 3**: Lemma 9 shows both the existence and uniqueness of the solution to ($\boldsymbol{W}_{\text{OV}}$-SVM).

2. **Frobenius norm formula of the restricted form**: Using the same SVD decomposition as in Lemma 4, Lemma 10 gives $\|\boldsymbol{W}_{\text{OV}}\|_{\text{F}}$ in closed form (35).

3. **Optimization**: Lemma 11 finds the minimum of (35).

4. **Solution characterization**: Theorem 7 gives the form of the solution to ($\boldsymbol{W}_{\text{OV}}$-SVM) and finishes the proof for Theorem 6.

**Lemma 9** (Existence and uniqueness of a restricted form solution to ($\boldsymbol{W}_{\text{OV}}$-SVM)). *Suppose $\boldsymbol{W}_{\text{OV}}$ is the solution to the optimization problem ($\boldsymbol{W}_{\text{OV}}$-SVM). The solution must take the form in Equation (33)*

$$\boldsymbol{W}_{\text{OV}} = \begin{bmatrix} \overbrace{p_1\boldsymbol{I}_n + p_2\boldsymbol{E}_n \cdots p_1\boldsymbol{I}_n + p_2\boldsymbol{E}_n}^{m_{\text{train}} \text{ blocks}} & \overbrace{f_1\boldsymbol{I}_n + f_2\boldsymbol{E}_n \cdots f_1\boldsymbol{I}_n + f_2\boldsymbol{E}_n}^{m_{\text{test}} \text{ blocks}} & \beta_1\mathbf{1}_n & \beta_2\mathbf{1}_n \\ \underbrace{q_1\boldsymbol{I}_n + q_2\boldsymbol{E}_n \cdots q_1\boldsymbol{I}_n + q_2\boldsymbol{E}_n}_{m_{\text{train}} \text{ blocks}} & \underbrace{g_1\boldsymbol{I}_n + g_2\boldsymbol{E}_n \cdots g_1\boldsymbol{I}_n + g_2\boldsymbol{E}_n}_{m_{\text{test}} \text{ blocks}} & \gamma_1\mathbf{1}_n & \gamma_2\mathbf{1}_n \end{bmatrix}. \tag{33}$$

*with $p_1$, $p_2$, $q_1$, $q_2$, $f_1$, $f_2$, and $g_1$, $g_2$ satisfying*

$$\begin{aligned} & p_1, f_1, q_1 \geq 1, \\ & p_1 + p_2 + \beta_1 \geq q_1 + q_2 + \gamma_1 + 1, \\ & q_1 + q_2 + \gamma_2 \geq p_1 + p_2 + \beta_2 + 1, \\ & f_1 + f_2 + \beta_1 \geq (g_1 \vee 0) + g_2 + \gamma_1 + 1. \end{aligned} \tag{34}$$

*Proof of Lemma 9.* The existence proof is the same as the proof for Lemma 3. Since the Frobenius norm is strongly convex, the solution is unique. $\qquad\square$

**Lemma 10.** *The $\boldsymbol{W}_{\text{OV}}$ in restricted form Equation (33) has the Frobenius norm.*

$$\|\boldsymbol{W}_{\text{OV}}\|_{\text{F}} = \left( n \left[ \begin{array}{c} m_{\text{train}}((p_1 + p_2)^2 + (n-1)p_2^2 + (q_1 + q_2)^2 + (n-1)q_2^2) \\ +m_{\text{test}}((f_1 + f_2)^2 + (n-1)f_2^2 + (g_1 + g_2)^2 + (n-1)g_2^2) \\ +\beta_1^2 + \gamma_2^2 + \beta_2^2 + \gamma_1^2 \end{array} \right] \right)^{1/2}. \tag{35}$$

*Proof of Lemma 10.* The square of the Frobenius norm is the sum of the squares of these $2n$ singular values:

$$\|\boldsymbol{W}_{\text{OV}}^{\text{F}}\|_F^2 = (\sigma_1^{(1)})^2 + (\sigma_1^{(2)})^2 + (n-1)(\sigma_2^{(1)})^2 + (n-1)(\sigma_2^{(2)})^2$$

From Lemma 4, we know that $(\sigma_1^{(k)})^2 = \lambda_1^{(k)}$ (eigenvalues of $\boldsymbol{H}_1$) and $(\sigma_2^{(k')})^2 = \lambda_2^{(k')}$ (eigenvalues of $\boldsymbol{H}_2$). So,

$$\|\boldsymbol{W}_{\text{OV}}^{\text{F}}\|_F^2 = \lambda_1^{(1)} + \lambda_1^{(2)} + (n-1)(\lambda_2^{(1)} + \lambda_2^{(2)})$$

Since the sum of eigenvalues of a matrix is its trace, we have $\lambda_1^{(1)} + \lambda_1^{(2)} = \text{Tr}(\boldsymbol{H}_1)$, $\lambda_2^{(1)} + \lambda_2^{(2)} = \text{Tr}(\boldsymbol{H}_2)$. Therefore,

$$\begin{aligned} \|\boldsymbol{W}_{\text{OV}}^{\text{F}}\|_F^2 &= \text{Tr}(\boldsymbol{H}_1) + (n-1)\text{Tr}(\boldsymbol{H}_2) \\ &= n(C_{A1} + C_{A2} + C_{D1} + C_{D2}) \\ &= n \left[ \begin{array}{c} m_{\text{train}}((p_1 + p_2)^2 + (n-1)p_2^2 + (q_1 + q_2)^2 + (n-1)q_2^2) \\ +m_{\text{test}}((f_1 + f_2)^2 + (n-1)f_2^2 + (g_1 + g_2)^2 + (n-1)g_2^2) \\ +\beta_1^2 + \gamma_2^2 + \beta_2^2 + \gamma_1^2 \end{array} \right]. \end{aligned}$$

This proves Lemma 10. $\qquad\square$

**Lemma 11.** *Let the expression $\|\boldsymbol{W}_{\mathsf{OV}}\|_F$ be defined as Equation* (35). *The closed-form minimum of $\|\boldsymbol{W}_{\mathsf{OV}}\|_F$ is given by*

$$\min \|\boldsymbol{W}_{\mathsf{OV}}\|_F = \sqrt{n}\left(2m_{\mathsf{train}}(1 - 1/n) + m_{\mathsf{test}}(1 - 1/n) + 1\right)^{1/2},$$

*where the minimum is achieved at $p_1^\star = f_1^\star = q_1^\star = 1, p_2^\star = f_2^\star = q_2^\star = -1/n, \beta_1^\star = \gamma_2^\star = 1/2, \gamma_1^\star = \beta_2^\star = -1/2$, and $g_1^\star = g_2^\star = 0$.*

*Proof of Lemma* 11. Given the constraints in Equation (34), we have that

$$\|\boldsymbol{W}_{\mathsf{OV}}\|_F \geq \left(n\left[\begin{array}{c} m_{\mathsf{train}}(np_2^2 + 2p_2 + 1 + nq_2^2 + 2q_2 + 1) \\ + m_{\mathsf{test}}(nf_2^2 + 2f_2 + 1 + (g_1 + g_2)^2 + (n-1)g_2^2) + 1) \end{array}\right]\right)^{1/2},$$

where the equality holds when $p_1^\star = f_1^\star = q_1^\star = 1$ and $\beta_1^\star = \gamma_2^\star = 1/2, \gamma_1^\star = \beta_2^\star = -1/2$. This lower bound is a quadratic form, which we can show that $p_2^\star = f_2^\star = q_2^\star = -1/n$ and $g_1^\star = g_2^\star = 0$ gives its minimum. This finishes the proof of Lemma 11. $\qquad\square$

**Theorem 7.** *Therefore, we have that*

$$\boldsymbol{W}_{\mathsf{OV}} = \left[\overbrace{\begin{matrix} \boldsymbol{I}_n - \boldsymbol{E}_n/n \cdots \boldsymbol{I}_n - \boldsymbol{E}_n/n \\ \boldsymbol{I}_n - \boldsymbol{E}_n/n \cdots \boldsymbol{I}_n - \boldsymbol{E}_n/n \end{matrix}}^{m_{\mathsf{train}}\text{ blocks}} \quad \overbrace{\begin{matrix} \boldsymbol{I}_n - \boldsymbol{E}_n/n \cdots \boldsymbol{I}_n - \boldsymbol{E}_n/n \\ \boldsymbol{0}\cdots\boldsymbol{0} \end{matrix}}^{m_{\mathsf{test}}\text{ blocks}} \quad \begin{matrix} \boldsymbol{1}_n/2 & -\boldsymbol{1}_n/2 \\ -\boldsymbol{1}_n/2 & \boldsymbol{1}_n/2 \end{matrix}\right]. \quad (36)$$

$$\underbrace{\phantom{\boldsymbol{I}_n - \boldsymbol{E}_n/n \cdots \boldsymbol{I}_n - \boldsymbol{E}_n/n}}_{m_{\mathsf{train}}\text{ blocks}} \qquad \underbrace{\phantom{\boldsymbol{I}_n - \boldsymbol{E}_n/n \cdots \boldsymbol{I}_n}}_{m_{\mathsf{test}}\text{ blocks}}$$

*It implies that for any $s \in \mathcal{S}_{\mathsf{test}}$ and any $a' \in \mathcal{A}_2 \setminus \{a^*(s, r_2)\}$, it holds that $h_{(s,r_2),a'} = 0$. This proves Theorem* 6.

*Proof of Theorem* 7. We take the results from Lemma 11 into the restricted form Equation (33) and get Equation (36). Given any $s_{i,j} \in \mathcal{S}_{\mathsf{test}}$ and $c_k \in \mathcal{A}_2$ where $k \neq i$, we have

$$h_{(s_{i,j},r_2),c_k} = [\boldsymbol{0}_n^\top, \boldsymbol{e}_i^\top - \boldsymbol{e}_k^\top]\boldsymbol{W}_{\mathsf{OV}} \cdot \begin{bmatrix} 0 \\ \vdots \\ \boldsymbol{e}_i \\ 0 \\ \vdots \\ 0 \\ 1 \end{bmatrix} = 0.$$

This proves Theorem 7. $\qquad\square$

## B  Proof of Section 4.2

In this section, we provide the complete proof of Theorem 3 via gradient flow analysis. We first present several key lemmas to prove Theorem 3. In Lemma 12, we give the gradient form of the reparameterized parameters to simplify the analysis. In Lemma 13, we prove the Lipschitzness of the gradient flow, and Lemma 14 proves that the gradient flow also satisfies permutation equivariance. Using these two lemmas, we are able to conclude the symmetry in parameters by employing a standard argument for the uniqueness of ODE solutions in Lemma 15.

**Useful notations.** We introduce useful notations for this section. Let $\mathsf{vec}(\boldsymbol{A})$ denote the vectorization of a matrix $\boldsymbol{A}$. Specifically, for $\boldsymbol{A} = (a_{ij})_{i=1,j=1}^{m,n} \in \mathbb{R}^{m \times n}$, we have $\mathsf{vec}(\boldsymbol{A}) = (a_{11}, \ldots, a_{1n}, a_{21}, \ldots, a_{2n}, \ldots, a_{m1}, \ldots, a_{mn})^\top \in \mathbb{R}^{mn}$. We use $\|\boldsymbol{A}\|_{\mathsf{op}}$ to denote the operator norm of a matrix $\boldsymbol{A}$. For any vector $\boldsymbol{u} \in \mathbb{R}^n$, let $\mathsf{diag}(\boldsymbol{u}) \in \mathbb{R}^{n \times n}$ denote a diagonal matrix whose diagonal entries are corresponding entries in $\boldsymbol{u}$. Let $\mathbb{1}(\mathcal{E})$ denote the indicator function of an event $\mathcal{E}$.

**Lemma 12.** *Recall that for any $z \in \mathcal{V}$, we have $W_{\mathsf{OV}}(a, z) = \boldsymbol{e}_a^\top \boldsymbol{W}_{\mathsf{OV}} \boldsymbol{e}_z$ and $W_{\mathsf{KQ}}(z) = \boldsymbol{e}_z^\top \boldsymbol{W}_{\mathsf{KQ}} \boldsymbol{e}_{\texttt{<EOS>}}$. Then for a fixed $(a, s) \in \mathcal{A} \times \mathcal{S}$, the gradient of $W_{\mathsf{OV}}(a, s)$ is given by:*

$$\partial_{W_{\mathsf{OV}}(a,s)}\mathcal{L}(\tilde{\boldsymbol{\theta}}) = -W_{\mathsf{KQ}}(s) \cdot \sum_{r \in \mathcal{R}} p(s, r) \cdot \left(\mathbb{1}(a = a^*(s, r)) - p_{\tilde{\boldsymbol{\theta}}}(a|s, r)\right).$$

*Similarly, for a fixed $(a, r) \in \mathcal{A} \times \mathcal{R}$, we have:*

$$\partial_{W_{\mathsf{OV}}(a,r)}\mathcal{L}(\tilde{\boldsymbol{\theta}}) = -W_{\mathsf{KQ}}(r) \cdot \sum_{s \in \mathcal{S}} p(s, r) \cdot \left(\mathbb{1}(a = a^*(s, r)) - p_{\tilde{\boldsymbol{\theta}}}(a|s, r)\right).$$

*Moreover, for* `<EOS>` *token and a fixed $a \in \mathcal{A}$, we have:*

$$\partial_{W_{\mathsf{OV}}(a, \texttt{<EOS>})}\mathcal{L}(\tilde{\boldsymbol{\theta}}) = -W_{\mathsf{KQ}}(\texttt{<EOS>}) \cdot \sum_{s \in \mathcal{S}} \sum_{r \in \mathcal{R}} p(s, r) \cdot \left(\mathbb{1}(a = a^*(s, r)) - p_{\tilde{\boldsymbol{\theta}}}(a|s, r)\right).$$

*Lastly, for $s \in \mathcal{S}$, we have:*

$$\partial_{W_{\mathsf{KQ}}(s)}\mathcal{L}(\tilde{\boldsymbol{\theta}}) = -\sum_{a \in \mathcal{A}} W_{\mathsf{OV}}(a, s) \sum_{r \in \mathcal{R}} p(s, r) \cdot \left(\mathbb{1}(a = a^*(s, r)) - p_{\tilde{\boldsymbol{\theta}}}(a|s, r)\right).$$

*Proof.* First, note that the logit function can be written as:

$$f_{\tilde{\boldsymbol{\theta}}}(z_{1:T}, a) = \boldsymbol{e}_a^\top \boldsymbol{W}_{\mathsf{OV}} \boldsymbol{X}^\top \boldsymbol{X} \boldsymbol{W}_{\mathsf{KQ}} \boldsymbol{x}_T = \sum_{t \in [T]} W_{\mathsf{OV}}(a, z_t) W_{\mathsf{KQ}}(z_t). \tag{37}$$

Then for any $z \in \mathcal{V}$, we get

$$\begin{aligned}
\partial_{W_{\mathsf{OV}}(a,z)} \left(\boldsymbol{e}_{a'}^\top \boldsymbol{W}_{\mathsf{OV}} \boldsymbol{X}^\top \boldsymbol{X} \boldsymbol{W}_{\mathsf{KQ}} \boldsymbol{x}_T\right) &= \mathbb{1}(a = a') C(z_{1:T}, z) W_{\mathsf{KQ}}(z), \\
\partial_{W_{\mathsf{KQ}}(z)} \left(\boldsymbol{e}_{a'}^\top \boldsymbol{W}_{\mathsf{OV}} \boldsymbol{X}^\top \boldsymbol{X} \boldsymbol{W}_{\mathsf{KQ}} \boldsymbol{x}_T\right) &= C(z_{1:T}, z) W_{\mathsf{OV}}(a', z),
\end{aligned} \tag{38}$$

where $C(z_{1:T}, z)$ is the number of occurrences of $z$ in the sequence $z_{1:T}$. Recall that we can simplify the loss function in (4) as

$$\begin{aligned}
\mathcal{L}(\tilde{\boldsymbol{\theta}}) &= \mathbb{E}_{z_{1:T+1}} \left[-\log \frac{\exp(\boldsymbol{e}_{z_{T+1}}^\top \boldsymbol{W}_{\mathsf{OV}} \boldsymbol{X}^\top \boldsymbol{X} \boldsymbol{W}_{\mathsf{KQ}} \boldsymbol{x}_T)}{\sum_{z' \in \mathcal{A}} \exp(\boldsymbol{e}_{z'}^\top \boldsymbol{W}_{\mathsf{OV}} \boldsymbol{X}^\top \boldsymbol{X} \boldsymbol{W}_{\mathsf{KQ}} \boldsymbol{x}_T)}\right] \\
&= \mathbb{E}_{z_{1:T+1}}[-\boldsymbol{e}_{z_{T+1}}^\top \boldsymbol{W}_{\mathsf{OV}} \boldsymbol{X}^\top \boldsymbol{X} \boldsymbol{W}_{\mathsf{KQ}} \boldsymbol{x}_T + \log \sum_{z' \in \mathcal{A}} \exp(\boldsymbol{e}_{z'}^\top \boldsymbol{W}_{\mathsf{OV}} \boldsymbol{X}^\top \boldsymbol{X} \boldsymbol{W}_{\mathsf{KQ}} \boldsymbol{x}_T)].
\end{aligned}$$

Using (38), we get

$$\begin{aligned}
\partial_{W_{\mathsf{OV}}(a,z)}\mathcal{L}(\tilde{\boldsymbol{\theta}}) &= W_{\mathsf{KQ}}(z) \mathbb{E}_{z_{1:T+1}} \Big[- \mathbb{1}(a = z_{T+1}) C(z_{1:T}, z) \\
&\qquad\qquad + \frac{\sum_{z' \in \mathcal{A}} \exp(\boldsymbol{e}_{z'}^\top \boldsymbol{W}_{\mathsf{OV}} \boldsymbol{X}^\top \boldsymbol{X} \boldsymbol{W}_{\mathsf{KQ}} \boldsymbol{x}_T) \mathbb{1}(a = z') C(z_{1:T}, z)}{\sum_{z' \in \mathcal{A}} \exp(\boldsymbol{e}_{z'}^\top \boldsymbol{W}_{\mathsf{OV}} \boldsymbol{X}^\top \boldsymbol{X} \boldsymbol{W}_{\mathsf{KQ}} \boldsymbol{x}_T)} \Big] \\
&= -W_{\mathsf{KQ}}(z) \mathbb{E}_{z_{1:T}} \Big[C(z_{1:T}, z) \big(\mathbb{1}(a = a^*(z_{1:T})) - p_{\tilde{\boldsymbol{\theta}}}(a|z_{1:T})\big)\Big].
\end{aligned}$$

Similarly,

$$\begin{aligned}
\partial_{W_{\mathsf{KQ}}(z)}\mathcal{L}(\tilde{\boldsymbol{\theta}}) &= \mathbb{E}_{z_{1:T}} \Big[- C(z_{1:T}, z) W_{\mathsf{OV}}(a^*(z_{1:T}), z) + \sum_{a \in \mathcal{A}} p_{\tilde{\boldsymbol{\theta}}}(a|z_{1:T}) W_{\mathsf{OV}}(a, z) C(z_{1:T}, z)\Big] \\
&= \mathbb{E}_{z_{1:T}} \Big[C(z_{1:T}, z) \big(- W_{\mathsf{OV}}(a^*(z_{1:T}), z) + \sum_{a \in \mathcal{A}} p_{\tilde{\boldsymbol{\theta}}}(a|z_{1:T}) W_{\mathsf{OV}}(a, z)\big)\Big] \\
&= -\sum_{a \in \mathcal{A}} W_{\mathsf{OV}}(a, z) \mathbb{E}_{z_{1:T}} \Big[C(z_{1:T}, z) \big(\mathbb{1}(a = a^*(z_{1:T})) - p_{\tilde{\boldsymbol{\theta}}}(a|z_{1:T})\big)\Big].
\end{aligned}$$

Let $p(s) = \sum_{z_{1:T}} p(z_{1:T}) \mathbb{1}(s \in z_{1:T}) = \mathbb{E}_{\mathcal{D}_{\mathsf{train}}}[\mathbb{1}(s \in z_{1:T})]$, which is the marginal probability of observing $s$ under the data-generating distribution $\mathcal{D}_{\mathsf{train}}$. Similarly, we can define $p(r)$ and $p(s, r)$

for any $r \in \mathcal{R}$ and any $(s, r) \in \mathcal{S} \times \mathcal{R}$. Then for a fixed $(a, s) \in \mathcal{A} \times \mathcal{S}$, we have

$$
\begin{aligned}
\partial_{W_{\mathsf{OV}}(a,s)}\mathcal{L}(\tilde{\boldsymbol{\theta}}) &= -W_{\mathsf{KQ}}(s)\mathbb{E}_{z_{1:T} \sim \mathcal{D}_{\mathrm{train}}}\Big[C(z_{1:T}, s)\big(\mathbb{1}(a = a^*(z_{1:T})) - p_{\tilde{\boldsymbol{\theta}}}(a|z_{1:T})\big)\Big] \\
&\stackrel{(a)}{=} -W_{\mathsf{KQ}}(s)\mathbb{E}_{z_{1:T} \sim \mathcal{D}_{\mathrm{train}}}\Big[\mathbb{1}(s \in z_{1:T})\big(\mathbb{1}(a = a^*(z_{1:T})) - p_{\tilde{\boldsymbol{\theta}}}(a|z_{1:T})\big)\Big] \\
&= -W_{\mathsf{KQ}}(s)\sum_{z_{1:T}} p(z_{1:T})\mathbb{1}(s \in z_{1:T})\big(\mathbb{1}(a = a^*(z_{1:T})) - p_{\tilde{\boldsymbol{\theta}}}(a|z_{1:T})\big) \\
&= -W_{\mathsf{KQ}}(s)p(s)\sum_{z_{1:T}} \Big(\frac{p(z_{1:T})\mathbb{1}(s \in z_{1:T})}{\sum_{z'_{1:T}} p(z'_{1:T})\mathbb{1}(s \in z'_{1:T})}\big(\mathbb{1}(a = a^*(z_{1:T})) - p_{\tilde{\boldsymbol{\theta}}}(a|z_{1:T})\big)\Big) \\
&= -W_{\mathsf{KQ}}(s) \cdot p(s) \cdot \mathbb{E}_{z_{1:T}}\Big[\mathbb{1}(a = a^*(z_{1:T})) - p_{\tilde{\boldsymbol{\theta}}}(a|z_{1:T}) \mid s \in z_{1:T}\Big] \\
&= -W_{\mathsf{KQ}}(s) \cdot \sum_{r \in \mathcal{R}} p(s, r) \cdot \mathbb{E}_{z_{1:T}}\Big[\mathbb{1}(a = a^*(z_{1:T})) - p_{\tilde{\boldsymbol{\theta}}}(a|z_{1:T}) \mid s, r \in z_{1:T}\Big] \\
&\stackrel{(b)}{=} -W_{\mathsf{KQ}}(s) \cdot \sum_{r \in \mathcal{R}} p(s, r) \cdot \big(\mathbb{1}(a = a^*(s, r)) - p_{\tilde{\boldsymbol{\theta}}}(a|s, r)\big),
\end{aligned}
$$

where (a) comes from the fact that $C(z_{1:T}, s) = \mathbb{1}(s \in z_{1:T})$ as $s$ occurs at most once in the sequence from the task definition and (b) comes from the fact that $z_{1:T}$ only contains $(s, r, \texttt{<EOS>})$ and thus $p_{\tilde{\boldsymbol{\theta}}}(\cdot|z_{1:T}) = p_{\tilde{\boldsymbol{\theta}}}(\cdot|s, r)$.

Similarly, consider a fixed $(a, r) \in \mathcal{A} \times \mathcal{R}$, we have:

$$
\begin{aligned}
\partial_{W_{\mathsf{OV}}(a,r)}\mathcal{L}(\tilde{\boldsymbol{\theta}}) &= -W_{\mathsf{KQ}}(r) \cdot p(r) \cdot \mathbb{E}_{z_{1:T}}\Big[\mathbb{1}(a = a^*(z_{1:T})) - p_{\tilde{\boldsymbol{\theta}}}(a|z_{1:T}) \mid s \in z_{1:T}\Big] \\
&= -W_{\mathsf{KQ}}(r) \cdot \sum_{s \in \mathcal{S}} p(s, r) \cdot \big(\mathbb{1}(a = a^*(s, r)) - p_{\tilde{\boldsymbol{\theta}}}(a|s, r)\big).
\end{aligned}
$$

Lastly for any $a \in \mathcal{A}$, we have:

$$
\begin{aligned}
\partial_{W_{\mathsf{OV}}(a,\texttt{<EOS>})}\mathcal{L}(\tilde{\boldsymbol{\theta}}) &= -W_{\mathsf{KQ}}(\texttt{<EOS>})\sum_{z_{1:T}} p(z_{1:T})\big(\mathbb{1}(a = a^*(z_{1:T})) - p_{\tilde{\boldsymbol{\theta}}}(a|z_{1:T})\big) \\
&= -W_{\mathsf{KQ}}(\texttt{<EOS>}) \cdot \sum_{s \in \mathcal{S}}\sum_{r \in \mathcal{R}} p(s, r) \cdot \big(\mathbb{1}(a = a^*(s, r)) - p_{\tilde{\boldsymbol{\theta}}}(a|s, r)\big).
\end{aligned}
$$

We conclude by proving the gradient in terms of $W_{\mathsf{KQ}}(s)$ for any $s \in \mathcal{S}$. We have:

$$
\begin{aligned}
\partial_{W_{\mathsf{KQ}}(s)}\mathcal{L}(\tilde{\boldsymbol{\theta}}) &= -\sum_{a \in \mathcal{A}} W_{\mathsf{OV}}(a, s) \cdot p(s) \cdot \mathbb{E}_{z_{1:T}}\Big[\mathbb{1}(a = a^*(z_{1:T})) - p_{\tilde{\boldsymbol{\theta}}}(a|z_{1:T}) \mid s \in z_{1:T}\Big] \\
&= -\sum_{a \in \mathcal{A}} W_{\mathsf{OV}}(a, s)\sum_{r \in \mathcal{R}} p(s, r) \cdot \big(\mathbb{1}(a = a^*(s, r)) - p_{\tilde{\boldsymbol{\theta}}}(a|s, r)\big).
\end{aligned}
$$

$\square$

**Lemma 13** (Lipschitz gradient). *Let $d_0 := (|\mathcal{A}| + 1) \cdot |\mathcal{V}|$ and concatenate the parameters by*

$$
\boldsymbol{w} = \begin{bmatrix} \mathsf{vec}(\boldsymbol{W}_{\mathsf{OV}}) \\ \boldsymbol{W}_{\mathsf{KQ}} \end{bmatrix} \in \mathbb{R}^{d_0}.
$$

*Let $F(\boldsymbol{w}(t))$ be the vector field in the ODE*

$$
\dot{\boldsymbol{w}}(t) = F\big(\boldsymbol{w}(t)\big) = -\nabla_{\boldsymbol{w}}\mathcal{L}(\boldsymbol{w}),
$$

*where we omit the index $t$ and set $\boldsymbol{w}(t) = \boldsymbol{w}$ when the context is clear and $\mathcal{L}(\boldsymbol{w})$ is the cross entropy loss given by:*

$$
\mathcal{L}(\boldsymbol{w}) = \sum_{s,r} p(s, r)\left[-\log \frac{\exp\big(f_{\boldsymbol{w}}\big((s, r), a^*(s, r)\big)\big)}{\sum_{a \in \mathcal{A}} \exp\big(f_{\boldsymbol{w}}\big((s, r), a\big)\big)}\right],
$$

*where the logit function $f_{\boldsymbol{w}}((s,r),a)$ follows Eq. (37) with $z_{1:T} = (s,r,\texttt{<EOS>})$. Assuming that $\|\boldsymbol{w}\|_2 \leq R$ for some constant $R > 0$, then for any fixed $(s,r,a)$, $f_{\boldsymbol{w}}((s,r),a)$ is Lipschitz in $\boldsymbol{w}$ with constant $L_1 := 4R$, i.e., for any $\boldsymbol{w}_1, \boldsymbol{w}_2 \in \mathbb{R}^{d_0}$ with $\|\boldsymbol{w}_1\|_2 \leq R, \|\boldsymbol{w}_2\|_2 \leq R$, it holds that*

$$f_{\boldsymbol{w}_1}((s,r),a) - f_{\boldsymbol{w}_2}((s,r),a) \leq L_1 \|\boldsymbol{w}_1 - \boldsymbol{w}_2\|.$$

*Moreover, there exists a constant $L := 4|\mathcal{A}|(L_1 R + 1)$ such that*

$$\|F(\boldsymbol{w}_1) - F(\boldsymbol{w}_2)\| \leq L \|\boldsymbol{w}_1 - \boldsymbol{w}_2\|.$$

*Proof.* Observing that $f_{\boldsymbol{w}}((s,r),a)$ is bilinear in $\boldsymbol{w}$, i.e.,

$$
\begin{aligned}
f_{\boldsymbol{w}}((s,r),a) &= \sum_{t \in [T]} W_{\mathsf{OV}}(a, z_t) W_{\mathsf{KQ}}(z_t) \\
&= W_{\mathsf{OV}}(a,s) W_{\mathsf{KQ}}(s) + W_{\mathsf{OV}}(a,r) W_{\mathsf{KQ}}(r) + W_{\mathsf{OV}}(a,\texttt{<EOS>}) W_{\mathsf{KQ}}(\texttt{<EOS>}) \\
&= \boldsymbol{w}^\top \boldsymbol{M}((s,r),a) \boldsymbol{w},
\end{aligned}
$$

where $\boldsymbol{M}((s,r),a) \in \mathbb{R}^{d_0 \times d_0}$ has only three non-zero entries, which are equal to one, where the positions depend on $((s,r),a)$. We omit the index $((s,r),a)$ when the context is clear. Then for any $\boldsymbol{w}_1, \boldsymbol{w}_2 \in \mathbb{R}^{d_0}$ with Euclidean norm bounded by $R$, we have

$$
\begin{aligned}
&f_{\boldsymbol{w}_1}((s,r),a) - f_{\boldsymbol{w}_2}((s,r),a) \\
&= \boldsymbol{w}_1^\top \boldsymbol{M} \boldsymbol{w}_1 - \boldsymbol{w}_2^\top \boldsymbol{M} \boldsymbol{w}_2 \\
&= \boldsymbol{w}_1^\top \boldsymbol{M} \boldsymbol{w}_1 - \boldsymbol{w}_1^\top \boldsymbol{M} \boldsymbol{w}_2 + \boldsymbol{w}_1^\top \boldsymbol{M} \boldsymbol{w}_2 - \boldsymbol{w}_2^\top \boldsymbol{M} \boldsymbol{w}_2 \\
&\leq (\|\boldsymbol{w}_1\| + \|\boldsymbol{w}_2\|) \cdot \|\boldsymbol{M}\|_F \cdot \|\boldsymbol{w}_1 - \boldsymbol{w}_2\| \\
&\overset{(a)}{\leq} 4R \|\boldsymbol{w}_1 - \boldsymbol{w}_2\| = L_1 \|\boldsymbol{w}_1 - \boldsymbol{w}_2\|,
\end{aligned}
\tag{39}
$$

where (a) uses $\|\boldsymbol{M}\|_F = \sqrt{3} \leq 2$. Moreover, the gradient of $f_{\boldsymbol{w}}((s,r),a)$ is also Lipschitz:

$$\nabla f_{\boldsymbol{w}_1}((s,r),a) - \nabla f_{\boldsymbol{w}_2}((s,r),a) = (\boldsymbol{M} + \boldsymbol{M}^\top)(\boldsymbol{w}_1 - \boldsymbol{w}_2) \leq 4 \|\boldsymbol{w}_1 - \boldsymbol{w}_2\|. \tag{40}$$

Now let $p_k(\boldsymbol{u}) = \frac{\exp(u_k)}{\sum_{j=1}^K \exp(u_j)}$ and $g_k(\boldsymbol{u}) = -\log p_k(\boldsymbol{u})$ for $\boldsymbol{u} \in \mathbb{R}^K$ where $K = |\mathcal{A}|$, and denote $p(\boldsymbol{u}) = (p_1(\boldsymbol{u}), \ldots, p_K(\boldsymbol{u}))^\top \in \mathbb{R}^K$, $g(\boldsymbol{u}) = (g_1(\boldsymbol{u}), \ldots, g_K(\boldsymbol{u}))^\top \in \mathbb{R}^K$. Then the gradient and hessian of $g_k(\boldsymbol{u})$ are given by:

$$
\begin{aligned}
\nabla g_k(\boldsymbol{u}) &= p(\boldsymbol{u}) - \boldsymbol{e}_k \in \mathbb{R}^K, \\
H(\boldsymbol{u}) &= \mathsf{diag}(p(\boldsymbol{u})) - p(\boldsymbol{u}) p(\boldsymbol{u})^\top \in \mathbb{R}^{K \times K}.
\end{aligned}
$$

By the mean-value theorem, we have:

$$\|\nabla g_k(\boldsymbol{u}) - \nabla g_k(\boldsymbol{v})\| \leq \sup_{w \in \mathbb{R}^K} \|H(\boldsymbol{w})\|_{\mathsf{op}} \|\boldsymbol{u} - \boldsymbol{v}\| \leq \|\boldsymbol{u} - \boldsymbol{v}\|. \tag{41}$$

Let $\boldsymbol{u}_{\boldsymbol{w}}(s,r) = f_{\boldsymbol{w}}((s,r),\cdot) \in \mathbb{R}^{|\mathcal{A}|}$ be the logit vector for input $(s,r)$ and denote

$$\alpha_a(\boldsymbol{u}_{\boldsymbol{w}}(s,r)) := [\nabla g_{a^*(s,r)}(\boldsymbol{u}_{\boldsymbol{w}}(s,r))]_a = p_a(\boldsymbol{u}_{\boldsymbol{w}}(s,r)) - \mathbb{1}(a = a^*(s,r)).$$

Then the gradient of the loss function can be written as

$$\nabla \mathcal{L}(\boldsymbol{w}) = \sum_{s,r} p(s,r) \nabla_{\boldsymbol{w}} g_{a^*(s,r)}(\boldsymbol{u}_{\boldsymbol{w}}(s,r)) = \sum_{s,r} p(s,r) \sum_{a \in \mathcal{A}} \alpha_a(\boldsymbol{u}_{\boldsymbol{w}}(s,r)) \nabla f_{\boldsymbol{w}}((s,r),a).$$

For brevity let $\nabla f_{\boldsymbol{w},a} := \nabla f_{\boldsymbol{w}}((s,r),a)$ and note that $\|\nabla f_{\boldsymbol{w},a}\| = \|(\boldsymbol{M} + \boldsymbol{M}^\top)\boldsymbol{w}\| \leq 4R$. Combining Eq. (39) and (41) we have:

$$\|F(\boldsymbol{w}_1) - F(\boldsymbol{w}_2)\|$$
$$=\|\nabla\mathcal{L}(\boldsymbol{w}_1) - \nabla\mathcal{L}(\boldsymbol{w}_2)\|$$
$$\leq \sum_{s,r} p(s,r) \left\| \sum_{a\in\mathcal{A}} \alpha_a(\boldsymbol{u}_{\boldsymbol{w}_1}(s,r))\nabla f_{\boldsymbol{w}_1,a} - \sum_{a\in\mathcal{A}} \alpha_a(\boldsymbol{u}_{\boldsymbol{w}_2}(s,r))\nabla f_{\boldsymbol{w}_2,a} \right\|$$
$$\leq \max_{s,r} \left\| \sum_{a\in\mathcal{A}} \Big( \big(\alpha_a(\boldsymbol{u}_{\boldsymbol{w}_1}(s,r)) - \alpha_a(\boldsymbol{u}_{\boldsymbol{w}_2}(s,r))\big)\nabla f_{\boldsymbol{w}_1,a} + \alpha_a(\boldsymbol{u}_{\boldsymbol{w}_2}(s,r))\big(\nabla f_{\boldsymbol{w}_1,a} - \nabla f_{\boldsymbol{w}_2,a}\big) \Big) \right\|$$
$$\overset{(a)}{\leq} \max_{s,r} \left( \sum_{a\in\mathcal{A}} \|\nabla f_{\boldsymbol{w}_1,a}\| \cdot \|\boldsymbol{u}_{\boldsymbol{w}_1}(s,r) - \boldsymbol{u}_{\boldsymbol{w}_2}(s,r)\| + 4|\mathcal{A}|\max_a |\alpha_a(\boldsymbol{u}_{\boldsymbol{w}_2}(s,r))| \cdot \|\boldsymbol{w}_1 - \boldsymbol{w}_2\| \right)$$
$$\overset{(b)}{\leq} \|\boldsymbol{w}_1 - \boldsymbol{w}_2\| \cdot \left( L_1 \sum_{a\in\mathcal{A}} \|\nabla f_{\boldsymbol{w}_1,a}\| + 4|\mathcal{A}| \right)$$
$$\leq 4|\mathcal{A}|\big(L_1 R + 1\big)\|\boldsymbol{w}_1 - \boldsymbol{w}_2\|,$$

where (a) follows Eq. (40) and Eq. (41) and (b) uses Eq. (39) and $|\alpha_a(\boldsymbol{u}_{\boldsymbol{w}_2}(s,r))| \leq 1$ for any $a \in \mathcal{A}$. $\qquad\square$

Before proceeding, we provide the following definition.

**Definition 1** (Data permutation)**.** *Let* $d_0 := (|\mathcal{A}| + 1) \cdot |\mathcal{V}|$. *Consider the flattened parameter* $\boldsymbol{w}$ *defined as*

$$\boldsymbol{w} = \begin{bmatrix} \mathsf{vec}(\boldsymbol{W}_{\mathsf{OV}}) \\ \boldsymbol{W}_{\mathsf{KQ}} \end{bmatrix} \in \mathbb{R}^{d_0}.$$

*Recall* $|\mathcal{A}_1| = |\mathcal{A}_2| = n$. *Let* $\sigma$ *be any permutation over* $[n]$ *where* $\sigma(i) \neq i$ *for any* $i \in [n]$ *and* $\pi : \mathcal{V} \to \mathcal{V}$ *be a permutation function determined by* $\sigma$. *Specifically, for any* $i \in [n], j \in [m]$, *we have*

$$\pi(b_i) = b_{\sigma(i)}, \pi(c_i) = c_{\sigma(i)}, \pi(s_{i,j}) = s_{\sigma(i),j},$$

*and we have* $\pi(v) = v$ *for* $v \in \{r_1, r_2, \mathtt{<EOS>}\}$. *Moreover, let* $\boldsymbol{P}_\pi$ *be a permutation matrix built on permutation* $\pi$ *defined as follows. First, we have*

$$\boldsymbol{P} := \boldsymbol{P}_\pi = \begin{bmatrix} \boldsymbol{P}_{\mathsf{OV}} & \boldsymbol{0} \\ \boldsymbol{0} & \boldsymbol{P}_{\mathsf{KQ}} \end{bmatrix} \in \mathbb{R}^{d_0 \times d_0},$$

*where we omit the subscript* $\pi$ *for brevity. We denote the resulting* $\mathsf{OV}$ *block in* $\boldsymbol{w}$ *after permutation as* $(\boldsymbol{Pw})_{\mathsf{OV}} := \boldsymbol{P}_{\mathsf{OV}}\mathsf{vec}(\boldsymbol{W}_{\mathsf{OV}}) \in \mathbb{R}^{|\mathcal{V}| \cdot |\mathcal{A}|}$. *Similarly we denote* $(\boldsymbol{Pw})_{\mathsf{KQ}} := \boldsymbol{P}_{\mathsf{KQ}}\boldsymbol{W}_{\mathsf{KQ}} \in \mathbb{R}^{|\mathcal{V}|}$. *Then, each entry of* $\boldsymbol{Pw}$ *is defined as follows:*

- $\forall s \in \mathcal{S}_{\mathsf{test}}, \quad \forall a \in \mathcal{A}_2 : \quad (\boldsymbol{Pw})_{\mathsf{OV}}(a,s) = W_{\mathsf{OV}}(\pi(a),s),$

- $\forall s \in \mathcal{S}_{\mathsf{train}}, \quad \forall a \in \mathcal{A} : \quad (\boldsymbol{Pw})_{\mathsf{OV}}(a,s) = W_{\mathsf{OV}}(\pi(a),\pi(s)),$

- $\forall v \in \{r_1, r_2, \mathtt{<EOS>}\}, \quad \forall a \in \mathcal{A}_2 : \quad (\boldsymbol{Pw})_{\mathsf{OV}}(a,v) = W_{\mathsf{OV}}(\pi(a),v),$

- $\forall s \in \mathcal{S}_{\mathsf{train}} : \quad (\boldsymbol{Pw})_{\mathsf{KQ}}(s) = W_{\mathsf{KQ}}(\pi(s)),$

- *Otherwise,* $(\boldsymbol{Pw})_{\mathsf{OV}}(a,v) = W_{\mathsf{OV}}(a,v), (\boldsymbol{Pw})_{\mathsf{KQ}}(v) = W_{\mathsf{KQ}}(v).$

**Lemma 14** (Gradient is permutation equivariant)**.** *Recall the flattened parameter* $\boldsymbol{w}$ *and the permutation matrix* $\boldsymbol{P}$ *defined in Definition 1. Denote* $F(\boldsymbol{w}) = -\nabla\mathcal{L}(\boldsymbol{w})$ *following Lemma 13, then we have*

$$\forall s \in \mathcal{S}_{\mathsf{test}}, \quad \forall a \in \mathcal{A}_2 : \quad [F(\boldsymbol{Pw})]_{\mathsf{OV}}(a,s) = [\boldsymbol{P}F(\boldsymbol{w})]_{\mathsf{OV}}(a,s), \tag{42a}$$
$$\forall s \in \mathcal{S}_{\mathsf{train}}, \quad \forall a \in \mathcal{A} : \quad [F(\boldsymbol{Pw})]_{\mathsf{OV}}(a,s) = [\boldsymbol{P}F(\boldsymbol{w})]_{\mathsf{OV}}(a,s), \tag{42b}$$
$$\forall v \in \mathcal{R} \cup \{\mathtt{<EOS>}\}, \quad \forall a \in \mathcal{A}_2 : \quad [F(\boldsymbol{Pw})]_{\mathsf{OV}}(a,v) = [\boldsymbol{P}F(\boldsymbol{w})]_{\mathsf{OV}}(a,v), \tag{42c}$$
$$\forall s \in \mathcal{S}_{\mathsf{train}} : \quad [F(\boldsymbol{Pw})]_{\mathsf{KQ}}(s) = [\boldsymbol{P}F(\boldsymbol{w})]_{\mathsf{KQ}}(s), \tag{42d}$$

*which implies the gradeint of* $\mathcal{L}$ *w.r.t* $\boldsymbol{w}$ *is permutation equivariant, i.e.,*

$$\nabla\mathcal{L}(\boldsymbol{Pw}) = \boldsymbol{P}\nabla\mathcal{L}(\boldsymbol{w}).$$

*Proof.* We examine Eq. (42a) - (42d) one by one.

**Part 1: Proof of Eq. (42a).** Given a fixed $s \in \mathcal{S}_{\text{test}}, a \in \mathcal{A}_2$, we will prove

$$[F(\boldsymbol{P}\boldsymbol{w})]_{\text{OV}}(a, s) = [\boldsymbol{P}F(\boldsymbol{w})]_{\text{OV}}(a, s). \tag{43}$$

Using Lemma 12, we have:

$$\begin{aligned}
[F(\boldsymbol{w})]_{\text{OV}}(a, s) &= -\partial_{W_{\text{OV}}(a,s)} \mathcal{L}(\boldsymbol{w}) \\
&= W_{\text{KQ}}(s) \sum_{r \in \mathcal{R}} p(s, r)[\mathbb{1}(a = a^*(s, r)) - p_{\boldsymbol{w}}(a|s, r)] \\
&\stackrel{(a)}{=} W_{\text{KQ}}(s) \cdot p(s, r_1)[\mathbb{1}(a = a^*(s, r_1)) - p_{\boldsymbol{w}}(a|s, r_1)],
\end{aligned}$$

where (a) comes from the fact that $p(s, r_2) = 0$ since $s \in \mathcal{S}_{\text{test}}$. Then for the right side of Eq. (43), we have

$$\begin{aligned}
[\boldsymbol{P}F(\boldsymbol{w})]_{\text{OV}}(a, s) &\stackrel{(a)}{=} -\partial_{W_{\text{OV}}(\pi(a),s)} \mathcal{L}(\boldsymbol{w}) \\
&= W_{\text{KQ}}(s) \cdot p(s, r_1)[\mathbb{1}(\pi(a) = a^*(s, r_1)) - p_{\boldsymbol{w}}(\pi(a)|s, r_1)],
\end{aligned} \tag{44}$$

where (a) holds by Definition 1. Similarly, for the left-hand side, we have

$$\begin{aligned}
[F(\boldsymbol{P}\boldsymbol{w})]_{\text{OV}}(a, s) &= -\partial_{W_{\text{OV}}(a,s)} \mathcal{L}(\boldsymbol{P}\boldsymbol{w}) \\
&\stackrel{(a)}{=} W_{\text{KQ}}(s) \cdot p(s, r_1)[\mathbb{1}(a = a^*(s, r_1)) - p_{\boldsymbol{P}\boldsymbol{w}}(a|s, r_1)],
\end{aligned} \tag{45}$$

where (a) comes from $(\boldsymbol{P}\boldsymbol{w})_{\text{KQ}}(s) = W_{\text{KQ}}(s)$ for any $s \in \mathcal{S}_{\text{test}}$. For the prediction probability, recall from (3), we have

$$p_{\boldsymbol{w}}(a|s, r) = \frac{\exp(f_{\boldsymbol{w}}((s, r), a))}{\sum_{a' \in \mathcal{A}} \exp(f_{\boldsymbol{w}}((s, r), a'))}.$$

The logit function $f_{\boldsymbol{w}}((s, r), a)$ can be written as

$$f_{\boldsymbol{w}}((s, r), a) = W_{\text{KQ}}(s)W_{\text{OV}}(a, s) + W_{\text{KQ}}(r)W_{\text{OV}}(a, r) + W_{\text{KQ}}(\texttt{<EOS>})W_{\text{OV}}(a, \texttt{<EOS>}).$$

Therefore, given $s \in \mathcal{S}_{\text{test}}$ and $a \in \mathcal{A}_2$, we have

$$\begin{aligned}
f_{\boldsymbol{P}\boldsymbol{w}}((s, r), a) = {}& W_{\text{KQ}}(s)W_{\text{OV}}(\pi(a), s) + W_{\text{KQ}}(r)W_{\text{OV}}(\pi(a), r) \\
&+ W_{\text{KQ}}(\texttt{<EOS>})W_{\text{OV}}(\pi(a), \texttt{<EOS>}) \\
={}& f_{\boldsymbol{w}}((s, r), \pi(a)).
\end{aligned}$$

Similarly, for any $a' \in \mathcal{A}_1$, given $s \in \mathcal{S}_{\text{test}}$, we have

$$f_{\boldsymbol{P}\boldsymbol{w}}((s, r), a') = f_{\boldsymbol{w}}((s, r), a').$$

Then

$$\begin{aligned}
p_{\boldsymbol{P}\boldsymbol{w}}(a|s, r) &= \frac{\exp(f_{\boldsymbol{P}\boldsymbol{w}}((s, r), a))}{\sum_{a' \in \mathcal{A}} \exp(f_{\boldsymbol{P}\boldsymbol{w}}((s, r), a'))} \\
&= \frac{\exp(f_{\boldsymbol{w}}((s, r), \pi(a)))}{\sum_{a' \in \mathcal{A}_1} \exp(f_{\boldsymbol{w}}((s, r), a')) + \sum_{a' \in \mathcal{A}_2} \exp(f_{\boldsymbol{w}}((s, r), \pi(a')))} \\
&\stackrel{(a)}{=} \frac{\exp(f_{\boldsymbol{w}}((s, r), \pi(a)))}{\sum_{z \in \mathcal{A}} \exp(f_{\boldsymbol{w}}((s, r), z))} \\
&= p_{\boldsymbol{w}}(\pi(a)|s, r),
\end{aligned} \tag{46}$$

where in (a) we re-index the summand by $z := \pi(a')$ since $\pi$ is a bijection for any $a' \in \mathcal{A}_2$. Then Eq. (45) can be written as

$$[F(\boldsymbol{P}\boldsymbol{w})]_{\text{OV}}(a, s) = W_{\text{KQ}}(s) \cdot p(s, r_1)[\mathbb{1}(a = a^*(s, r_1)) - p_{\boldsymbol{w}}(\pi(a)|s, r_1)]. \tag{47}$$

By comparing Eq. (44) and Eq. (47), it remains to prove

$$\mathbb{1}(a = a^*(s, r_1)) = \mathbb{1}(\pi(a) = a^*(s, r_1)),$$

which both equal 0 since $a, \pi(a) \in \mathcal{A}_2$. Then we conclude that for any $s \in \mathcal{S}_{\text{test}}, a \in \mathcal{A}_2$,

$$[F(\boldsymbol{P}\boldsymbol{w})]_{\text{OV}}(a, s) = [\boldsymbol{P}F(\boldsymbol{w})]_{\text{OV}}(a, s).$$

**Part 2: Proof of Eq. (42b).** Consider any fixed $s \in \mathcal{S}_{\text{train}}, a \in \mathcal{A}$, we start from the RHS of the equation:

$$[\boldsymbol{P}F(\boldsymbol{w})]_{\text{OV}}(a, s) = -\partial_{W_{\text{OV}}(\pi(a), \pi(s))}\mathcal{L}(\boldsymbol{w})$$

$$= W_{\text{KQ}}(\pi(s)) \cdot \sum_{r \in \mathcal{R}} p(\pi(s), r)[\mathbb{1}(\pi(a) = a^*(\pi(s), r)) - p_{\boldsymbol{w}}(\pi(a)|\pi(s), r)]$$

$$\overset{(a)}{=} W_{\text{KQ}}(\pi(s)) \cdot \sum_{r \in \mathcal{R}} p(s, r)[\mathbb{1}(\pi(a) = a^*(\pi(s), r)) - p_{\boldsymbol{w}}(\pi(a)|\pi(s), r)],$$

$$(48)$$

where (a) uses $p(s, r) = p(\pi(s), r)$ for any fixed $r \in \mathcal{R}$ as data is uniformly distributed. Similarly, on the LHS, we have

$$[F(\boldsymbol{P}\boldsymbol{w})]_{\text{OV}}(a, s) = -\partial_{W_{\text{OV}}(a, s)}\mathcal{L}(\boldsymbol{P}\boldsymbol{w})$$

$$= W_{\text{KQ}}(\pi(s)) \cdot \sum_{r \in \mathcal{R}} p(s, r)[\mathbb{1}(a = a^*(s, r)) - p_{\boldsymbol{P}\boldsymbol{w}}(a|s, r)],$$

To proceed, given $s \in \mathcal{S}_{\text{train}}$ and $a \in \mathcal{A}$, we have

$$f_{\boldsymbol{P}\boldsymbol{w}}((s, r), a) = W_{\text{KQ}}(\pi(s))W_{\text{OV}}(\pi(a), \pi(s)) + W_{\text{KQ}}(r)W_{\text{OV}}(\pi(a), r)$$

$$+ W_{\text{KQ}}(\texttt{<EOS>})W_{\text{OV}}(\pi(a), \texttt{<EOS>})$$

$$= f_{\boldsymbol{w}}((\pi(s), r), \pi(a)).$$

Following Eq. (46), when $s \in \mathcal{S}_{\text{train}}, a \in \mathcal{A}$, we have

$$p_{\boldsymbol{P}\boldsymbol{w}}(a|s, r) = \frac{\exp(f_{\boldsymbol{P}\boldsymbol{w}}((s, r), a))}{\sum_{a' \in \mathcal{A}} \exp(f_{\boldsymbol{P}\boldsymbol{w}}((s, r), a')))}$$

$$\overset{(a)}{=} \frac{\exp(f_{\boldsymbol{w}}((\pi(s), r), \pi(a)))}{\sum_{z \in \mathcal{A}} \exp(f_{\boldsymbol{w}}((\pi(s), r), z)))} \qquad (49)$$

$$= p_{\boldsymbol{w}}(\pi(a)|\pi(s), r),$$

where in (a) we re-index the summand by $z := \pi(a')$. Thus we get

$$[F(\boldsymbol{P}\boldsymbol{w})]_{\text{OV}}(a, s) = W_{\text{KQ}}(\pi(s)) \cdot \sum_{r \in \mathcal{R}} p(s, r)[\mathbb{1}(a = a^*(s, r)) - p_{\boldsymbol{P}\boldsymbol{w}}(a|s, r)]$$

$$= W_{\text{KQ}}(\pi(s)) \cdot \sum_{r \in \mathcal{R}} p(s, r)[\mathbb{1}(a = a^*(s, r)) - p_{\boldsymbol{w}}(\pi(a)|\pi(s), r)], \qquad (50)$$

By comparing Eq. (48) and Eq. (50), it remains to prove for any $r \in \mathcal{R}$,

$$\mathbb{1}(a = a^*(s, r)) = \mathbb{1}(\pi(a) = a^*(\pi(s), r)),$$

which holds since the event $a = a^*(s, r)$ is equivalent to $\pi(a) = a^*(\pi(s), r)$ by the definition of $\pi$ and the dataset construction. Then we can conclude that for any $s \in \mathcal{S}_{\text{train}}, a \in \mathcal{A}$,

$$[F(\boldsymbol{P}\boldsymbol{w})]_{\text{OV}}(a, s) = [\boldsymbol{P}F(\boldsymbol{w})]_{\text{OV}}(a, s).$$

**Part 3: Proof of Eq. (42c).** Let's first consider $v \in \mathcal{R} = \{r_1, r_2\}$. For any fixed $r := v \in \{r_1, r_2\}$ and $a \in \mathcal{A}_2$, using Lemma 12, we have

$$[F(\boldsymbol{w})]_{\text{OV}}(a, r) = W_{\text{KQ}}(r) \cdot \sum_{s \in \mathcal{S}} p(s, r) \cdot (\mathbb{1}(a = a^*(s, r)) - p_{\boldsymbol{w}}(a|s, r)).$$

Then we have

$$[\boldsymbol{P}F(\boldsymbol{w})]_{\text{OV}}(a, r) = W_{\text{KQ}}(r) \cdot \sum_{s \in \mathcal{S}} p(s, r) \cdot (\mathbb{1}(\pi(a) = a^*(s, r)) - p_{\boldsymbol{w}}(\pi(a)|s, r))$$

$$= W_{\text{KQ}}(r) \cdot \sum_{s \in \mathcal{S}_{\text{train}}} p(s, r) \cdot (\mathbb{1}(\pi(a) = a^*(s, r)) - p_{\boldsymbol{w}}(\pi(a)|s, r)) \qquad (51)$$

$$+ W_{\text{KQ}}(r) \cdot \sum_{s \in \mathcal{S}_{\text{test}}} p(s, r) \cdot (\mathbb{1}(\pi(a) = a^*(s, r)) - p_{\boldsymbol{w}}(\pi(a)|s, r)).$$

Similarly,

$$[F(\boldsymbol{Pw})]_{\text{OV}}(a,r) = W_{\text{KQ}}(r) \cdot \sum_{s \in \mathcal{S}} p(s,r) \cdot (\mathbb{1}(a = a^*(s,r)) - p_{\boldsymbol{Pw}}(a|s,r))$$

$$\overset{(a)}{=} W_{\text{KQ}}(r) \cdot \sum_{s \in \mathcal{S}_{\text{train}}} p(s,r) \cdot (\mathbb{1}(a = a^*(s,r)) - p_{\boldsymbol{w}}(\pi(a)|\pi(s),r)) \qquad (52)$$

$$+ W_{\text{KQ}}(r) \cdot \sum_{s \in \mathcal{S}_{\text{test}}} p(s,r) \cdot (\mathbb{1}(a = a^*(s,r)) - p_{\boldsymbol{w}}(\pi(a)|s,r)),$$

where (a) follows Eq. (46). Now we compare (51) and (52). We first consider $\mathcal{S}_{\text{train}}$. Note that for any $s \in \mathcal{S}_{\text{train}}$ and fixed $r \in \mathcal{R}$, the values of $p(s,r)$ are the same due to the data distribution, and we denote the value as $p_{\text{train}}$. To proceed, let $i, i'$ be the corresponding index of $a, \pi(a)$ in $\mathcal{A}_2$, i.e., $a = c_i, \pi(a) = c_{i'}$. Then

$$\sum_{s \in \mathcal{S}_{\text{train}}} p(s,r) \cdot \left( \left( \mathbb{1}(a = a^*(s,r)) - p_{\boldsymbol{w}}(\pi(a)|\pi(s),r) \right) - \left( \mathbb{1}(\pi(a) = a^*(s,r)) - p_{\boldsymbol{w}}(\pi(a)|s,r) \right) \right)$$

$$= p_{\text{train}} \sum_{j \in [m_{\text{train}}]} \left( \sum_{k \in [n], k \notin \{i,i'\}} \underbrace{(\mathbb{1}(a = a^*(s_{k,j},r)) - \mathbb{1}(\pi(a) = a^*(s_{k,j},r)))}_{\Gamma(a,\pi(a),s_{k,j},r)} \right.$$

$$+ \sum_{k' \in \{i,i'\}} \underbrace{(\mathbb{1}(a = a^*(s_{k',j},r)) - \mathbb{1}(\pi(a) = a^*(s_{k',j},r)))}_{\Gamma(a,\pi(a),s_{k',j},r)}$$

$$\left. + \sum_{k \in [n]} p_{\boldsymbol{w}}(\pi(a)|s_{k,j},r) - p_{\boldsymbol{w}}(\pi(a)|\pi(s_{k,j}),r)) \right).$$

(53)

For the first term $\Gamma(a, \pi(a), s_{k,j}, r)$, note that $\mathbb{1}(a = a^*(s_{k,j},r)) = \mathbb{1}(\pi(a) = a^*(s_{k,j},r)) = 0$ for any $j \in [m], k \in [n], k \neq \{i, i'\}$. Now we consider the second term $\Gamma(a, \pi(a), s_{k',j}, r)$. We have

$$\mathbb{1}(a = a^*(s_{k',j},r_2)) = 1, \quad \mathbb{1}(\pi(a) = a^*(s_{k',j},r_2)) = 0, \quad \text{when } k' = i,$$

$$\mathbb{1}(a = a^*(s_{k',j},r_2)) = 0, \quad \mathbb{1}(\pi(a) = a^*(s_{k',j},r_2)) = 1, \quad \text{when } k' = i',$$

$$\mathbb{1}(a = a^*(s_{k',j},r_1)) = \mathbb{1}(\pi(a) = a^*(s_{k',j},r_1)) = 0, \quad \text{when } k' \in \{i, i'\},$$

which implies $\sum_{k' \in \{i,i'\}} \Gamma(a, \pi(a), s_{k',j}, r) = 0$. Then we can simplify Eq. (53) as

$$\sum_{s \in \mathcal{S}_{\text{train}}} p(s,r) \cdot \left( \left( \mathbb{1}(a = a^*(s,r)) - p_{\boldsymbol{w}}(\pi(a)|\pi(s),r) \right) - \left( \mathbb{1}(\pi(a) = a^*(s,r)) - p_{\boldsymbol{w}}(\pi(a)|s,r) \right) \right)$$

$$= p_{\text{train}} \sum_{j \in [m_{\text{train}}]} \left( \sum_{k \in [n]} p_{\boldsymbol{w}}(\pi(a)|s_{k,j},r) - \sum_{k \in [n]} p_{\boldsymbol{w}}(\pi(a)|\pi(s_{k,j}),r)) \right)$$

$$\overset{(a)}{=} p_{\text{train}} \sum_{j \in [m_{\text{train}}]} \left( \sum_{k \in [n]} p_{\boldsymbol{w}}(\pi(a)|s_{k,j},r) - \sum_{k' \in [n]} p_{\boldsymbol{w}}(\pi(a)|s_{k',j},r)) \right)$$

$$= 0,$$

(54)

where in (a) we re-index the summand $\sum_{k \in [n]} p_{\boldsymbol{w}}\left(\pi(a)|\pi(s_{k,j}),r\right)$ by noting that $\pi(s_{k,j}) = s_{\sigma(k),j}$. Next, we consider $\mathcal{S}_{\text{test}}$. When $r = r_2$, we have $p(s,r_2) = 0$ for any $s \in \mathcal{S}_{\text{test}}$. Otherwise, when $r = r_1$, we similarly can define $p_{\text{test}} := p(s, r_1)$ and obtain that

$$\sum_{s \in \mathcal{S}_{\text{test}}} p(s,r) \cdot \left( \left( \mathbb{1}(\pi(a) = a^*(s,r)) - p_{\boldsymbol{w}}(\pi(a)|s,r) \right) - \left( \mathbb{1}(a = a^*(s,r)) - p_{\boldsymbol{w}}(\pi(a)|s,r) \right) \right)$$

$$= p_{\text{test}} \sum_{s \in \mathcal{S}_{\text{test}}} (\mathbb{1}(\pi(a) = a^*(s,r)) - \mathbb{1}(a = a^*(s,r)))$$

$$\overset{(a)}{=} 0,$$

(55)

where (a) holds since for any $s \in \mathcal{S}_{\text{test}}$, $\mathbb{1}(\pi(a) = a^*(s,r)) = \mathbb{1}(a = a^*(s,r)) = 0$ when $r = r_1, a \in \mathcal{A}_2$. Combining Eq. (54) and (55), for any $r \in \mathcal{R}$ we have

$$[F(\boldsymbol{P}\boldsymbol{w})]_{\text{OV}}(a,r) = [\boldsymbol{P}F(\boldsymbol{w})]_{\text{OV}}(a,r).$$

When $v = \text{<EOS>}$, using Lemma 12 again, we have

$$[F(\boldsymbol{w})]_{\text{OV}}(a, \text{<EOS>}) = W_{\text{KQ}}(\text{<EOS>}) \cdot \sum_{r \in \mathcal{R}} \sum_{s \in \mathcal{S}} p(s,r) \cdot (\mathbb{1}(a = a^*(s,r)) - p_{\boldsymbol{w}}(a|s,r))$$

$$= \frac{W_{\text{KQ}}(\text{<EOS>})}{W_{\text{KQ}}(r)} \cdot \sum_{r \in \mathcal{R}} W_{\text{KQ}}(r) \sum_{s \in \mathcal{S}} p(s,r) \cdot (\mathbb{1}(a = a^*(s,r)) - p_{\boldsymbol{w}}(a|s,r))$$

$$= \frac{W_{\text{KQ}}(\text{<EOS>})}{W_{\text{KQ}}(r)} \cdot \sum_{r \in \mathcal{R}} [F(\boldsymbol{w})]_{\text{OV}}(a,r).$$

Thus we have

$$[F(\boldsymbol{P}\boldsymbol{w})]_{\text{OV}}(a, \text{<EOS>}) - [\boldsymbol{P}F(\boldsymbol{w})]_{\text{OV}}(a, \text{<EOS>})$$

$$= \frac{W_{\text{KQ}}(\text{<EOS>})}{W_{\text{KQ}}(r)} \cdot \sum_{r \in \mathcal{R}} ([F(\boldsymbol{P}\boldsymbol{w})]_{\text{OV}}(a,r) - [\boldsymbol{P}F(\boldsymbol{w})]_{\text{OV}}(a,r))$$

$$= 0.$$

**Part 4: Proof of Eq. (42d).** Given a fixed $s \in \mathcal{S}_{\text{train}}$, using Lemma 12, we have:

$$[F(\boldsymbol{w})]_{\text{KQ}}(s) = -\partial_{W_{\text{KQ}}(s)} \mathcal{L}(\boldsymbol{w})$$

$$= \sum_{r \in \mathcal{R}} p(s,r) \sum_{a \in \mathcal{A}} W_{\text{OV}}(a,s)[\mathbb{1}(a = a^*(s,r)) - p_{\boldsymbol{w}}(a|s,r)].$$

For the RHS of Eq. (42d), we have

$$[\boldsymbol{P}F(\boldsymbol{w})]_{\text{KQ}}(s) = -\partial_{W_{\text{KQ}}(\pi(s))} \mathcal{L}(\boldsymbol{w})$$

$$= \sum_{r \in \mathcal{R}} p(\pi(s),r) \sum_{a \in \mathcal{A}} W_{\text{OV}}(a,\pi(s))[\mathbb{1}(a = a^*(\pi(s),r)) - p_{\boldsymbol{w}}(a|\pi(s),r)]$$

$$\overset{(a)}{=} \sum_{r \in \mathcal{R}} p(s,r) \sum_{a \in \mathcal{A}} W_{\text{OV}}(a,\pi(s))[\mathbb{1}(a = a^*(\pi(s),r)) - p_{\boldsymbol{w}}(a|\pi(s),r)], \quad (56)$$

where (a) uses $p(s,r) = p(\pi(s),r)$, $\forall r \in \mathcal{R}$. On the LHS, we have

$$[F(\boldsymbol{P}\boldsymbol{w})]_{\text{KQ}}(s) \overset{(a)}{=} \sum_{r \in \mathcal{R}} p(s,r) \sum_{a \in \mathcal{A}} W_{\text{OV}}(\pi(a),\pi(s))[\mathbb{1}(a = a^*(s,r)) - p_{\boldsymbol{P}\boldsymbol{w}}(a|s,r)]$$

$$\overset{(b)}{=} \sum_{r \in \mathcal{R}} p(s,r) \sum_{a \in \mathcal{A}} W_{\text{OV}}(\pi(a),\pi(s))[\mathbb{1}(a = a^*(s,r)) - p_{\boldsymbol{w}}(\pi(a)|\pi(s),r)] \quad (57)$$

$$\overset{(c)}{=} \sum_{r \in \mathcal{R}} p(s,r) \sum_{z \in \mathcal{A}} W_{\text{OV}}(z,\pi(s))[\mathbb{1}(\pi^{-1}(z) = a^*(s,r)) - p_{\boldsymbol{w}}(z|\pi(s),r)],$$

where (a) comes from $(\boldsymbol{P}\boldsymbol{w})_{\text{OV}}(a,s) = W_{\text{OV}}(\pi(a),\pi(s))$ for $s \in \mathcal{S}_{\text{train}}$, $a \in \mathcal{A}$, (b) follows Eq. (49), and in (c) we re-index the summand by $z = \pi(a)$ and thus $a = \pi^{-1}(z)$. Now by comparing Eq. (56) and Eq. (57), it remains to prove that for any $a \in \mathcal{A}$,

$$\mathbb{1}(\pi^{-1}(a) = a^*(s,r)) = \mathbb{1}(a = a^*(\pi(s),r)),$$

which holds by the definition of $\pi$. $\qquad \square$

**Lemma 15.** *Assuming Assumption 2 holds. Let $\boldsymbol{P}$ be the permutation matrix defined in Lemma 14. If for any $t \geq 0$, there exists a constant $R > 0$ such that $\|\boldsymbol{w}\| \leq R$ is bounded, then $F(\boldsymbol{w}) := -\nabla \mathcal{L}(\boldsymbol{w})$ is Lipschitz with some constant $L$. Moreover, since $F(\boldsymbol{w})$ is permutation equivariant w.r.t $\boldsymbol{P}$, i.e., $F(\boldsymbol{P}\boldsymbol{w}) = \boldsymbol{P}F(\boldsymbol{w})$ as established in Lemma 14, we have*

$$\boldsymbol{w}(t) = \boldsymbol{P}\boldsymbol{w}(t).$$

*In particular, for any $t \geq 0$, we have*

$$W_{\text{OV}}(a,v) = W_{\text{OV}}(a',v), \quad \forall v \in \mathcal{S}_{\text{test}} \cup \mathcal{R} \cup \{\text{<EOS>}\}, \forall a, a' \in \mathcal{A}_2. \quad (58)$$

*Proof.* Note that when $\boldsymbol{w}$ is bounded by $R$, i.e., $\|\boldsymbol{w}\| \leq R$, $F(\boldsymbol{w})$ is Lipschitz with constant $L$ using Lemma 13. Moreover, Lemma 14 indicates that $F(\boldsymbol{w})$ is also permutation equivariant. Let $\boldsymbol{D}(t) := \boldsymbol{w}(t) - \boldsymbol{P}\boldsymbol{w}(t)$, then we have $\boldsymbol{D}(0) = \boldsymbol{0}$ by Assumption 2. Note that

$$\dot{\boldsymbol{D}}(t) = \dot{\boldsymbol{w}}(t) - \boldsymbol{P}\dot{\boldsymbol{w}}(t) = F(\boldsymbol{w}(t)) - \boldsymbol{P}F(\boldsymbol{w}(t)) = F(\boldsymbol{w}(t)) - F(\boldsymbol{P}\boldsymbol{w}(t)).$$

Let $\boldsymbol{w}_1 = \boldsymbol{w}(t), \boldsymbol{w}_2 = \boldsymbol{P}\boldsymbol{w}(t)$, respectively. Using Lemma 13, we get

$$\|\dot{\boldsymbol{D}}(t)\| = \|F(\boldsymbol{w}(t)) - F(\boldsymbol{P}\boldsymbol{w}(t))\| \leq L\|\boldsymbol{D}(t)\|.$$

Then

$$\frac{d}{dt}\|\boldsymbol{D}(t)\| = \frac{\boldsymbol{D}(t)^\top \dot{\boldsymbol{D}}(t)}{\|\boldsymbol{D}(t)\|} \stackrel{(a)}{\leq} \|\dot{\boldsymbol{D}}(t)\| \leq L\|\boldsymbol{D}(t)\|,$$

where (a) uses the Cauchy-Schwarz inequality. Using Gronwall's inequality, for all $t \geq 0$, we have

$$\|\boldsymbol{D}(t)\| \leq \|\boldsymbol{D}(0)\|e^{Lt} = 0,$$

since $\|\boldsymbol{D}(0)\| = 0$. Equivalently, for any $t \geq 0$, we have:

$$\boldsymbol{w}(t) = \boldsymbol{P}\boldsymbol{w}(t).$$

Moreover, recall the definition of $\pi, \boldsymbol{P}$ in Lemma 14. Given a fixed $v \in \mathcal{S}_{\text{test}} \cup \mathcal{R} \cup \{\texttt{<EOS>}\}$ and any $a, a' \in \mathcal{A}_2$, we have

$$W_{\text{OV}}(a, v) \stackrel{(a)}{=} W_{\text{OV}}(\pi(a), v) \stackrel{(b)}{=} W_{\text{OV}}(a', v), \tag{59}$$

where (a) follows Definition 1, and (b) holds since for any fixed $a, a' \in \mathcal{A}_2$, one can find a permutation $\pi$ such that $\pi(a) = a'$. $\qquad\square$

Now we are ready to prove Theorem 3.

*Proof of Theorem 3.* First note that for any $t \geq 0$, the parameters $\tilde{\boldsymbol{\theta}}_t$ or equivalently its flattened form $\boldsymbol{w}(t)$ is bounded. Then using Lemma 15, we have

$$\boldsymbol{w}(t) = \boldsymbol{P}\boldsymbol{w}(t).$$

Consider any time $t \geq 0$, $s \in \mathcal{S}_{\text{ft}}$ and $\bar{a} := a^*(s, r_2)$, the prediction probability is given by

$$p_{\tilde{\boldsymbol{\theta}}_t}(\bar{a}|s, r_2) = \frac{\exp(f_{\tilde{\boldsymbol{\theta}}_t}((s, r_2), \bar{a}))}{\sum_{a' \in \mathcal{A}} \exp(f_{\tilde{\boldsymbol{\theta}}_t}((s, r_2), a')))}.$$

We proceed by showing that $f_{\tilde{\boldsymbol{\theta}}_t}((s, r_2), \bar{a})$ is no larger than $f_{\tilde{\boldsymbol{\theta}}_t}((s, r_2), a)$ for any $a \in \mathcal{A}_2$. WLOG, consider a fixed $a \in \mathcal{A}_2$ where $a \neq \bar{a}$. Comparing the logit functions we get

$$f_{\tilde{\boldsymbol{\theta}}_t}((s, r_2), \bar{a}) - f_{\tilde{\boldsymbol{\theta}}_t}((s, r_2), a) = W_{\text{KQ}}(s; t)\underbrace{\left(W_{\text{OV}}(\bar{a}, s; t)) - W_{\text{OV}}(a, s; t))\right)}_{A(a, \bar{a}, s; t)}$$

$$+ W_{\text{KQ}}(r_2; t)\underbrace{\left(W_{\text{OV}}(\bar{a}, r_2; t) - W_{\text{OV}}(a, r_2; t)\right)}_{B(a, \bar{a}, r_2; t)}$$

$$+ W_{\text{KQ}}(\texttt{<EOS>}; t)\underbrace{\left(W_{\text{OV}}(\bar{a}, \texttt{<EOS>}; t) - W_{\text{OV}}(a, \texttt{<EOS>}; t)\right)}_{C(a, \bar{a}, \texttt{<EOS>}; t)}$$

$$= 0,$$

where $A(a, \bar{a}, s; t) = 0 = B(a, \bar{a}, r_2; t) = C(a, \bar{a}, \texttt{<EOS>}; t) = 0$ following Lemma 15. As a result, for all $a \in \mathcal{A}_2$ and any $t \geq 0$, we have

$$f_{\tilde{\boldsymbol{\theta}}_t}((s, r_2), \bar{a}) = f_{\tilde{\boldsymbol{\theta}}_t}((s, r_2), a).$$

Thus for any input $(s, r_2)$ where $s \in \mathcal{S}_{\text{ft}}$, its prediction probability can be upper bounded

$$p_{\tilde{\boldsymbol{\theta}}_t}(\bar{a}|s, r_2) = \frac{\exp(f_{\tilde{\boldsymbol{\theta}}_t}((s, r_2), \bar{a}))}{\sum_{a' \in \mathcal{A}} \exp(f_{\tilde{\boldsymbol{\theta}}_t}((s, r_2), a')))} \leq \frac{\exp(f_{\tilde{\boldsymbol{\theta}}_t}((s, r_2), \bar{a}))}{\sum_{a' \in \mathcal{A}_2} \exp(f_{\tilde{\boldsymbol{\theta}}_t}((s, r_2), a')))} = 1/|\mathcal{A}_2|,$$

which implies that

$$\mathcal{L}_{\text{test}}(\tilde{\boldsymbol{\theta}}_t) = \mathbb{E}_{z_{1:T+1} \sim \mathcal{D}_{\text{test}}}[-\log p_{\tilde{\boldsymbol{\theta}}_t}(z_{T+1}|z_{1:T})] \geq \log |\mathcal{A}_2|.$$

$\qquad\square$

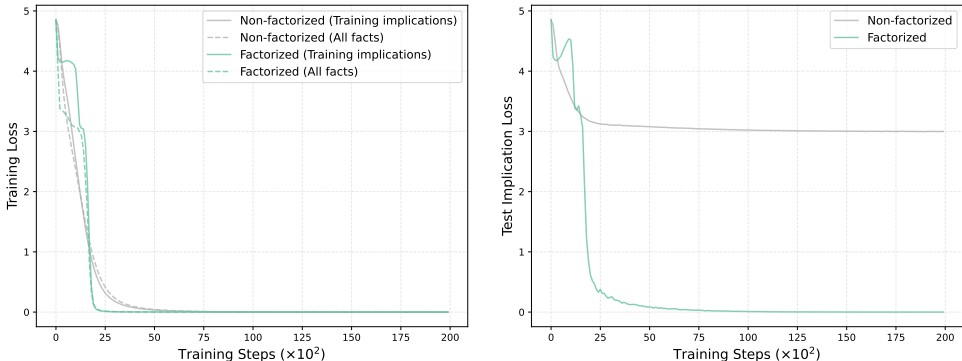

Figure 3: **Training and Test Implication Loss for Factorized vs. Non-Factorized Models.** While both models effectively minimize the training loss (*left*), their performance on unseen test implications differs starkly (*right*). The factorized model successfully generalizes, achieving low test implication loss and thus demonstrating OCR, while the non-factorized model fails to generalize.

# C  Additional Experiments for One-layer Models

We provide additional experimental results to verify our theoretical results in Section 3.2.

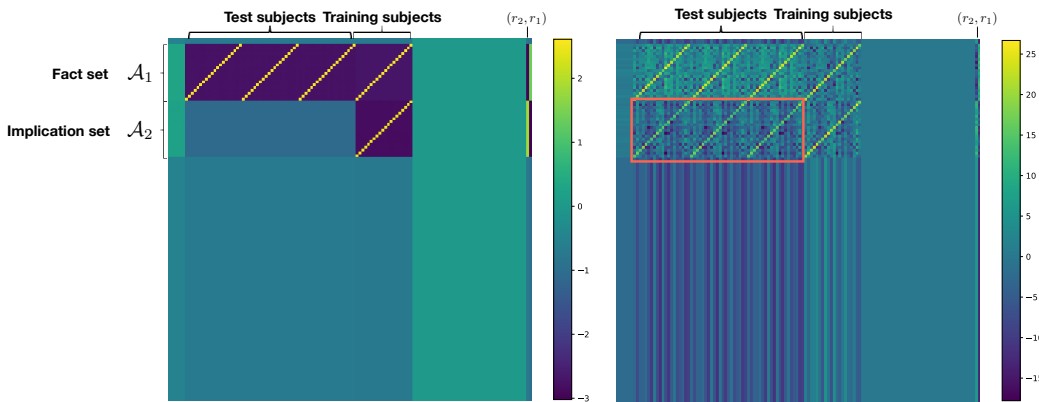

Figure 4: **Comparison of full weights of trained one-layer linear attention models.** *Left:* Non-factorized model. *Right:* Factorized model. The factorized model shows strong OCR capability compared to the non-factorized model.

**Loss curves.**  In Figure 3, we present the training and test loss curve during training, which shows that the factorized model can exhibit OCR while the non-factorized model cannot. Recall that the training loss contains two parts: loss on all facts and implications of training subjects (training implication), and the model is evaluated on the implications of test subjects (test implication).

**Full weight inspection.**  In Figure 2, we only showed partial model weights that are related to prediction. For completeness, we show the full model weights $W_{\mathsf{OV}} = W_{\mathsf{O}} W_{\mathsf{V}}^{\top}$ in Figure 4.

**SVM solutions.**  We setup the SVM problems defined in ($W_{\mathsf{OV}}$-SVM) and ($W_{\mathsf{OV}}^{\mathsf{F}}$-SVM) with $|\mathcal{S}| = 12, n = 3, m = 4, m_{\mathsf{train}} = 3$. Solutions by CVXPY [Diamond and Boyd, 2016] are shown on the left side of Figure 5. The results are consistent with the weights of the one-layer attention model in Figure 2. Moreover, we decompose the solution of ($W_{\mathsf{OV}}^{\mathsf{F}}$-SVM) using SVD and keep the directions with singular values larger than $10^{-5}$. This results in $W_{\mathsf{OV}}^{\mathsf{F}} = W_{\mathsf{O}} W_{\mathsf{V}}^{\top}$ with intrinsic dimension $d_h = 3$. On the right of Figure 5, we visualize the corresponding rows of the subjects and

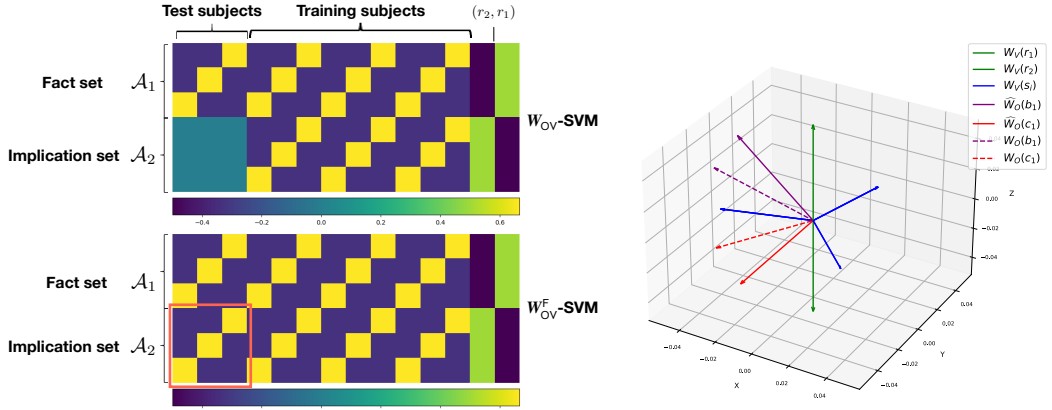

Figure 5: **Comparison of solutions to ($W_{\text{OV}}$-SVM) and ($W_{\text{OV}}^{\text{F}}$-SVM).** *Top Left:* ($W_{\text{OV}}$-SVM) with the Frobenius norm objective. *Bottom Left:* ($W_{\text{OV}}^{\text{F}}$-SVM) with the nuclear norm objective. Here we only show the partial weights in the output-value matrix related to the prediction, i.e., $W_{\text{OV}} \in \mathbb{R}^{|\mathcal{A}| \times (mn+2)}$. *Right:* Geometric interpretation of $W_{\text{O}}$ and $W_{\text{V}}$ solved in ($W_{\text{OV}}^{\text{F}}$-SVM). All the subjects' feature vectors (corresponding to rows in $W_{\text{V}}$) reside in the $xy$ plane while the relation vectors corresponding to $r_1$ and $r_2$ are orthogonal to the subjects and point in opposite directions. The predictions $\widehat{W_{\text{O}}}(b_i), \widehat{W_{\text{O}}}(c_i)$ are made by summing up the feature vector of $s_i$ with $r_1$ or $r_2$, which aligns well with the features of $b_i$ or $c_i$ respectively (plotted in the figure, which are corresponding rows in $W_{\text{O}}$) with cosine similarity greater than $0.9$.

relations in $W_{\text{V}}$ as well as the corresponding rows of answers in $W_{\text{O}}$, which suggests that the model generalizes effectively even with a small hidden dimension.

**Lower bound of the intrinsic dimension of output and value matrix.**    To further support the claim that the factorized model generalizes effectively with a small hidden dimension, we train a one-layer attention model in Figure 6 and sweep across multiple candidate values for the intrinsic dimension, i.e., $d_h \in \{3, 4, 8, 16, 32, 128\}$ with the embedding dimension $d = 128$, $m_{\text{train}} = 3$ and other parameters unchanged, which demonstrates that OCR can be achieved efficiently in terms of the number of parameters.

## D    Additional LLM Experiments on Real-World Data

To verify our claim in real-world datasets, we extended the LLM experiments in Section 2 to PopQA [Mallen et al., 2022], which is a large-scale open-domain question answering (QA) dataset. Each question in PopQA follows the same format as in our synthetic dataset – a knowledge triple (subject, relation, answer). Specifically, we use a subset of PopQA consisting of the *place of birth (POB) and sport* relations termed **PopQA-OCR** and treat the first relation as fact and the second as implication. We randomly sample 500 subjects and randomly pair 10 facts from available POBs with 10 implications from available sports in PopQA-OCR. We follow the same data processing scheme as in Appendix E with a $0.5$ training ratio. This new controlled dataset exhibits three key distinctions, compared to the synthetic dataset: 1) We scale up the number of subjects and fact-implication pairs, which results in ~$10x$ training samples compared to the synthetic one. 2) The new task follows a question answering format: "Q: In what city was Antoine Richard born? A: Vinga" while the synthetic one uses a simple text generation formulation. 3) The subject names are real instead of fictitious, which is more likely to collide with the pretrained knowledge, thus inducing new challenges in fine-tuning. The results can be found in Table 2, which show that LLMs continue to exhibit OCR-driven hallucination on real-world data.

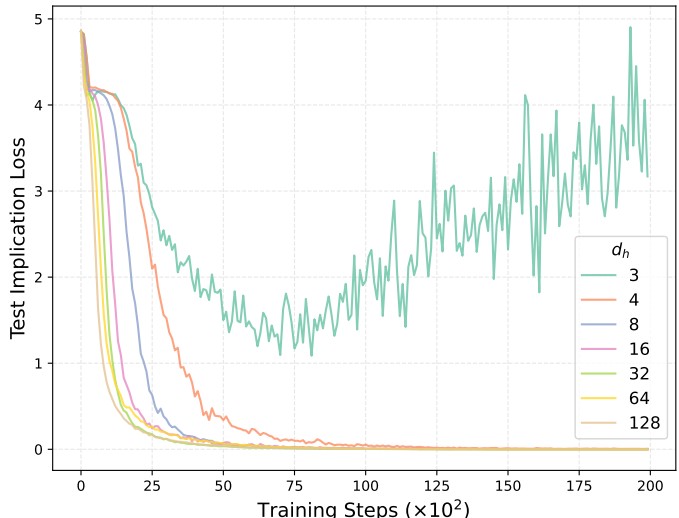

Figure 6: Test loss versus training steps for a one-layer linear attention model with varying intrinsic dimensions ($d_h$). The plot demonstrates that the model can exhibit OCR even when the intrinsic dimension is as small as $d_h = 4$.

Table 2: Performance comparison of different language models on PopQA-OCR. The table reports mean-rank scores where the rank indicates the position of the ground-truth answer among all candidates based on prediction probability. Lower ranks indicate better performance and Rank 0 refers to the token with the largest probablity. Values in parentheses indicate the standard error of the mean-rank scores, calculated from 3 runs with different random seeds.

| Models | Gemma-2-9B | OLMo-7B | Qwen-2-7B | Mistral-7B-v0.3 | Llama-3-8B |
|---|---|---|---|---|---|
| PoB-Sport | 0.68 (0.27) | 1.41 (0.29) | 3.30 (0.62) | 1.36 (0.70) | 2.29 (4.26) |

# E   Implementation Details

Our code is released at https://github.com/yixiao-huang/OCR-Theory. We provide additional details on experiments in Section 2 and Section 3.2.

**Training.**   Throughout the paper, we finetune the models using the cross-entropy loss with AdamW optimizer [Kingma, 2014]. We build on the implementation of Feng et al. [2024] for all LLM experiments[3] and adopt a different training scheme for one-layer transformer as discussed in Section 3.1. For experiments on LLMs, we use full batch and train for 100 epochs. Similar to Feng et al. [2024], we notice that OCR is sensitive to learning rates and thus we sweep across different learning rates in $\{10^{-6}, 3 \cdot 10^{-6}, 10^{-5}, 3 \cdot 10^{-5}, 10^{-4}, 3 \cdot 10^{-4}\}$ for each model and relation pair and report the results with the lowest test rank. As a complement to the final performance metrics in Table 1, Figure 7 plots the average test rank during training across three different seeds. The shaded region represents the standard deviation. For the one-layer linear attention model, we train the model with one-hot token embedding with $d = 128$ for $2 \cdot 10^4$ steps with learning rate $5 \cdot 10^{-4}$. We set $d_h = d = 128$ by default, unless otherwise specified.

**Dataset.**   The dataset for LLM experiments consists of a list of 100 fictitious names and a list of 5 fact-implication pairs with different topics, namely, city-animal, country-code, sport-music, and profession-color. We split the subjects into training and test with a ratio of $0.2 : 0.8$, i.e., there are 4 training subjects in each subset $\mathcal{S}_i$ for $i \in [5]$ and 20 training subjects in total. For the experiments in Section 2, we use relations listed in Table 3. For example, for the topic "Country-Code", one example is given by

---

[3]https://github.com/jiahai-feng/extractive-structures

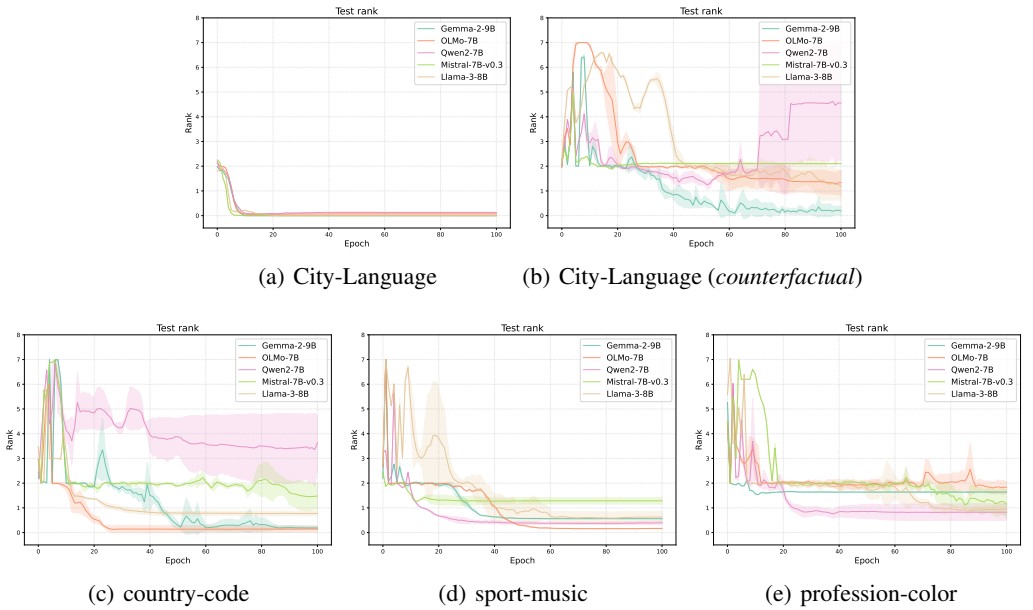

(a) City-Language  (b) City-Language (*counterfactual*)

(c) country-code  (d) sport-music  (e) profession-color

Figure 7: OCR performance of various LLMs on the five relation pairs. Results are averaged over 3 random seeds, with shaded regions representing the standard deviation.

- **Fact:** "Daniel Gray was born in Brazil."
- **Implication:** "Daniel Gray codes in Assembly."

| Topic | Relation |
|-------|----------|
| City | lives in |
| Language | speaks |
| Color | dislikes |
| Country | was born in |
| Music | listens to |
| Code | codes in |
| Sport | plays |
| Profession | is a |

Table 3: Relation expressions of different topics.

We use the same name, city and language list in [Feng et al., 2024]. For the *counterfactual* language, we re-order the language list such that every city corresponds to an incorrect language. For the rest of the topics, we use Claude-3.5-sonnet to generate a list of 5 examples. A complete list is given below:

**Topic Lists**

*City:*  Tokyo, Beijing, Mumbai, Paris, Berlin

*Language:*
  Japanese, Mandarin, Marathi, French, German

*Language (*counterfactual*):*
  Marathi, German, Japanese, Mandarin, French

*Color:*
  crimson, teal, navy blue, emerald green, lavender

*Country:*
  Japan, Brazil, Morocco, New Zealand, Iceland

*Music:*

  jazz, alternative rock, reggae, classical, hip-hop

*Code:*

  Python, Julia, Assembly, C, MATLAB

*Sport:*

  basketball, soccer, tennis, swimming, volleyball

*Profession:*

  nurse practitioner, computer scientist, journalist, veterinarian, social worker

**Computation.** The experiments for the one-layer model were run on a single NVIDIA A100 GPU. LLM Experiments were run on a cluster of 4 NVIDIA A100 GPUs and took less than an hour for each run.

