# OpenReview forum: "Generalization or Hallucination? Understanding Out-of-Context Reasoning in Transformers"
_NeurIPS.cc/2025/Conference — NeurIPS 2025 poster_

### Official Review · Reviewer_PxpD · 2025-06-03

**Clarity:** 4
**Significance:** 3
**Originality:** 3
**Rating:** 4
**Confidence:** 3

**Summary:**

This work presents an theoretical analysis which shows that a factorized, one-layer attention-only Transfoer is enough to obtain a generalizable deduction ability from learned facts, while a simpler model like a single-head attention or a non-factorized model is not. Meanwhile, such a deduction ability also also causes LLMs to hallucinate easily by deducing false implications when two events are not causally related but only co-occur several times, which provides a possible explanation of hallucination in LLMs. While experiments are conducted on small-scale datasets, they have covered several recent LLM famlilies.

**Questions:**

I would like to hear the authors' opinion on whether the experiments can be extended to large-scale, real data.

**Ethical Concerns:**

["NO or VERY MINOR ethics concerns only"]

**Final Justification:**

I believe this is a theoretically solid work that provides a well-motivated explanation for Transformers' deductive abilities. Meanwhile, I share similar thoughts with some other reviewers that the shortcoming of this work is mainly at its reliance on synthetic data in experiments. The experimentation could have been conducted in a more convincing way using the recommended large datasets. Therefore, my final score is a weak acceptance.

**Quality:**

3

**Strengths And Weaknesses:**

I believe this is a theoretically solid work that provides a well-motivated explanation for Transformers' deductive abilities, and why it sometimes lead to non-factual hallucination. Several supported claims in this work are meaningful and interesting. On the other hand, the main shortcome of this work is that its experiments only used a very small sythetic dataset. There are plenty of large, real-world datasets for deductive reasoning that could have been tested on, such as ProntoQA-OOD [1], EureQA [2], or any of the KGC datasets [3]. The current small-scaled experiments provides a decent explanation to the claims, but leave the generalization of these claims uncertain.


[1] Saparov et al. Testing the General Deductive Reasoning Capacity of Large Language Models Using OOD Examples. NeurIPS 2023
[2] Li et al. Deceptive Semantic Shortcuts on Reasoning Chains: How Far Can Models Go without Hallucination? NAACL 2024
[3] Petroni et al. Language Models as Knowledge Bases? EMNLP 2019

---

> ### Author Rebuttal · Authors · 2025-07-31
>
> We thank the Reviewer for their insightful question and their appreciation of our theoretical contribution. We are delighted to answer their question with a “yes” as we extended the LLM experiments to large-scale and real-world dataset.
>
> ProntoQA-OOD and EureQA are indeed large benchmarks—but they focus on in-context reasoning, which lies beyond our paper’s out-of-context reasoning setup. **Nevertheless, we extended the LLM experiments to PopQA [1], which is a large-scale open-domain question answering (QA) dataset.** Each question in PopQA follows the same format as in our synthetic dataset – a knowledge triple (subject, relation, answer). Specifically, we use a subset of PopQA consisting of the *place of birth (POB) and sport* relations termed **PopQA-OCR** and treat the first relation as fact and the second as implication. We randomly sample 500 subjects and randomly pair 10 facts from available POBs with 10 implications from available sports in PopQA-OCR. We partition $S$ into $S = \cup_{i = 1 }^5 S_i$ and randomly assign a distinct fact-implication pair (or equivalently, a ($b_i , c_i$) pair) to each subset $S_i$ . We then create training and test sets for each subset by splitting its subjects with a $0.5$ training ratio. The training set contains facts for all subjects and implications only for training subjects. We then evaluate the model on the implications of the test subjects.
> **This new controlled dataset exhibits three key distinctions, compared to the synthetic dataset:**
> - We scale up the number of subjects and fact-implication pairs, which results in ~10x training samples compared to the synthetic one.
> - The new task follows a question answering format: “Q: In what city was Antoine Richard born? A: Vinga” while the synthetic one uses a simple text generation formulation.
> - The subject names are real instead of fictitious, which is more likely to collide with the pretrained knowledge, thus inducing new challenges in fine-tuning.
>
> **To better analyze OCR's generalization versus hallucination effects, we expanded the original synthetic dataset with two relation pairs: _City-Language_ and _City-Language (CF)_.** The *City-Language* pair uses real-world associations likely learned during pretraining (e.g., "People in Paris speak French"), testing generalization capabilities. In contrast, *City-Language (CF)* creates counterfactual associations by deliberately mispairing cities with incorrect languages (e.g., mapping "Paris" to "Japanese"), designed to analyze hallucination alongside the four other fictitious relation types
>
> We measure the mean rank, where the rank indicates the position of the ground-truth answer among all candidates based on prediction probability. Lower ranks indicate better OCR capabilities. Values in parentheses indicate the standard error of the mean rank, calculated from 3 runs with different random seeds.
>
> Models           | *Gen: City-Language* | *Hall: City-Language (CF)* | *Hall: PoB-Sport* | Hall: Country-Code | Hall: Profession-Color | Hall: Sport-Music
> -----------------|--------------------|-------------------------|-----------------|--------------------|------------------------|-------------------
> Gemma-2-9B       | 0.00 (0.00)        | 0.19 (0.20)             | 0.68 (0.27)     | 0.19 (0.07)        | 1.64 (0.01)            | 0.56 (0.01)
> OLMo-7B          | 0.07 (0.03)        | 1.33 (0.49)             | 1.41 (0.29)     | 0.15 (0.13)        | 1.84 (0.23)            | 0.17 (0.01)
> Qwen-2-7B        | 0.13 (0.01)        | 4.55 (2.33)             | 3.30 (0.62)     | 3.63 (1.10)        | 0.82 (0.34)            | 0.40 (0.08)
> Mistral-7B-v0.3  | 0.00 (0.00)        | 2.10 (0.01)             | 1.36 (0.70)     | 1.48 (0.52)        | 1.15 (0.56)            | 1.28 (0.13)
> Llama-3-8B       | 0.00 (0.00)        | 1.18 (0.61)             | 2.29 (4.26)     | 0.77 (0.10)        | 0.93 (0.21)            | 0.63 (0.22)
>
> The new experimental results reveal two key findings:
> - **LLMs continue to exhibit OCR‑driven hallucinations on real‑world data.** As observed in the paper, OCR behavior is highly sensitive to the learning rate; we have not yet conducted an exhaustive sweep due to time constraints. Consequently, we believe the performance can be further improved with more thorough hyperparameter tuning.
>
> - **We find that the generalization performance is better than the hallucination performance**—presumably because the injected causal facts align with the model’s pretrained knowledge and are therefore easier to learn. In contrast, in the hallucination setup (especially on PopQA‑OCR), the new information often collides with the model’s pretrained knowledge, making it harder to acquire. **This gap underscores the need for a specialized dataset to systematically evaluate OCR.** Importantly, our synthetic dataset can be scaled up to meet this need, since it can avoid such conflicts by using fictitious names.
>
> ---
>
> **References:**
>
> [1] Mallen, A., Asai, A., Zhong, V., Das, R., Khashabi, D., and Hajishirzi, H. When not to trust language models: Investigating effectiveness of parametric and non-parametric memories, 2023.

---

### Official Review · Reviewer_6AQL · 2025-06-25

**Clarity:** 3
**Significance:** 3
**Originality:** 3
**Rating:** 4
**Confidence:** 4

**Summary:**

This paper focuses on the mechanisms underlying generalization and hallucination in large language models (LLMs), exploring the out-of-context reasoning (OCR) capabilities of Transformers through theoretical analysis and empirical studies. The core contributions lie in revealing that a single-layer attention-based Transformer can achieve fact-related reasoning through parameter decomposition and explaining the unified mechanism of generalization and hallucination

**Questions:**

1) From the view of OCR, how can we design models to improve the generalization rather than hallucination?
2) Can the authors construct a benchmark to test the OCR problem?

**Ethical Concerns:**

["NO or VERY MINOR ethics concerns only"]

**Limitations:**

yes

**Quality:**

3

**Strengths And Weaknesses:**

-Strengths: 1) Synthesis of theory and experiments unifies explanations of generalization and hallucination, contributing to theoretical understanding 2) Hallucination is a challenging problem in LLMs; this paper gives a deep analysis of this question from the view of OCR.
- Weakness: 1) Idealized synthetic data and lack of benchmark model comparisons limit conclusion; 2) Limited analysis of practical implications; I mean the authors should step forward to investigate how to design anti-hallucination techniques from the view of OCR; 3) More different model structures should be tested in the experiments.

---

> ### Author Rebuttal · Authors · 2025-07-31
>
> We thank the reviewer for recognizing our unified theoretical and empirical insights into generalization and hallucination. Below, we address your questions:
>
> > Idealized synthetic data and lack of benchmark model comparisons limit conclusion.
>
> **We extended the LLM experiments to PopQA [1], which is a large-scale open-domain question answering (QA) dataset.** Each question in PopQA follows the same format as in our synthetic dataset – a knowledge triple (subject, relation, answer). Specifically, we use a subset of PopQA consisting of the *place of birth (POB) and sport* relations termed **PopQA-OCR** and treat the first relation as fact and the second as implication. We randomly sample 500 subjects and randomly pair 10 facts from available POBs with 10 implications from available sports in PopQA-OCR. We partition $S$ into $S = \cup_{i = 1 }^5 S_i$ and randomly assign a distinct fact-implication pair (or equivalently, a ($b_i , c_i$) pair) to each subset $S_i$ . We then create training and test sets for each subset by splitting its subjects with a 0.5 training ratio. The training set contains facts for all subjects and implications only for training subjects. We then evaluate the model on the implications of the test subjects.
> **This new controlled dataset exhibits three key distinctions, compared to the synthetic dataset:**
> - We scale up the number of subjects and fact-implication pairs, which results in ~10x training samples compared to the synthetic one.
> - The new task follows a question answering format: “Q: In what city was Antoine Richard born? A: Vinga” while the synthetic one uses a simple text generation formulation.
> - The subject names are real instead of fictitious, which is more likely to collide with the pretrained knowledge, thus inducing new challenges in fine-tuning.
>
> **To better analyze OCR's generalization versus hallucination effects, we expanded the original synthetic dataset with two relation pairs: _City-Language_ and _City-Language (CF)_.** The *City-Language* pair uses real-world associations likely learned during pretraining (e.g., "People in Paris speak French"), testing generalization capabilities. In contrast, *City-Language (CF)* creates counterfactual associations by deliberately mispairing cities with incorrect languages (e.g., mapping "Paris" to "Japanese"), designed to analyze hallucination alongside the four other fictitious relation types.
>
> Below are the updated results for the newly added relation pairs. The first column reports performance on the generalization task (*City-Language*), while the remaining columns show hallucination cases across four counterfactual relations. We measure the mean rank, where the rank indicates the position of the ground-truth answer among all candidates based on prediction probability. Lower ranks indicate better OCR capabilities. Values in parentheses indicate the standard error of the mean rank, calculated from 3 runs with different random seeds.
>
> Models           | *Gen: City-Language* | *Hall: City-Language (CF)* | *Hall: PoB-Sport* | Hall: Country-Code | Hall: Profession-Color | Hall: Sport-Music
> -----------------|--------------------|-------------------------|-----------------|--------------------|------------------------|-------------------
> Gemma-2-9B       | 0.00 (0.00)        | 0.19 (0.20)             | 0.68 (0.27)     | 0.19 (0.07)        | 1.64 (0.01)            | 0.56 (0.01)
> OLMo-7B          | 0.07 (0.03)        | 1.33 (0.49)             | 1.41 (0.29)     | 0.15 (0.13)        | 1.84 (0.23)            | 0.17 (0.01)
> Qwen-2-7B        | 0.13 (0.01)        | 4.55 (2.33)             | 3.30 (0.62)     | 3.63 (1.10)        | 0.82 (0.34)            | 0.40 (0.08)
> Mistral-7B-v0.3  | 0.00 (0.00)        | 2.10 (0.01)             | 1.36 (0.70)     | 1.48 (0.52)        | 1.15 (0.56)            | 1.28 (0.13)
> Llama-3-8B       | 0.00 (0.00)        | 1.18 (0.61)             | 2.29 (4.26)     | 0.77 (0.10)        | 0.93 (0.21)            | 0.63 (0.22)
>
> The new experimental results reveal two key findings:
> - **LLMs continue to exhibit OCR‑driven hallucinations on real‑world data.** As observed in the paper, OCR behavior is highly sensitive to the learning rate; we have not yet conducted an exhaustive sweep due to time constraints. Consequently, we believe the performance can be further improved with more thorough hyperparameter tuning.
>
> - **We find that the generalization performance is better than the hallucination performance**—presumably because the injected causal facts align with the model’s pretrained knowledge and are therefore easier to learn. In contrast, in the hallucination setup (especially on PopQA‑OCR), the new information often collides with the model’s pretrained knowledge, making it harder to acquire. **This gap underscores the need for a specialized dataset to systematically evaluate OCR.** Importantly, our synthetic dataset can avoid such conflicts by using fictitious names and can be scaled up by adding more subject–relation pairs and by incorporating richer, multi‑step causal graphs (as suggested by Reviewer XZxz) to meet this need. We view this expansion as an exciting avenue for future work.
>
>
> > Limited analysis of practical implications; I mean the authors should step forward to investigate how to design anti-hallucination techniques from the view of OCR.
>
> This is an excellent question. Our analysis suggests that because both generalization and hallucination arise from the same underlying mechanism, naively reducing hallucination risks degrading benign generalization. To further analyze this, we drew on mechanistic interpretability tools—specifically the logit lens [2]—which project each hidden state back into vocabulary logits via the output embedding matrix. We measured the layer at which the correct target label’s logit at the final sequence position first exceeds $0.01$. The results below reveal a clear contrast between generalization and hallucination cases:
>
> | Models | Generalization (City-Language) | Hallucination (Country-Code) |
> |---|---|---|
> | Gemma-2-9B | 28 | 4 |
> | Olmo-7B | 21 | 6 |
>
> **Strikingly, in generalization cases (e.g., *city-language*), the correct label’s logit consistently emerges in the deeper layers, whereas in hallucination cases it peaks much earlier.** This layer‑wise separation indicates a viable strategy to distinguish—and thus mitigate—hallucination without harming generalization. While designing concrete mitigation techniques lies beyond the scope of this work, it indicates a promising direction for future research. The layer-wise separation we uncover offers a solid foundation for developing targeted approaches that suppress hallucinations without impairing generalization.
>
> > More different model structures should be tested in the experiments.
>
> In our LLM experiments, we already evaluated several transformer-based families of varying architectures to validate our theoretical findings. Due to computational constraints, however, we were unable to include even larger-scale models in this study. We agree that extending our evaluation to additional—and particularly larger—model classes would be valuable, and we plan to explore this in future work as more compute becomes available
>
> > From the view of OCR, how can we design models to improve the generalization rather than hallucination?
>
> Please see our response to the second weakness above.
>
> > Can the authors construct a benchmark to test the OCR problem?
>
> Please see our response to the first weakness above.
>
> ---
>
> **References:**
>
> [1] Mallen, A., Asai, A., Zhong, V., Das, R., Khashabi, D., and Hajishirzi, H. When not to trust language models: Investigating effectiveness of parametric and non-parametric memories, 2023.
>
> [2] nostalgebraist. 2020. interpreting gpt: the logit lens.

---

> > ### Comment · Reviewer_6AQL · 2025-08-01
> > **Read the responses**
> >
> > Thanks for your responses. By considering the authors' rebuttals and comments from other reviewers, I decide to keep my ratings.

---

> ### Author Response · Authors · 2025-08-03
>
> Dear Reviewer 6AQL,
>
> Thank you again for your insightful comments and for spending time reviewing our responses. In response to your questions, we have:
>
> - Extended our evaluation to a real-world LLM dataset (PopQA).
>
> - Presented a logit-lens analysis to distinguish hallucination from generalization and highlighted a potential approach for mitigating hallucination in OCR.
>
> At your convenience, we would greatly appreciate any further feedback or questions you may have. We hope these revisions have addressed your concerns and are happy to clarify any remaining points. We know this is a busy time and sincerely appreciate the care you have devoted to our manuscript.

---

### Official Review · Reviewer_XZxz · 2025-07-02

**Clarity:** 3
**Significance:** 2
**Originality:** 3
**Rating:** 4
**Confidence:** 3

**Summary:**

This paper empirically and theoretically demonstrates that a single-layer linear attention can perform out-of-context reasoning (OCR). Specifically, the transformer can predict $s',r_2\rightarrow b$ given $s,r_1\rightarrow a$,   $s,r_2\rightarrow b$,  and $s',r_1\rightarrow a$. Further experiments on synthetic tasks prove that real-world LLMs also exhibit such OCR capability. However, such capability can cause hallucinations once the associations in training data are not causal relationships, revealing a potential cause of the hallucinations.

**Questions:**

Could you extend the identified capability of transformers to perform the OCR task defined in your paper to more general reasoning modes? I think either empirically or theoretically extending your findings to explain the failure or capability in more general reasoning patterns will be helpful to improve your work.

**Ethical Concerns:**

["NO or VERY MINOR ethics concerns only"]

**Final Justification:**

Thank you for the clarification. After reading your response, I better understand the challenges involved in extending OCR capabilities to more complex scenarios. I also appreciate your refinement of the term “generalization” into the more precise and rigorous notion of “OCR capability,” which appropriately addresses the concerns around its interpretation. Given the nontrivial nature of theoretically establishing OCR capability and its fundamental significance, I have decided to raise my score to 4.

**Limitations:**

yes

**Quality:**

3

**Strengths And Weaknesses:**

**Strengths**

1. This work provides novel insights into the formation of hallucinations in LLMs by theoretically proving that a single-layer linear attention is capable of achieving OCR.

2. This work theoretically reveals the defect of the parameterization of the output-value matrix widely adopted by previous works.

3. The paper is well-written and easy to follow.

**Weaknesses**

The theory and experiments consider a very simple case where only two facts are correlated. While I acknowledge that such a setup may reflect certain real-world situations, it fails to capture the complexity of most reasoning patterns. In more realistic cases, the "facts" can be modeled as a causal graph with multiple variables and relationships. For instance, in your OCR training data, since $s,r_1\rightarrow a \Rightarrow s,r_2\rightarrow b$, we can imagine a causal graph in which there exists a confounder $C$ such that $C\rightarrow A$ and $C\rightarrow B$, where event $A$ represents  $s,r_1\rightarrow a$ and event $B$ represents $s, r_2\rightarrow b$. $A$ and $B$ are correlated through $C$. However, this represents only a rudimentary causal graph and offers limited applicability to more complex scenarios.

Moreover, according to the definition in your paper, whether the model "generalizes" or "hallucinates" depends on whether the correlation between $A$ and $B$ is causal. Specifically, if $A\rightarrow B$ across training and test data, then the model "generalizes"; otherwise, the model produces hallucinations. In my opinion, however, "generalization" should refer to the model's ability to learn generalizable features or relationships, such as capturing a causal link in a causal graph, so it may not be suitable to regard the capability of successfully solving the tasks in Fig. 5 as "generalization".

---

> ### Author Rebuttal · Authors · 2025-07-31
>
> We thank the reviewer for their insightful feedback and for highlighting our novel theoretical insights, our analysis of the common output-value reparameterization defect, and the clarity of our presentation. Here are our responses to your concerns:
>
> > The theory and experiments consider a very simple case where only two facts are correlated. While I acknowledge that such a setup may reflect certain real-world situations, it fails to capture the complexity of most reasoning patterns. In more realistic cases, the "facts" can be modeled as a causal graph with multiple variables and relationships. For instance, in your OCR training data, since $s,r_1\rightarrow a \Rightarrow s,r_2\rightarrow b$, we can imagine a causal graph in which there exists a confounder $C$ such that $C\rightarrow A$ and $C\rightarrow B$, where event $A$ represents $s,r_1\rightarrow a$ and event $B$ represents $s, r_2\rightarrow b$. $A$ and $B$ are correlated through $C$. However, this represents only a rudimentary causal graph and offers limited applicability to more complex scenarios.
>
> Although our setup is simple, it captures one of the most fundamental causal-graph structures, and its analysis is both meaningful and highly nontrivial. As mentioned in Section 1.1, prior work has examined OCR from multiple angles [1–4], yet a deep theoretical understanding remains elusive. **As a first step, our study provides rigorous theoretical analysis and practical insights that help unify generalization and hallucination.** We agree that extending this framework to more complex causal graphs is an important next step, but it requires highly nontrivial effort (since the analysis of our setting already requires very sophisticated techniques) and is beyond the scope of our paper. We also emphasize that building a solid fundation on the most fundamental setup is of equal importance, which is the main focus of our paper.
>
> **Moreover, our theoretical analysis provides new insights into the minimization of non-smooth norms**—a central component of matrix completion algorithms. This topic is under-explored in the literature [5], largely because the non-smooth nature of the nuclear norm hinders rigorous analysis (Similar issues in the vector form would be LASSO, which is already extremely complex in theory). By exploiting a key observation on the symmetry induced by the minimization process (Lemmas 7-10), our paper offers one of the first concrete characterizations of the solutions to nuclear norm minimization.
>
>
> > Moreover, according to the definition in your paper, whether the model "generalizes" or "hallucinates" depends on whether the correlation between $A$ and $B$ is causal. Specifically, if $A\rightarrow B$ across training and test data, then the model "generalizes"; otherwise, the model produces hallucinations. In my opinion, however, "generalization" should refer to the model's ability to learn generalizable features or relationships, such as capturing a causal link in a causal graph, so it may not be suitable to regard the capability of successfully solving the tasks in Fig. 5 as "generalization".
>
> We agree that framing generalization as “the model’s ability to learn truly generalizable features or relationships—e.g., capturing an actual causal link in a causal graph”—is indeed consistent with prior OCR work [4]. However, this raises the issue that a model could “learn” spurious rules (for instance, “If Raul lives in France, then Raul codes in Java”) even when there is no true causal link, which we would classify as hallucination rather than genuine generalization. To distinguish these two behaviors in LLMs, we therefore introduce a more precise criterion based on the presence or absence of a causal relationship between the two relations. In this paper, we show that both hallucination on causally unrelated knowledge and true generalization on causal links arise from the same underlying mechanism—OCR. Finally, we appreciate the reviewer’s suggestion to correct the caption of Figure 2 and 5: we will replace “generalization” with “OCR capability” in the revision.
>
> > Could you extend the identified capability of transformers to perform the OCR task defined in your paper to more general reasoning modes? I think either empirically or theoretically extending your findings to explain the failure or capability in more general reasoning patterns will be helpful to improve your work.
>
> Please see our response to the first weakness above.
>
> ---
>
> **References:**
>
> [1] Dmitrii Krasheninnikov, Egor Krasheninnikov, and David Krueger. Out-of-context meta-learning in large language models. In ICLR 2023 Workshop on Mathematical and Empirical Understanding of Foundation Models, 2023.
>
> [2] Lukas Berglund, Asa Cooper Stickland, Mikita Balesni, Max Kaufmann, Meg Tong, Tomasz Korbak, Daniel Kokotajlo, and Owain Evans. Taken out of context: On measuring situational awareness in llms. arXiv preprint arXiv:2309.00667, 2023a.
>
> [3] Zeyuan Allen-Zhu and Yuanzhi Li. Physics of language models: Part 3.2, knowledge manipulation. arXiv preprint arXiv:2309.14402, 2023.
>
> [4] Jiahai Feng, Stuart Russell, and Jacob Steinhardt. Extractive structures learned in pretraining enable generalization on finetuned facts. arXiv preprint arXiv:2412.04614, 2024.
>
> [5] Tim Hoheisel and Elliot Paquette. Uniqueness in nuclear norm minimization: Flatness of the nuclear norm sphere and simultaneous polarization. Journal of Optimization Theory and Applications, 197(1):252–276, Feb 2023. doi: https://doi.org/10.1007/s10957-023-02167-7. URL https://link.springer.com/article/10.1007/s10957-023-02167-7.

---

### Official Review · Reviewer_7jeV · 2025-07-02

**Clarity:** 3
**Significance:** 3
**Originality:** 3
**Rating:** 5
**Confidence:** 3

**Summary:**

This paper explores how transformer models can both generalize from new information and hallucinate incorrect facts. Specifically, it investigates a phenomenon called Out-of-Context Reasoning (OCR) where a model makes inferences based on partial patterns seen during training. The authors ask whether this same reasoning mechanism is responsible for both useful generalization and harmful hallucination. They theoretically analyze a very simple model and surprisingly show that even this basic model can perform OCR, learning patterns from just a few examples. Their analysis shows that this ability comes from the way the model is trained, specifically how gradient descent favors certain types of solutions that can link facts together. However, this also means the model may make false connections when two things only appear together but aren’t truly related.

**Questions:**

See above

**Ethical Concerns:**

["NO or VERY MINOR ethics concerns only"]

**Final Justification:**

All my concerns have been properly addressed. I raised my score from 4 to 5.

**Limitations:**

The authors adequately discuss the technical limitations of their work, particularly the focus on one-layer transformers and the use of synthetic data.

**Quality:**

3

**Strengths And Weaknesses:**

Strengths
- Interesting topic that investigates why LLMs can both generalize well and hallucinate.
- Shows that even a one-layer attention-only transformer can perform OOC reasoning
- Connects model behavior to gradient descent’s implicit bias.


Weaknesses
- Focuses only on one-layer transformers; unclear whether the conclusions fully apply to complex models.
- It is fine to use synthetic data for evaluation, but it would be helpful to see and discuss how the model performs on tasks such as question answering.
- Identifies the cause of hallucination that is a good result, but how can we reduce or prevent it, based on the analysis provided in the paper?
- The proposed optimization as well as the use of SVMs are difficult to follow, could the authors provide some clarifications?

---

> ### Author Rebuttal · Authors · 2025-07-31
>
> We thank the Reviewer for their thoughtful feedback and for recognizing our theoretical contributions to understanding generalization and hallucination in LLMs.
>
> > Weakness 1
>
> We focus on a one-layer transformer for our theoretical analysis since a one-layer transformer is sufficient to perform the OCR task and discover the underlying mechanism. Despite its simplicity, previous work doesn’t have a deep understanding of the mechanism and our theoretical analysis is highly non-trivial and provides practical insights on linking generalization and hallucination.
> Although the result might not be fully the same for complex models, our experiments for LLMs show that the theoretical results also apply to more complex models. We further conducted a probing experiment to verify this. Figure 7 (right) in Appendix D gives two important findings for one-layer transformers.
> - (C1) The prediction is made by summing up the feature vector of subject $s\_i$ and relation $r\_1$ or $r\_2$.
> - (C2) Subjects that correspond to the same fact-implication pair will point to the same direction in hidden representation.
>
> **We hypothesis that these two findings also hold in LLMs.** We collect the hidden states at the last position in the last layer for all subjects with two different relations (*country-code*) - termed as **fact states**, $H\_1$ and **implication states**, $H\_2$, where $H\_1, H\_2 \in \mathbb{R}^{|S| \times d}$
> can be understood as the prediction vector. We introduce three metrics for verifying the two claims in LLMs.
> - **Average-pairwise correlation (APC)** measures how consistently vectors within the same group point in the same direction.
> $$
>  \mathrm{APC}
>  = \frac{2}{b(b-1)}
>  \sum\_{1 \le i < j \le b}
>  \frac{h\_i^\top h\_j}{||h\_i||||h\_j||},
>  $$
>  where $h\_i, h\_j \in \mathbb{R}^d$ and $b$ is group size. $\mathrm{APC}=1$ iff all vectors are colinear. To verify **Claim C1**, we can compute the difference matrix $W\_1 = H\_1 - H\_2$. If C1 holds, then every row of $W\_1$ will align with the direction of $r\_1 - r\_2$, i.e., $\rm APC(W\_1) \approx 1$.
> - **Intra-group variance** quantifies the spread of each group around its own mean, averaged over $K$ groups.
>  $$
>  V\_{\mathrm{intra}}
>  = \frac{1}{K}\sum\_{k=1}^K
>  \frac{1}{n\_k}\sum\_{i:g\_i=k}
>  ||h\_i - \mu^{(k)}||^2,
>  $$
>  where $\displaystyle \mu^{(k)} = \frac{1}{n\_k}\sum\_{g\_i=k}h\_i$ is the $k$-th group mean.
> - **Inter-group variance** measures how far apart the group centroids are from the global mean.
>  $$
>  V\_{\mathrm{inter}}
>  = \frac{1}{K}\sum\_{k=1}^K
>  ||\mu^{(k)} - \mu||^2,
>  \quad
>  \mu = \frac{1}{b}\sum\_{i=1}^b h\_i.
>  $$
>  A larger $V\_{\mathrm{inter}}$ indicates greater separation between groups. To verify **Claim C2**, we form the “sum” feature for each example: $W\_{2} = H\_{1} + H\_{2}$. We then partition these vectors into $n$ groups by their fact–implication relation pair.If $r\_{1}\approx -r\_{2}$, then
>
>   $$
>   [W\_{2}]\_i
>   =(s\_i + r\_1)+(s\_i + r\_2)
>   \approx 2s\_i,
>   $$
>
> so that
> - $V\_{\mathrm{intra}}\approx 0$ (all vectors in each group collapse to the same direction), and
> - $V\_{\mathrm{inter}}>0$ (different subjects $s\_i$ yield distinct directions).
>  - **When $r\_1 \neq -r\_2$:** The constant shift $r\_1 + r\_2$ appears in every row of $W\_{2}$, but adding a uniform vector to all samples does *not* change either intra‐ or inter‐group variance, since variance is invariant under translation.
>
> The results for Gemma-2-9B and Olmo-7B on relation pair *country-code* is shown below:
> - Result of Gemma-2-9B
>
> | Metrics| Epoch 0 | Epoch 40 | Epoch 100 |
> |-------------|---------|----------|-----------|
> |APC| 0.89	| 0.93 	| 0.94|
> | Intra-group variance | 0.20	| 0.02	| 0.02|
> | Inter-group variance| 0.0	| 0.02 	| 0.28|
>
> - Result of Olmo-7B
>
> |Metrics| Epoch 0 | Epoch 40 | Epoch 100 |
> |---|--------|--|---|
> | APC 	| 0.82	| 0.87 	| 0.88	|
> | Intra-group variance 	| 0.24	| 0.10 	| 0.09	|
> | Inter-group variance 	| 0.01	| 0.95 	| 0.94	|
>
> We observe three consistent trends across both models as training proceeds from Epoch 0 → 40 → 100:
> - The average pairwise correlation steadily increases, showing that vectors within each fact–implication group become more tightly aligned.
> - Intra-group variance drops sharply, indicating that same-group representations collapse toward a common direction.
> - Inter-group variance rises, meaning that different subject groups diverge from one another.
>
> > Weakness 2
>
> Thanks for the valuable suggestion. We have extended the LLM experiments to PopQA [1], which is a large-scale open-domain question answering (QA) dataset. We show that LLMs continue to exhibit OCR‑driven hallucinations on real‑world data. Due to the character limit, please see our response to Reviewer PxpD for further details.
>
> > Weakness 3
>
> This is an excellent question. Our analysis suggests that because both generalization and hallucination arise from the same underlying mechanism, naively reducing hallucination risks degrading benign generalization. To further analyze this, we drew on mechanistic interpretability tools—specifically the logit lens [2]—which project each hidden state back into vocabulary logits via the output embedding matrix. We measured the layer at which the correct target label’s logit at the final sequence position first exceeds $0.01$. The results below reveal a clear contrast between generalization and hallucination cases:
>
> | Models | Generalization (City-Language) | Hallucination (Country-Code) |
> |---|---|---|
> | Gemma-2-9B | 28 | 4 |
> | Olmo-7B | 21 | 6 |
>
> **Strikingly, in generalization cases (i.e., *city-language*), the correct label’s logit consistently emerges in the deeper layers, whereas in hallucination cases it peaks much earlier.** This layer‑wise separation indicates a viable strategy to distinguish—and thus mitigate—hallucination without harming generalization. While designing concrete mitigation techniques lies beyond the scope of this work, it indicates a promising direction for future research. The layer-wise separation we uncover offers a solid foundation for developing targeted approaches that suppress hallucinations without impairing generalization.
>
> > Weakness 4
>
> Thanks for pointing that out. We hereby post two clarifications:
> 1. **Link with SVM:** Soudry et al. [3] first showed that gradient descent on exponential‑tailed losses (e.g., logistic regression) converges to the maximum‑margin solution of an equivalent SVM: it separates positive and negative logits under margin constraints. A one‑layer transformer trained with cross‑entropy loss—once its attention weights are fixed—behaves like a multiclass extension of logistic regression. Building on this insight [4, 5], we derive Theorem 1: the factorized and non‑factorized models correspond to solutions of two different SVM objectives, which we term the $W\_{OV}^F$‑SVM and the $W\_{OV}$‑SVM, respectively.
> 2. **Difference in optimizing the factorized vs. non-factorized models:** As shown in Theorem 1, the factorized model’s SVM objective takes the form:
> $$
> \min\_{W\_O,W\_V} \frac{1}{2}(||W\_O||\_F^2 + ||W\_V||\_F^2)
> \quad \text{subject to margin constraints on }W\_O W\_V^\top.
> $$
>
> Let $||W||\_*$ denote its nuclear norm. Because
>
> $$
> ||W\_{OV}^F||\_*^2 =\min\_{W\_O W\_V^\top = W\_{OV}^F}\frac{1}{2}(||W\_O||\_F^2 + ||W\_V||\_F^2),
> $$
>
> it is equivalent to minimizing the nuclear norm of $W\_{OV}^F$ with margin constraints on it.
> By contrast, the non‑factorized model solves:
>
> $$
> \min\_{W\_{OV}}\frac{1}{2}||W\_{OV}||\_F^2
> \quad\text{subject to those identical constraints.}
> $$
>
> The two formulations differ only in whether they penalize the Frobenius norm or the nuclear norm which is the sum of singular values. **This shift—from a Frobenius‑norm penalty to a nuclear‑norm penalty—is what ultimately drives the divergence in their OCR behaviors:**
>
> We can view the optimization problem as completing a lookup table represented by the matrix $W$, where each column indexes either a subject or a relation, and each row corresponds to a fact or an implication. Because the Frobenius norm is the sum of the squares of all entries in $W$, any entry not involved in a training constraint is driven to be zero to minimize the objective. Since test implications do not appear in the training data, their corresponding entries remain zero. This explains why the red‑boxed region of the non‑factorized model $W_{OV}$ in Figure 4 consists of zeros.
>
> In contrast, the nuclear norm is defined as the sum of a matrix’s singular values, with each singular value representing an independent orthogonal direction of the transformation. If we were to force the red‑boxed region (in Figure 4) of the factorized model $W_O W_V^\top$ to be zero, we would introduce additional orthogonal components, thereby increasing the nuclear norm. To avoid this penalty, the factorized model naturally assigns nonzero values in that region.
> In summary, the difference in the objectives of the two SVM problems leads to two different solutions, which further results in different OCR capabilities. We will include the above two points in the extra page in the camera-ready version.
>
> ---
>
> **References:**
>
> [1] Mallen, A., Asai, A., Zhong, V., Das, R., Khashabi, D., and Hajishirzi, H. When not to trust language models: Investigating effectiveness of parametric and non-parametric memories, 2023.
>
> [2] nostalgebraist. 2020. interpreting gpt: the logit lens.
>
> [3] Daniel Soudry, Elad Hoffer, Mor Shpigel Nacson, Suriya Gunasekar, and Nathan Srebro. The implicit bias of gradient descent on separable data. Journal of Machine Learning Research, 19(70):1–57, 2018.
>
> [4] Kaifeng Lyu and Jian Li. Gradient descent maximizes the margin of homogeneous neural networks. arXiv preprint arXiv:1906.05890, 2019.
>
> [5] Gal Vardi, Ohad Shamir, and Nati Srebro. On margin maximization in linear and relu networks. Advances in Neural Information Processing Systems, 35:37024–37036, 2022.

---

> > ### Comment · Reviewer_7jeV · 2025-08-04
> >
> > Thanks for the clarifications. I am alright with these points.

---

> ### Author Response · Authors · 2025-08-03
>
> Dear Reviewer 7jeV,
>
> Thank you again for your insightful comments. In response to your questions, we have:
>
> - Extended our evaluation to a real-world LLM dataset (PopQA),
>
> - Presented a logit-lens analysis to distinguish hallucination from generalization and highlighted a potential approach for mitigating hallucination in OCR.
>
> - Provided a more detailed explanation of our theoretical results, and
>
> - Verified our theoretical claims in real LLMs using a probing experiment
>
> At your convenience, we would greatly appreciate any further feedback or questions you may have. We believe we have addressed your previous concerns and are happy to clarify any remaining points.
>
> We know this is a busy period and sincerely appreciate the time and care you have devoted to our manuscript.

---

### Note · Authors · 2025-08-12

We are grateful to the reviewers for their thorough feedback. Their insights have helped us clarify our contributions and strengthen our work on understanding generalization and hallucination in transformers. We are encouraged by the reviewers' recognition of several strengths:
- **The paper is well-motivated**, tackling the key challenge of why LLMs generalize yet also hallucinate (7jev, 6AQL, PxpD).
- **We provided a solid theoretical foundation**, showing how Out-of-Context Reasoning emerges in transformers and is linked to hallucination (7jev, XZxz, 6AQL, PxpD).
- **Our theory offered novel insights** which connects model behavior to the implicit bias of gradient descent and reveals defects in common reparameterizations (7jev, XZxz).
- **The synthesis of theory and experiments** provides a unified explanation for these seemingly contradictory behaviors (6AQL, PxpD).

We summarize the main concerns and how we addressed them:
- **Evaluation on real-world data (7jeV, 6AQL, PxpD):**
We added experiments on the ***large-scale PopQA dataset***, which validates our claims on real-world data. We stress that synthetic data remains crucial for controlled OCR evaluation, as it avoids knowledge collision. We plan to expand our synthetic data into a public benchmark.
- **Mitigating hallucination (7jeV, 6AQL):**
To explore this direction, we performed a logit lens analysis and found ***a clear layer-wise separation***: correct logits emerge late in generalization but peak early in hallucination. This distinct internal signature indicates a path for mitigating hallucinations without harming generalization, which we leave for future work.
- **Causal graph setup (XZxz):**
We emphasize that our setup is fundamental. Our main contribution is providing ***the first rigorous theoretical analysis for this core causal structure***, a non-trivial task that remained elusive to prior work. Furthermore, our analysis introduces novel insights for non-smooth norm minimization, with broad applicability to fields such as matrix completion.

We will incorporate all of these new results, analyses, and clarifications into the camera-ready version of the manuscript. We are grateful that our rebuttal successfully addressed the reviewers' concerns. This was explicitly acknowledged by Reviewers 7jeV and XZxz during the discussion, and importantly, **no new or outstanding concerns were raised by any reviewer.** This positive outcome is also reflected in the score raise from Reviewer XZxz.

---

### Decision · Program_Chairs · 2025-09-17

**Decision:**

Accept (poster)

**Comment:**

The paper studies how Transformers learn to deduce implications from facts seen during training. In particular, the authors analyze a simplified one-layer Transformer with fixed attention weights (essentially reduces to a logistic regression problem) and show that the inductive bias of gradient descent to low nuclear norm solution leads to solutions that link facts together resulting in both generalization and hallucinations. This crucially utilizes the factored representation of the value and output weight matrices. The authors support their hypothesis through evaluation on LLMs using mostly synthetic data (extended to real datasets during rebuttal period).

The problem being considered is interesting and relevant, and the formulation is clean. While the theoretical analysis seems mostly based on prior work and in a simplified setting (attention weights fixed), the paper does add to our understanding of the relationship of different parameterizations and their impact on OOD performance, that may be of practical relevance. Therefore, I recommend accept.

I encourage the authors to make their exposition more transparent and correctly positioned with respect to prior work. In particular,
-  With the attention fixed, the problem reduces to matrix factorization with a rank-1 structure with factored parameterization versus full matrix parameterization. This is well-studied in the gradient descent on matrix factorization literature, and the authors should cite this line of work and be transparent about their own contributions. The authors should also clarify the importance of Transformers for their results, given that the core attention-component is frozen.
- Additionally, there has been work on understanding OOD performance based on implicit bias [1,2]. I encourage the authors to discuss these works.

[1] Abbe, E., Bengio, S., Cornacchia, E., Kleinberg, J., Lotfi, A., Raghu, M. and Zhang, C., 2022. Learning to reason with neural networks: Generalization, unseen data and boolean measures. Advances in Neural Information Processing Systems, 35, pp.2709-2722.
[2] Abbe, E., Bengio, S., Lotfi, A. and Rizk, K., 2024. Generalization on the unseen, logic reasoning and degree curriculum. Journal of Machine Learning Research, 25(331), pp.1-58.